# Convolutions and More as Einsum: A Tensor Network Perspective with Advances for Second-Order Methods

**Felix Dangel**
Vector Institute
Toronto, Canada
fdangel@vectorinstitute.ai

## Abstract

Despite their simple intuition, convolutions are more tedious to analyze than dense layers, which complicates the transfer of theoretical and algorithmic ideas to convolutions. We simplify convolutions by viewing them as tensor networks (TNs) that allow reasoning about the underlying tensor multiplications by drawing diagrams, manipulating them to perform function transformations like differentiation, and efficiently evaluating them with einsum. To demonstrate their simplicity and expressiveness, we derive diagrams of various autodiff operations and popular curvature approximations with full hyper-parameter support, batching, channel groups, and generalization to any convolution dimension. Further, we provide convolution-specific transformations based on the connectivity pattern which allow to simplify diagrams before evaluation. Finally, we probe performance. Our TN implementation accelerates a recently-proposed KFAC variant up to 4.5 x while removing the standard implementation's memory overhead, and enables new hardware-efficient tensor dropout for approximate backpropagation.

## 1 Introduction

Convolutional neural networks [CNNs, 39] mark a milestone in the development of deep learning architectures as their 'sliding window' approach represents an important inductive bias for vision tasks. Their intuition is simple to explain with graphical illustrations [e.g. 21]. Yet, convolutions are more challenging to analyze than dense layers in multi-layer perceptrons (MLPs) or transformers [71]. One reason is that they are hard to express in matrix notation and—even in index notation—compact expressions that are convenient to work with only exist for special hyper-parameters [e.g. 27, 2]. Many hyper-parameters (stride, padding, ...) and additional features like channel groups [36] introduce even more complexity that is inherited by related routines, e.g. for autodiff. We observe a delay of analytic and algorithmic developments between MLPs vs. CNNs, e.g.

- Approximate Hessian diagonal: 1989 vs. 2024
- Hessian rank: 2021 vs. 2023
- Gradient descent learning dynamics: 2014 vs. 2023
- Neural tangent kernel (NTK): 2018 vs. 2019
- Kronecker-factored quasi-Newton methods: 2021 vs. 2022
- Kronecker-factored curvature (KFAC, KFRA, KFLR): (2015, 2017, 2017) vs. (2016, 2020, 2020)

The software support for less standard routines some of these methods require also reflects this gap. Some functions only support special dimensions [15]. Others use less efficient workarounds (§5.1) or are not provided at all (§B.4). And they are hard to modify as the code is either closed-source [12] or

38th Conference on Neural Information Processing Systems (NeurIPS 2024).

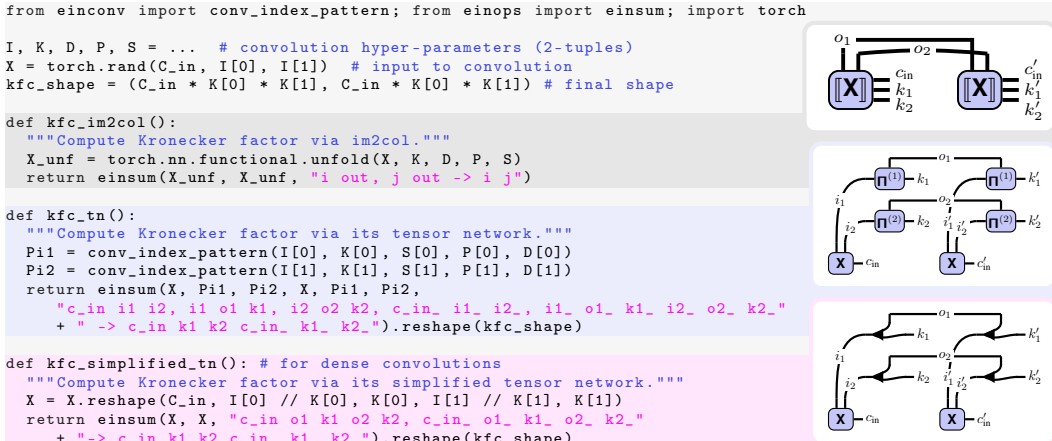

```
from einconv import conv_index_pattern; from einops import einsum; import torch

I, K, D, P, S = ...    # convolution hyper-parameters (2-tuples)
X = torch.rand(C_in, I[0], I[1])   # input to convolution
kfc_shape = (C_in * K[0] * K[1], C_in * K[0] * K[1]) # final shape

def kfc_im2col():
    """Compute Kronecker factor via im2col."""
    X_unf = torch.nn.functional.unfold(X, K, D, P, S)
    return einsum(X_unf, X_unf, "i out, j out -> i j")

def kfc_tn():
    """Compute Kronecker factor via its tensor network."""
    Pi1 = conv_index_pattern(I[0], K[0], S[0], P[0], D[0])
    Pi2 = conv_index_pattern(I[1], K[1], S[1], P[1], D[1])
    return einsum(X, Pi1, Pi2, X, Pi1, Pi2,
        "c_in i1 i2, i1 o1 k1, i2 o2 k2, c_in_ i1_ i2_, i1_ o1_ k1_ i2_ o2_ k2_"
        + " -> c_in k1 k2 c_in_ k1_ k2_").reshape(kfc_shape)

def kfc_simplified_tn(): # for dense convolutions
    """Compute Kronecker factor via its simplified tensor network."""
    X = X.reshape(C_in, I[0] // K[0], K[0], I[1] // K[1], K[1])
    return einsum(X, X, "c_in o1 k1 o2 k2, c_in_ o1_ k1_ o2_ k2_"
        + "-> c_in k1 k2 c_in_ k1_ k2_").reshape(kfc_shape)
```

Figure 1: Many convolution-related routines can be expressed as TNs and evaluated with `einsum`. We illustrate this for the input-based factor of KFAC for convolutions [KFC, 27], whose standard implementation (*top*) requires unfolding the input (high memory). The TN (*middle*) enables internal optimizations inside `einsum` (e.g. with contraction path optimizers like `opt_einsum` [66]). (*Bottom*) In many cases, the TN further simplifies due to structures in the index pattern, which reduces cost.

written in a low-level language. This complicates the advance of existing, and the exploration of new, algorithmic ideas for convolutions.

Here, we seek to reduce this complexity gap by viewing convolutions as tensor networks [TNs, 53, 6, 9] which express the underlying tensor multiplications as diagrams. These diagrams are simpler to parse than mathematical equations and can seamlessly be (i) manipulated to take derivatives, add batching, or extract sub-tensors, (ii) merged with other diagrams, and (iii) evaluated with `einsum`. This yields simple, modifiable implementations that benefit from automated under-the-hood-optimizations for efficient TN contraction developed by the quantum simulation community [e.g. 66, 25, 74, 13], like finding a high-quality contraction order or distributing computations:

1. We use the TN format of convolution from Hayashi et al. [29] to derive diagrams and `einsum` formulas for autodiff and less standard routines for curvature approximations with support for all hyper-parameters, batching, groups, and any dimension (Table 1).

2. We present transformations based on the convolution's connectivity pattern to re-wire and symbolically simplify TNs before evaluation (example in Figure 1).

3. We compare default and TN implementations, demonstrating optimal peak memory reduction and run time improvements up to 4.5 x for a recent KFAC variant, and showcase their flexibility to impose hardware-efficient dropout for randomized backpropagation.

Our work not only provides simpler perspectives and implementations that facilitate the exploration of algorithmic ideas for convolutions, but also directly advances second-order methods like KFAC: It enables more frequent pre-conditioner updates, using larger batches without going out of memory, and extending KFAC to transpose convolution. These improvements are important for second-order optimization and other applications like Laplace approximations [20] and influence functions [28].

## 2   Preliminaries

We briefly review 2d convolution (§2.1), tensor multiplication and `einsum` (§2.2), then introduce the graphical TN notation and apply it to convolution (§2.3). Bold lower-case ($a$), upper-case ($A$), and upper-case sans-serif ($\mathbf{A}$) symbols indicate vectors, matrices, and tensors. Entries follow the same convention but use regular font weight; $[\cdot]$ denotes slicing ($[A]_{i,j} = A_{i,j}$). Parenthesized indices mean reshapes, e.g. $[a]_{(i,j)} = [A]_{i,j}$ with $a$ the flattened matrix $A$.

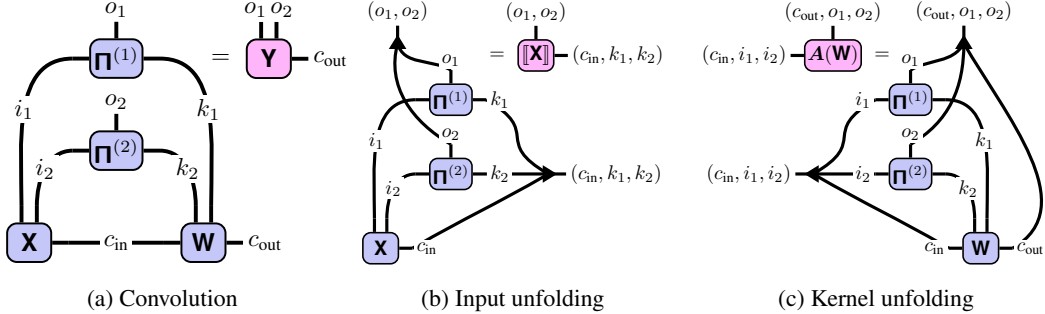

|  | (a) Convolution | (b) Input unfolding | (c) Kernel unfolding |

Figure 2: TNs of (a) 2d convolution and (b,c) connections to its matrix multiplication view. The connectivity along each dimension is explicit via an index pattern tensor $\Pi$.

## 2.1 Convolution

2d convolutions process channels of 2d signals $\mathbf{X} \in \mathbb{R}^{C_{\text{in}} \times I_1 \times I_2}$ with $C_{\text{in}}$ channels of spatial dimensions[1] $I_1, I_2$ by sliding a collection of $C_{\text{out}}$ filter banks, arranged in a kernel $\mathbf{W} \in \mathbb{R}^{C_{\text{out}} \times C_{\text{in}} \times K_1 \times K_2}$ with kernel size $K_1, K_2$, over the input. The sliding operation depends on various hyper-parameters [padding, stride, ..., see 21]. At each step, the filters are contracted with the overlapping area, yielding the channel values of a pixel in the output $\mathbf{Y} \in \mathbb{R}^{C_{\text{out}} \times O_1 \times O_2}$ with spatial dimensions $O_1, O_2$. Optionally, a bias from $\boldsymbol{b} \in \mathbb{R}^{C_{\text{out}}}$ is added per channel.

One way to implement convolution is via matrix multiplication [10], similar to fully-connected layers. First, one extracts the overlapping patches from the input for each output, then flattens and column-stacks them into a matrix $[\![\mathbf{X}]\!] \in \mathbb{R}^{C_{\text{in}} K_1 K_2 \times O_1 O_2}$, called the *unfolded input* (or im2col). Multiplying a matrix view $\boldsymbol{W} \in \mathbb{R}^{C_{\text{out}} \times C_{\text{in}} K_1 K_2}$ of the kernel onto the unfolded input then yields a matrix view $\boldsymbol{Y}$ of $\mathbf{Y}$ (the vector of ones, $\mathbf{1}_{O_1 O_2}$, copies the bias for each channel),

$$\boldsymbol{Y} = \boldsymbol{W}[\![\mathbf{X}]\!] + \boldsymbol{b}\,\mathbf{1}_{O_1 O_2}^{\top} \in \mathbb{R}^{C_{\text{out}} \times O_1 O_2} \, . \tag{1}$$

We can also view convolution as an affine map of the flattened input $\boldsymbol{x} \in \mathbb{R}^{C_{\text{in}} I_1 I_2}$ into a vector view $\boldsymbol{y}$ of $\mathbf{Y}$ with a Toeplitz-structured matrix $\boldsymbol{A}(\mathbf{W}) \in \mathbb{R}^{C_{\text{out}} O_1 O_2 \times C_{\text{in}} I_1 I_2}$,

$$\boldsymbol{y} = \boldsymbol{A}(\mathbf{W})\boldsymbol{x} + \boldsymbol{b} \otimes \mathbf{1}_{O_1 O_2} \in \mathbb{R}^{C_{\text{out}} O_1 O_2} \, . \tag{2}$$

This perspective is uncommon in code, but used in theoretical works [e.g. 65] as it highlights the similarity between convolutions and dense layers.

## 2.2 Tensor Multiplication

Tensor multiplication unifies outer (Kronecker), element-wise (Hadamard), and inner products and uses the input-output index relation to infer the multiplication type. We start with the binary case, then generalize to more inputs: Consider $\mathbf{A}, \mathbf{B}, \mathbf{C}$ whose index names are described by the index tuples $S_1, S_2, S_3$ where $S_3 \subseteq (S_1 \cup S_2)$ (converting tuples to sets if needed). Any product of $\mathbf{A}$ and $\mathbf{B}$ can be described by the multiplication operator $*_{(S_1, S_2, S_3)}$ with

$$\mathbf{C} = *_{(S_1, S_2, S_3)}(\mathbf{A}, \mathbf{B}) \quad \Leftrightarrow \quad [\mathbf{C}]_{S_3} = \sum_{(S_1 \cup S_2) \setminus S_3} [\mathbf{A}]_{S_1} [\mathbf{B}]_{S_2} \tag{3}$$

summing over indices that are not present in the output. E.g., for two matrices $\boldsymbol{A}, \boldsymbol{B}$, their product is $\boldsymbol{A}\boldsymbol{B} = *_{((i,j),(j,k),(i,k))}(\boldsymbol{A}, \boldsymbol{B})$ (see §H.2), their Hadamard product $\boldsymbol{A} \odot \boldsymbol{B} = *_{((i,j),(i,j),(i,j))}(\boldsymbol{A}, \boldsymbol{B})$, and their Kronecker product $\boldsymbol{A} \otimes \boldsymbol{B} = *_{((i,j),(k,l),((i,k),(j,l)))}(\boldsymbol{A}, \boldsymbol{B})$. Libraries support this functionality via einsum, which takes a string encoding of $S_1, S_2, S_3$, followed by $\mathbf{A}, \mathbf{B}$. It also accepts longer sequences $\mathbf{A}_1, \ldots, \mathbf{A}_N$ with index tuples $S_1, S_2, \ldots, S_N$ and output index tuple $S_{N+1}$,

$$\mathbf{A}_{N+1} = *_{(S_1, \ldots, S_N, S_{N+1})}(\mathbf{A}_1, \ldots, \mathbf{A}_N) \Leftrightarrow [\mathbf{A}_{N+1}]_{S_{N+1}} = \sum_{\left(\bigcup_{n=1}^{N} S_n\right) \setminus S_{N+1}} \left( \prod_{n=1}^{N} [\mathbf{A}_n]_{S_n} \right). \tag{4}$$

---

[1]We prefer $I_1, I_2$ over the more common choice $H, W$ to simplify the generalization to higher dimensions.

Table 1: Contraction expressions of operations related to 2d convolution. They include batching and channel groups, which are standard features in implementations. We describe each operation by a tuple of input tensors and a contraction string that uses the `einops` library's syntax [59] which can express index (un-)grouping. Some quantities are only correct up to a scalar factor which is suppressed for brevity. See §B for visualizations and Table B3 for more operations.

| Operation | Operands | Contraction string (`einops` [59] convention) |
|---|---|---|
| Conv. (no bias) | $\mathbf{X}, \mathbf{\Pi}^{(1)}, \mathbf{\Pi}^{(2)}, \mathbf{W}$ | `"n (g c_in) i1 i2, i1 o1 k1, i2 o2 k2, (g c_out) c_in k1 k2 -> n (g c_out) o1 o2"` |
| Unf. input (im2col) | $\mathbf{X}, \mathbf{\Pi}^{(1)}, \mathbf{\Pi}^{(2)}$ | `"n c_in i1 i2, i1 o1 k1, i2 o2 k2 -> n (c_in k1 k2) (o1 o2)"` |
| Unf. kernel (Toeplitz) | $\mathbf{\Pi}^{(1)}, \mathbf{\Pi}^{(2)}, \mathbf{W}$ | `"i1 o1 k1, i2 o2 k2, c_out c_in k1 k2 -> (c_out o1 o2) (c_in i1 i2)"` |
| Weight VJP | $\mathbf{X}, \mathbf{\Pi}^{(1)}, \mathbf{\Pi}^{(2)}, \mathbf{V^{(Y)}}$ | `"n (g c_in) i1 i2, i1 o1 k1, i2 o2 k2, n (g c_out) o1 o2 -> (g c_out) c_in k1 k2"` |
| Input VJP (tr. conv.) | $\mathbf{W}, \mathbf{\Pi}^{(1)}, \mathbf{\Pi}^{(2)}, \mathbf{V^{(Y)}}$ | `"(g c_out) c_in k1 k2, i1 o1 k1, i2 o2 k2, n (g c_out) o1 o2 -> n (g c_in) i1 i2"` |
| KFC/KFAC-expand | $\mathbf{X}, \mathbf{\Pi}^{(1)}, \mathbf{\Pi}^{(2)}, \mathbf{X}, \mathbf{\Pi}^{(1)}, \mathbf{\Pi}^{(2)}$ | `"n (g c_in) i1 i2, i1 o1 k1, i2 o2 k2, n (g c_in_) i1 i2, i1 o1 k1_, i2 o2 k2_ -> g (c_in k1 k2) (c_in_ k1_ k2_)"` |
| KFAC-reduce | $\mathbf{X}, \mathbf{\Pi}^{(1)}, \mathbf{\Pi}^{(2)}, \mathbf{X}, \mathbf{\Pi}^{(1)}, \mathbf{\Pi}^{(2)}$ | `"n (g c_in) i1 i2, i1 o1 k1, i2 o2 k2, n (g c_in_) i1 i2, i1 o1_ k1_, i2 o2_ k2_ -> g (c k1 k2) (c_ k1_ k2_)"` |

Binary and $N$-ary tensor multiplication are commutative: We can simultaneously permute operands and their index tuples without changing the result,

$$*_{(S_1,S_2,S_3)}(\mathbf{A}, \mathbf{B}) = *_{(S_2,S_1,S_3)}(\mathbf{B}, \mathbf{A}), \quad *_{(.,S_i,.,S_j,.)}(., \mathbf{A}_i, ., \mathbf{A}_j, .) = *_{(.,S_j,.,S_i,.)}(., \mathbf{A}_j, ., \mathbf{A}_i, .)$$

They are also associative, i.e. we can multiply operands in any order. However, the notation becomes involved as it requires additional set arithmetic to detect summable indices (see §H.1 for an example).

## 2.3 Tensor Networks & Convolution

A simpler way to understand tensor multiplications is via diagrams developed by e.g. Penrose [53]. Rank-$K$ tensors are represented by nodes with $K$ legs labelled by the index's name[2]. $\boxed{a}$–$i$ denotes a vector $\mathbf{a}$, $i$–$\boxed{B}$–$j$ a matrix $\mathbf{B}$, and $i$–$\boxed{C}$–$j_k$ a rank-3 tensor $\mathbf{C}$. A Kronecker delta $[\boldsymbol{\delta}]_{i,j} = \delta_{i,j}$ is simply a line, $j$–$\boxed{\delta}$–$i = j$–$\boxed{I}$–$i = j$——$i$. Multiplications are indicated by connections between legs. For inner multiplication, we join the legs of the involved indices, e.g. the matrix multiplication diagram is $i$–$\boxed{AB}$–$k = i$–$\boxed{A}$–$j$–$\boxed{B}$–$k$. Element-wise multiplication is similar, but with a leg sticking out. The Hadamard and Kronecker product diagrams are

$$i\text{--}\boxed{\boldsymbol{A} \odot \boldsymbol{B}}\text{--}j = i\text{--}\begin{bmatrix}\boxed{A}\\\boxed{B}\end{bmatrix}\text{--}j, \qquad (i,k)\text{--}\boxed{\boldsymbol{A} \otimes \boldsymbol{B}}\text{--}(j,l) = (i,k)\blacktriangleleft\begin{matrix}i\text{--}\boxed{A}\text{--}j\\k\text{--}\boxed{B}\text{--}l\end{matrix}\blacktriangleright(j,l). \tag{5}$$

Note that the outer tensor product is a rank-4 tensor and must be reshaped (indicated by black triangles[3]) into a matrix. This syntax allows for extracting and embedding tensors along diagonals; e.g. taking a matrix diagonal, $\boxed{\text{diag}(A)}$–$i = \llcorner\boxed{A}\lrcorner$–$i$, or forming a diagonal matrix, $i$–$\boxed{\text{diag}(a)}$–$i = i\llcorner\boxed{a}\lrcorner i$; and generalizes to larger diagonal blocks (§B). In the following, we stick to the simplest case to avoid the more advanced syntax. However, it shows the expressive power of TNs and is required to support common features of convolutions like channel groups (known as separable convolutions).

**Application to convolution:** We define a binary tensor $\mathbf{P} \in \{0,1\}^{I_1 \times O_1 \times K_1 \times I_2 \times O_2 \times K_2}$ which represents the connectivity pattern between input, output, and kernel. $P_{i_1,o_1,k_1,i_2,o_2,k_2}$ is 1 if input locations $(i_1, i_2)$ overlap with kernel positions $(k_1, k_2)$ when computing output locations $(o_1, o_2)$ and 0 otherwise. The spatial couplings are independent along each dimension, hence $\mathbf{P}$ decomposes into $P_{i_1,o_1,k_1,i_2,o_2,k_2} = \Pi^{(1)}_{i_1,o_1,k_1} \Pi^{(2)}_{i_2,o_2,k_2}$ where the index pattern tensor $\mathbf{\Pi}^{(j)} \in \{0,1\}^{I_j \times O_j \times K_j}$ encodes the connectivity along dimension $j$. With that, one obtains

$$Y_{c_{\text{out}},o_1,o_2} = b_{c_{\text{out}}} + \sum_{c_{\text{in}}=1}^{C_{\text{in}}} \sum_{i_1,i_2=1}^{I_1,I_2} \sum_{k_1,k_2=1}^{K_1,K_2} X_{c_{\text{in}},i_1,i_2} \Pi^{(1)}_{i_1,o_1,k_1} \Pi^{(2)}_{i_2,o_2,k_2} W_{c_{\text{out}},c_{\text{in}},k_1,k_2}$$

Without bias, this translates into the diagram in Figure 2a.

---

[2]We use identical shapes for all tensors. Leg orientation does not assign properties like co-/contra-variance.

[3]Reshape can be seen as multiplication with a one-hot tensor, but we decided to use a separate symbol to emphasize it merely serves for re-interpretation and does not cause much computation.

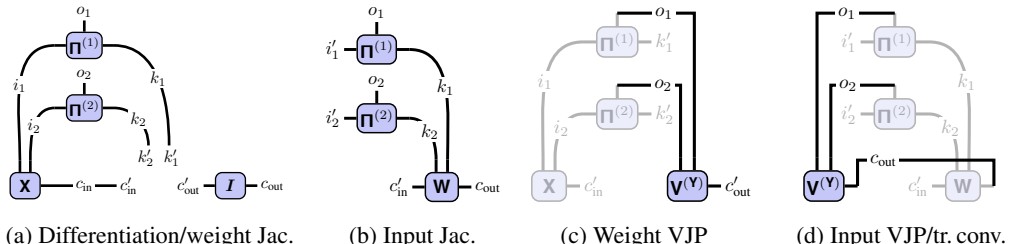

Figure 3: TN differentiation as graphical manipulation. (a) Differentiating convolution w.r.t. **W** is cutting it out of the diagram and yields the weight Jacobian. (b) Same procedure applied to the Jacobian w.r.t. **X**. (c) VJP for the weight and (d) input Jacobian (transpose convolution). Jacobians are shaded, only their contraction with $\mathbf{V}^{(\mathbf{Y})}$ is highlighted.

## 3 TNs for Convolution Operations

We now demonstrate the elegance of TNs for computing derivatives (§3.1), autodiff operations (§3.2), and approximate second-order information (§3.3) by graphical manipulation. For simplicity, we exclude batching (`vmap`-ing like in JAX [8]) and channel groups, and provide the diagrams with full support in §B. Table 1 summarizes our derivations (with batching and groups). As a warm-up, we identify the unfolded input and kernel from the matrix-multiplication view (Equations (1) and (2)). They follow by contracting the index patterns with either the input or kernel (Figures 2b and 2c),

$$[[\mathbf{X}]]_{(c_{\text{in}},k_1,k_2),(o_1,o_1)} = \sum_{i_1,i_2} X_{c_{\text{in}},i_1,i_2} \Pi^{(1)}_{i_1,o_1,k_1} \Pi^{(2)}_{i_2,o_2,k_2} ,$$

$$[\boldsymbol{A}(\mathbf{W})]_{(c_{\text{out}},o_1,o_2),(c_{\text{in}},i_1,i_2)} = \sum_{k_1,k_2} \Pi^{(1)}_{i_1,o_1,k_1} \Pi^{(2)}_{i_2,o_2,k_2} W_{c_{\text{out}},c_{\text{in}},k_1,k_2} .$$

### 3.1 Tensor Network Differentiation

Derivatives play a crucial role in theoretical and practical ML. First, we show that differentiating a TN diagram amounts to a simple graphical manipulation. Then, we derive the Jacobians of convolution. Consider an arbitrary TN represented by the tensor multiplication from Equation (4). The Jacobian tensor $[\mathbf{J}_{\mathbf{A}_j}\mathbf{A}_{N+1}]_{S_{N+1},S'_j} = \partial[\mathbf{A}_{N+1}]_{S_{N+1}}/\partial[\mathbf{A}_j]_{S'_j}$ w.r.t. an input $\mathbf{A}_j$ collects all partial derivatives and is addressed through indices $S_{n+1} \times S'_j$ with $S'_j$ an independent copy of $S_j$. Assume that $\mathbf{A}_j$ only enters once in the tensor multiplication. Then, taking the derivative of Equation (4) w.r.t. $[\mathbf{A}_j]_{S'_j}$ simply replaces the tensor by a Kronecker delta $\delta_{S_j,S'_j}$,

$$\frac{\partial[\mathbf{A}_{N+1}]_{S_{N+1}}}{\partial[\mathbf{A}_j]_{S'_j}} = \sum_{\left(\bigcup_{n=1}^{N}S_n\right)\setminus S_{n+1}} [\mathbf{A}_1]_{S_1} \cdots [\mathbf{A}_{j-1}]_{S_{j-1}} \prod_{i\in S_j} \delta_{i,i'} [\mathbf{A}_{j+1}]_{S_{j+1}} \cdots [\mathbf{A}_N]_{S_N} \quad (6)$$

If an index $i \in S_j$ is summed, $i \notin S_{n+1}$, we can sum the Kronecker delta $\delta_{i,i'}$, effectively replacing all occurrences of $i$ by $i'$. If instead $i$ is part of the output index, $i \in S_{n+1}$, the Kronecker delta remains part of the Jacobian and imposes structure. Figure 3a illustrates this process in diagrams for differentiating a convolution w.r.t. its kernel. Equation (6) amounts to cutting out the argument of differentiation and assigning new indices to the resulting open legs. For the weight Jacobian $\mathbf{J}_{\mathbf{W}}\mathbf{Y}$, this introduces structure: If we re-interpret the two sub-diagrams in Figure 3a as matrices, compare with the Kronecker diagram from Equation (5) and use Figure 2b, we find $[[\mathbf{X}]]^\top \otimes \boldsymbol{I}_{C_{\text{out}}}$ for the Jacobian's matrix view [e.g. 16]. Figure 3b shows the input Jacobian $\mathbf{J}_{\mathbf{X}}\mathbf{Y}$ which is a tensor view of $\boldsymbol{A}(\mathbf{W})$, as expected from the matrix-vector perspective of Equation (2).

Differentiating a TN is more convenient than using matrix calculus [44] as it amounts to a simple graphical manipulation, does not rely on a flattening convention, and therefore preserves the full index structure. The resulting TN can still be translated back to matrix language, if desired. It also simplifies the computation of higher-order derivatives (e.g. $\partial^2\mathbf{Y}/\partial\mathbf{w}\partial\mathbf{x}$), since differentiation yields another TN and can thus be repeated. If a tensor occurs more than once in a TN, the product rule applies and the derivative is a sum of TNs with one occurrence removed.

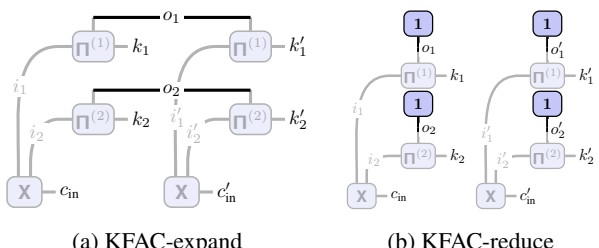

Figure 4: TNs of input-based Kronecker factors for KFAC approximations of the Fisher/GGN (no batching, no groups). The unfolded input is shaded, only additional contractions are highlighted. (a) $\Omega$ (KFC/KFAC-expand) from Grosse & Martens [27], (b) $\hat{\Omega}$ (KFAC-reduce) from Eschenhagen et al. [23] (vectors of ones effectively amount to sums).

(a) KFAC-expand          (b) KFAC-reduce

## 3.2 Autodiff & Connections to Transpose Convolution

Although Jacobians are useful, crucial routines for autodiff are vector-Jacobian and Jacobian-vector products (VJPs, JVPs). Both are simple to realize with TNs due to access to full Jacobians. VJPs are used in backpropagation to pull back a tensor $\mathbf{V}^{(\mathbf{Y})} \in \mathbb{R}^{C_{\text{out}} \times O_1 \times O_2}$ from the output to the input or weight space. The VJP results $\mathbf{V}^{(\mathbf{X})} \in \mathbb{R}^{C_{\text{in}} \times I_1 \times I_2}$ and $\mathbf{V}^{(\mathbf{W})} \in \mathbb{R}^{C_{\text{out}} \times C_{\text{in}} \times K_1 \times K_2}$ are

$$V^{(\mathbf{X})}_{c'_{\text{in}}, i'_1, i'_2} = \sum_{c_{\text{out}}, o_1, o_2} V^{(\mathbf{Y})}_{c_{\text{out}}, o_1, o_2} \frac{\partial Y_{c_{\text{out}}, o_1, o_2}}{\partial X_{c'_{\text{in}}, i'_1, i'_2}}, \qquad V^{(\mathbf{W})}_{c'_{\text{out}}, c'_{\text{in}}, k'_1, k'_2} = \sum_{c_{\text{out}}, o_1, o_2} V^{(\mathbf{Y})}_{c_{\text{out}}, o_1, o_2} \frac{\partial Y_{c_{\text{out}}, o_1, o_2}}{\partial W_{c'_{\text{out}}, c'_{\text{in}}, k'_1, k'_2}}.$$

Both are simply new TNs constructed from contracting the vector with the respective Jacobian, see Figures 3c and 3d (VJPs are analogous). The input VJP is often used to define transpose convolution [21]. In the matrix-multiplication perspective (Equation (2)), this operation is defined relative to a convolution with kernel $\mathbf{W}$ by multiplication with $\mathbf{A}(\mathbf{W})^\top$, i.e. using the same connectivity pattern but mapping from the convolution's output to input space. The TN in Figure 3d makes this sharing explicit and cleanly defines transpose convolution.[4]

## 3.3 Kronecker-factored Approximate Curvature

The Jacobian diagrams allow us to construct the TNs of second-order information like the Fisher/generalized Gauss-Newton (GGN) matrix and sub-tensors like its diagonal (§C). Here, we focus on the popular Kronecker-factored approximation of the GGN [47, 27, 23, 48] whose input-based Kronecker factor relies on the unfolded input $[\![\mathbf{X}]\!]$ which requires large memory. State-of-the-art libraries that provide access to KFAC [17, 51] also use this approach. Using TNs, we can often avoid expanding $[\![\mathbf{X}]\!]$ explicitly and save memory. Here, we describe the existing KFAC approximations and their TNs (see §5.1 for their run time evaluation).

**KFC (KFAC-expand):** Grosse & Martens [27] introduce a Kronecker approximation for the kernel's GGN, $G \approx \Omega \otimes \Gamma$ where $\Gamma \in \mathbb{R}^{C_{\text{out}} \times C_{\text{out}}}$ and the input-based factor $\Omega = [\![\mathbf{X}]\!][\![\mathbf{X}]\!]^\top \in \mathbb{R}^{C_{\text{in}} K_1 K_2 \times C_{\text{in}} K_1 K_2}$ (Figure 4a), the unfolded input's self-inner product (averaged over a batch).

**KFAC-reduce:** Eschenhagen et al. [23] generalized KFAC to graph neural networks and transformers based on the concept of weight sharing, also present in convolutions. They identify two approximations: KFAC-expand and KFAC-reduce. The former corresponds to KFC [27]. The latter shows similar performance in downstream tasks, but is cheaper to compute. It relies on the column-averaged unfolded input, i.e. the average over all patches sharing the same weights. KFAC-reduce approximates $G \approx \hat{\Omega} \otimes \hat{\Gamma}$ with $\hat{\Gamma} \in \mathbb{R}^{C_{\text{out}} \times C_{\text{out}}}$ and $\hat{\Omega} = 1/(O_1 O_2)^2 \mathbf{1}_{O_1 O_2}^\top [\![\mathbf{X}]\!](\mathbf{1}_{O_1 O_2}^\top [\![\mathbf{X}]\!])^\top \in \mathbb{R}^{C_{\text{in}} K_1 K_2 \times C_{\text{in}} K_1 K_2}$ (Figure 4b; averaged over a batch). For convolutions, this is arguably a 'more natural' approximation as it becomes exact in certain limits [23], in contrast to the expand approximation.

**KFAC for transpose convolution:** Our approach enables us to derive KFAC for transpose convolutions. We are not aware of previous works doing so. This seems surprising because, similar to §2.1, transpose convolution can be seen as matrix multiplication between the kernel and an unfolded input. From this formulation we can immediately obtain KFAC through the weight sharing view of Eschenhagen et al. [23]. The Kronecker factor requires unfolding the input similar to im2col, but for transpose convolutions. This operation is currently not provided by ML libraries. We can overcome this limitation, express the unfolding operation as TN, and—for the first time—establish KFAC (expand and reduce) for transpose convolutions (see §B.4 for details).

---

[4]Standalone implementations of transpose convolution require another parameter to unambiguously reconstruct the convolution's input dimension (see §D for how to compute $\Pi$ in this case).

Figure 5: TN illustrations of index pattern simplifications and transformations. See § D.3 for the math formulation.

(a) Kernel-output swap      (b) Dense convolution

(c) Down-sampling convolution

## 4   TN Simplifications & Implementation

Many convolutions in real-world CNNs use structured connectivity patterns that allow for simplifications which we describe here along with implementation aspects.

### 4.1   Index Pattern Structure & Simplifications

The index pattern $\boldsymbol{\Pi}$ encodes the connectivity of a convolution and depends on its hyper-parameters. Along one dimension, $\boldsymbol{\Pi} = \boldsymbol{\Pi}(I, K, S, P, D)$ with input size $I$, kernel size $K$, stride $S$, padding $P$, and dilation $D$. We provide pseudo-code for computing $\boldsymbol{\Pi}$ in §D which is easy to implement efficiently with standard functions from any numerical library (Algorithm D1). Its entries are

$$[\boldsymbol{\Pi}(I, K, S, P, D)]_{i,o,k} = \delta_{i,1+(k-1)D+(o-1)S-P}, \tag{7}$$

with $i = 1, \ldots, I, o = 1, \ldots, O, k = 1, \ldots, K$ and output size $O(I, K, S, P, D) = 1 + \lfloor (I+2P-(K+(K-1)(D-1)))/S \rfloor$. Since $\boldsymbol{\Pi}$ is binary and has size linear in $I, O, K$, it is cheap to pre-compute and cache. The index pattern's symmetries allow for re-wiring a TN. For instance, the symmetry of $(k, D)$ and $(o, S)$ in Equation (7) and $O(I, K, S, P, D)$ permits a *kernel-output swap*, exchanging the role of kernel and output dimension (Figure 5a). Rochette et al. [58] used this to phrase the per-example gradient computation (Figure 3c) as convolution.

For many convolutions of real-world CNNs (see §E for a hyper-parameter study) the index pattern possesses structure that simplifies its contraction with other tensors into either smaller contractions or reshapes: *Dense convolutions* use a shared kernel size and stride, and thus process non-overlapping adjacent tiles of the input. Their index pattern's action can be expressed as a cheap reshape (Figure 5b). Such convolutions are common in DenseNets [33], MobileNets [31, 60], ResNets [30], and ConvNeXts [42]. InceptionV3 [69] has 2d *mixed-dense convolutions* that are dense along one dimension. *Down-sampling convolutions* use a larger stride than kernel size, hence only process a sub-set of their input, and are used in ResNet18 [30], ResNext101 [72], and WideResNet101 [73]. Their pattern contracts with a tensor $\mathbf{V}$ like that of a dense convolution with a sub-tensor $\tilde{\mathbf{V}}$ (Figure 5c). §5.1 shows that those simplifications accelerate computations.

### 4.2   Practical Benefits of the TN Abstraction & Limitations for Convolutions

**Contraction order optimization:** There exist various orders in which to carry out the summations in a TN and their performance can vary by orders of magnitude. One extreme approach is to carry out all summations via nested for-loops. This so-called Feynman path integral algorithm requires little memory, but many FLOPS since it does not re-cycle intermediate results. The other extreme is sequential pair-wise contraction. This builds up intermediate results and can greatly reduce FLOPS. The schedule is represented by a binary tree, but the underlying search is in general at least #P-hard [14]. Fortunately, there exist heuristics to find high-quality contraction trees for TNs with hundreds of tensors [32, 25, 13], implemented in packages like `opt_einsum` [66].

**Index slicing:** A common problem with high-quality schedules is that intermediates exceed memory. Dynamic slicing [32] (e.g. `cotengra` [25]) is a simple method to decompose a contraction until it becomes feasible by breaking it up into smaller identical sub-tasks whose aggregation adds a small overhead. This enables peak memory reduction and distribution.

**Sparsity:** $\boldsymbol{\Pi}$ is sparse as only a small fraction of the input contributes to an output element. For a convolution with stride $S < K$ and default parameters ($P = 0, D = 1$), for fixed output and kernel indices $k, o$, there is exactly one non-zero entry in $[\boldsymbol{\Pi}]_{:,o,k}$. Hence $\mathtt{nnz}(\boldsymbol{\Pi}) = OK$, which corresponds to a sparsity of $1/I$. Padding leads to more kernel elements that do not contribute to an

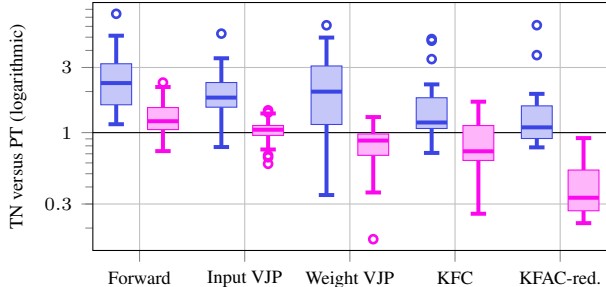

Figure 6: Run time ratios of TN (w/o simplifications) vs. standard implementation for dense convolutions of 9 CNNs. With simplifications, convolution and input VJP achieve median ratios slightly above 1, and the TN implementation is faster for weight VJP, KFC & KFAC-reduce. The code in Figure 1 corresponds to default, TN, and simplified TN KFC implementation.

output pixel, and therefore a sparser $\Pi$. For down-sampling and dense convolutions, we showed how $\Pi$'s algebraic structure allows to simplify its contraction. However, if that is not possible, $\Pi$ contains explicit zeros that add unnecessary FLOPS. One way to circumvent this is to match a TN with that of an operation with efficient implementation (like im2col, (transpose) convolution) using transformations like the *kernel-output swap* or by introducing identity tensors to complete a template, as done in Rochette et al. [58], Dangel [15] for per-sample gradients and im2col.

**Approximate contraction & structured dropout:** TNs offer a principled approach for stochastic approximation via Monte-Carlo estimation to save memory and run time at the cost of accuracy. The basic idea is best explained on a matrix product $C := AB = \sum_{n=1}^{N} [A]_{:,n} [B]_{n,:}$ with $A \in \mathbb{R}^{I \times N}, B \in \mathbb{R}^{N,O}$. To approximate the sum, we introduce a distribution over $n$'s range, then use column-row-sampling [CRS, 1] to form an unbiased Monte-Carlo approximation with sampled indices, which only requires the sub-matrices with active column-row pairs. Bernoulli-CRS samples without replacement by assigning a Bernoulli random variable $\text{Bernoulli}(\pi_n)$ with probability $\pi_n$ for column-row pair $n$ to be included in the contraction. The Bernoulli estimator is $\tilde{C} := \sum_{n=1}^{N} z_n/\pi_n [A]_{n,:} [B]_{n,:}$ with $z_n \sim \text{Bernoulli}(\pi_n)$. With a shared keep probability, $\pi_n := p$, this yields the unbiased estimator $C' = 1/p \sum_{n=1,...,N} A'B'$ where $A' = AK$ and $B' = KB$ with $K = \text{diag}(z_1, \ldots, z_N)$ are the sub-matrices of $A, B$ containing the active column-row pairs. CRS applies to a single contraction. For TNs with multiple sums, we can apply it individually, and also impose a distribution over the result indices, which computes a (scaled) sub-tensor.

## 5 Experiments

Here, we demonstrate computational benefits of TNs for less standard routines of second-order methods and showcase their flexibility to perform stochastic autodiff in novel ways.

### 5.1 Run Time Evaluation

We implement the presented TNs' contraction strings and operands[5] in PyTorch [52]. The simplifications from §4 can be applied on top and yield a modified `einsum` expression. To find a contraction schedule, we use `opt_einsum` [66] with default settings. We extract the unique convolutions of 9 architectures for ImageNet and smaller data sets, then compare some operations from Table 1 with their standard implementation on an Nvidia Tesla T4 GPU (16 GB); see §F for all details. Due to space constraints, we highlight important insights here and provide references to the corresponding material in the appendix. In general, the performance gap between standard and TN implementation decreases the less common an operation is (Figure F17); from forward pass (inference & training), to VJPs (training), to KFAC (training with a second-order method). This is intuitive as more frequently used routines have been optimized more aggressively.

**Impact of simplifications:** While general convolutions remain unaffected (Figure F18d) when applying the transformations of §4, mixed dense, dense, and down-sampling convolutions consistently enjoy significant run time improvements (Figures F18a to F18c). As an example, we show the performance comparison for dense convolutions in Figure 6: The performance ratio's median between TN and standard forward and input VJP is close to 1, that is both require almost the same time. In the median, the TN even outperforms PyTorch's highly optimized weight VJP, also for down-sampling

---

[5]`einsum` does not yet support index un-grouping, so we must reshape manually before and after.

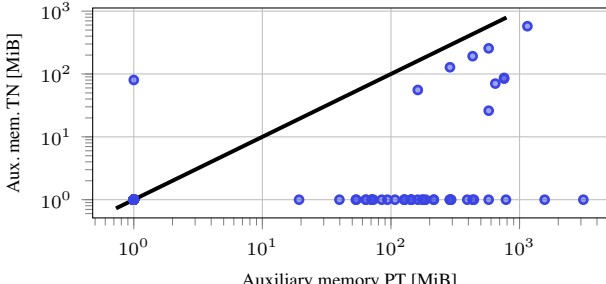

Figure 7: Extra memory used by the standard versus our TN implementation (simplifications enabled) of KFAC-reduce. Each point represents a convolution from 9 CNNs, clipped below by 1 MiB. TNs consistently use less memory than the standard implementation (one exception), and often no extra memory at all. We observe memory savings up to 3 GiB.

convolutions (Figure F21). For KFC, the median performance ratios are well below 1 for dense, mixed dense & sub-sampling convolutions (Figure F22).

**KFAC-reduce:** For all convolution types, the TN implementation achieves its largest improvements for $\hat{\Omega}$ and consistently outperforms the PyTorch implementation in the median when simplifications are enabled (Figure F23). The standard implementation unfolds the input, takes the row-average, then forms its outer product. The TN does not need to expand $[\![\mathbf{X}]\!]$ in memory and instead averages the index pattern tensors, which reduces peak memory and run time. We observe performance ratios down to $0.22\,\mathrm{x}$ (speed-ups up to $\approx 4.5\,\mathrm{x}$, Table F9) and consistently lower memory consumption with savings up to 3 GiB (Figure 7). Hence, our approach not only significantly reduces the overhead of 2nd-order optimizers based on KFAC-reduce, but also allows them to operate on larger batches without exceeding memory (Eschenhagen et al. [23] specifically mention memory as important limitation of their method). Other examples for KFAC algorithms where computing the input-based Kronecker factor adds significant time and memory overhead are that of Petersen et al. [54], Benzing [5] which only use $\Omega$ (setting $\Gamma \propto \boldsymbol{I}$), or Lin et al. [41, 40] which remove matrix inversion.

**Downstream improvements with KFAC-reduce:** To demonstrate the speed-ups of KFAC-reduce in practical algorithms, we apply our work to the SINGD optimizer [41] and benchmark the impact of our TN implementation on its memory and run time in comparison to SGD without momentum. Concretely, we investigate SINGD with KFAC-reduce and diagonal pre-conditioners on ResNet18 and VGG19 on ImageNet-like synthetic data $(3, 256, 256)$ using a batch size of 128. We measured per-iteration time and peak memory on an NVIDIA A40 with 48 GiB of RAM. For SINGD, we compare computing the Kronecker factors with the standard approach ('SINGD') via input unfolding versus our TN implementation ('SINGD+TN'). Table 2 summarizes the results.

On both nets, our TN implementation halves SINGD's run time, and almost completely eliminates the memory, overhead compared to SGD. On VGG19, it dramatically lowers the memory overhead, cutting it down by a factor of 2 from 32 GiB to 16 GiB. This enables using larger batches or more frequently updating the pre-conditioner, underlining the utility of our approach for reducing the computational gap between approximate second-order and first-order methods.

Table 2: Impact of our TN implementation on SINGD's run time and peak memory compared to SGD.

| | Optimizer | Per iter. [s] | Peak mem. [GiB] |
|---|---|---|---|
| ResNet18 | SGD | 0.12 (1.0 x) | 3.6 (1.0 x) |
| | SINGD | 0.19 (1.7 x) | 4.5 (1.3 x) |
| | SINGD+TN | 0.16 (1.3 x) | 3.6 (1.0 x) |
| VGG19 | SGD | 0.69 (1.0 x) | 14 (1.0 x) |
| | SINGD | 1.0 (1.5 x) | 32 (2.3 x) |
| | SINGD+TN | 0.80 (1.2 x) | 16 (1.1 x) |

## 5.2 Randomized Autodiff via Approximate Contraction

CRS is an alternative to checkpointing [26] to lower memory consumption of backpropagation [50, 11, 1]. Here, we focus on unbiased gradient approximations by applying the exact forward pass, but CRS when computing the weight VJP, which requires storing a sub-tensor of $\mathbf{X}$. For convolutions, the approaches of existing works are limited by the supported functionality of ML libraries. Adelman et al. [1] restrict to sampling $\mathbf{X}$ along $c_{\mathrm{in}}$, which eliminates many gradient entries as the index is part of the gradient. The randomized gradient would thus only train a sub-tensor of the kernel per step. Oktay et al. [50], Chen et al. [11] apply unstructured dropout to $\mathbf{X}$, store it in sparse form, and restore the sparsified tensor during the backward pass. This reduces memory, but not computation.

Our TN implementation is more flexible and can, for example, tackle spatial dimensions with CRS. This reduces memory to the same extent, but also run time due to fewer contractions. Importantly, it

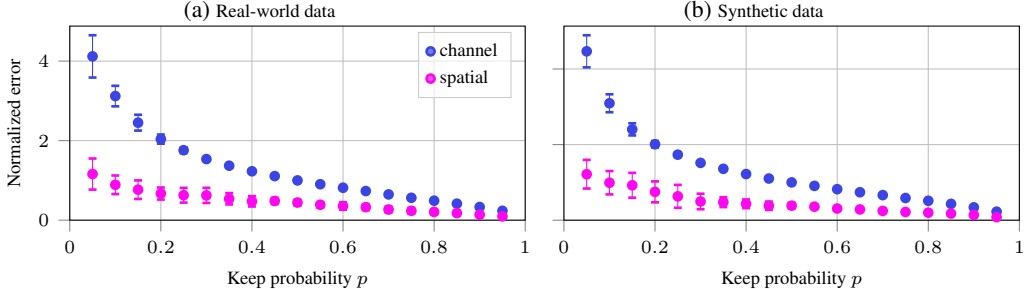

Figure 8: Sampling spatial axes is more effective than channels on both (a) real-world and (b) synthetic data. We take the untrained All-CNN-C [68] for CIFAR-100 with cross-entropy loss, disable dropout, and modify the convolutions to use a fraction $p$ of **X** when computing the weight gradient via Bernoulli-CRS. For mini-batches of size 128, we compute the deterministic gradients for all kernels, then flatten and concatenate them into a vector $\boldsymbol{g}$; likewise for its proxy $\hat{\boldsymbol{g}}$. CRS is described by $(p_{c_{\text{in}}}, p_{i_1}, p_{i_2})$, the keep rates along the channel and spatial dimensions. We compare channel and spatial sampling with same memory reduction, i.e. $(p, 1, 1)$ and $(1, \sqrt{p}, \sqrt{p})$. To measure approximation quality, we use the normalized residual norm $\|\boldsymbol{g}-\hat{\boldsymbol{g}}\|_2/\|\boldsymbol{g}\|_2$ and report mean and standard deviation of 10 different model and batch initializations.

does not zero out the gradient for entire filters. In Figure 8 we compare the gradient approximation errors of channel and spatial sub-sampling. For the same memory reduction, spatial sub-sampling yields a smaller approximation error on both real & synthetic data. E.g., instead of keeping 75 % of channels, we achieve the same approximation quality using only 35 % of pixels.

# 6 Related Work

**Structured convolutions:** We use the TN formulation of convolution from Hayashi et al. [29] who focus on connecting kernel factorizations to existing (depth-wise separable [31, 60], factored [69], bottleneck [30], flattened/CP decomposed, low-rank filter [67, 57, 70]) convolutions and explore new factorizations. Our work focuses on operations related to convolutions, diagram manipulations, the index pattern structure, and computational performance/flexibility. Structured convolutions integrate seamlessly with our framework by replacing the kernel with its factorized TN.

**Higher-order autodiff:** ML frameworks focus on differentiating scalar-valued objectives once. Recent works [37, 38, 43] developed a tensor calculus to compute higher-order derivatives of tensor-valued functions and compiler optimizations through linear algebra and common sub-expression elimination. Phrasing convolution as `einsum`, we allow it to be integrated into such frameworks, benefit from their optimizations, and complement them with our convolution-specific simplifications.

# 7 Conclusion

We used tensor networks (TNs), a diagrammatic representation of tensor multiplications, to simplify convolutions and many related operations. We derived the diagrams of autodiff and less standard routines for curvature approximations like KFAC with support for all hyper-parameters, channel groups, batching, and generalization to arbitrary dimensions. All amount to simple `einsum` expressions that can easily be modified—e.g. to perform stochastic backpropagation—and benefit from under-the-hood optimizations before evaluation. We complemented those by convolution-specific symbolic simplifications based on structure in the connectivity pattern and showed their effectiveness to advance second-order methods. Our TN implementation accelerates the computation of KFAC up to 4.5 x and uses significantly less memory. Beyond performance improvements, the simplifying perspective also allowed us to formulate KFAC for transpose convolution. More broadly, our work underlines the elegance of TNs for reasoning about tensor multiplications and function transformations (differentiation, batching, slicing, simplification) in terms of diagrams at less cognitive load without sacrificing rigour. We believe they are a powerful tool for the ML community that will open up new algorithmic possibilities due to their simplicity & flexibility.

## Acknowledgments and Disclosure of Funding

The author would like to thank Luca Thiede, Andres Fernandez Rodríguez, and Kirill Neklyudov for providing feedback to the manuscript. Resources used in preparing this research were provided, in part, by the Province of Ontario, the Government of Canada through CIFAR, and companies sponsoring the Vector Institute.

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

# Convolutions and More as Einsum: A Tensor Network Perspective with Advances for Second-Order Methods (Supplementary Material)

# A   Limitations

Here we comment on limitations on our approach.

**No common sub-expression elimination (CSE):**   Our implementation relies on `opt_einsum` which focuses on contraction order optimization. This optimization is efficient when all operands are different. However, with multiple occurrences of operands, computing shared sub-expressions might be an advantageous optimization approach which `opt_einsum` does not account for. The second-order quantity TNs from §C and §3.3 contain such sub-expressions, for instance $[\![\mathbf{X}]\!]$ and $\mathbf{1}_{O_1 O_2}^\top [\![\mathbf{X}]\!]$ in KFAC-expand and KFAC-reduce, and $\mathbf{S}^{(\mathbf{W})}$ in the GGN quantities from Figure C16. The efficiency of CSE depends on how costly the shared tensor is to compute. For instance, computing $\mathbf{S}^{(\mathbf{W})}$ is expensive and therefore CSE is the more suitable optimization technique. For the input-based Kronecker factors which require the unfolded input, either contraction path optimization or CSE might be better. This is because the optimal contraction order may not correspond to 2x input unfolding and exhibit more parallelism which may lead to faster run times on a GPU. It would be interesting to integrate CSE into the contraction path optimization and develop a heuristic to choose a contraction path, for instance based on a weighted sum of FLOPs and memory.

**No index slicing:**   We mention index slicing as a technique to reduce peak memory of, and distribute, TN contractions. However, our implementation does not use index slicing, although there are packages like `cotengra` [25] with an interface similar to `opt_einsum`. We did not experiment with index slicing as our benchmark uses a single GPU and did not encounter out-of-memory errors. Still, we mention this technique, as, in combination with CSE, it could automatically reduce peak memory of the GGN quantities from Figure C16 which suffer from high memory requirements.

# B   Visual Tour of Tensor Network Operations for Convolutions

Here, we extend the presented operations with a batch axis and allow for grouped convolutions.

## B.1   Convolution & First-order Derivatives

**Adding a batch dimension (`vmap`-ing):**   Adding a batch axis to all presented operations is trivial. We only need to add an additional leg to the batched tensors, and connect these legs via element-wise or inner multiplication, depending on whether the result tensor is batched or not.

**Grouped convolutions:**   Grouped convolutions were originally proposed by Krizhevsky et al. [36] and allow for parallelizing, distributing, and reducing the parameters of the convolution operation. They split $C_{\text{in}}$ input channels into $G$ groups of size $\tilde{C}_{\text{in}} := C_{\text{in}}/G$, then perform independent convolutions per group, each producing $\tilde{C}_{\text{out}} := C_{\text{out}}/G$ output channels which are concatenated in the output. Each group uses a kernel $\mathbf{W}_g$ of size $\tilde{C}_{\text{out}} \times \tilde{C}_{\text{in}} \times K_1 \times K_2$. These kernels are stacked into a single tensor $\mathbf{W} \in \mathbb{R}^{C_{\text{out}}, \tilde{C}_{\text{in}}, K_1, K_2}$ such that $[\mathbf{W}]_{(g,:),:,:,:} = \mathbf{W}_g$. To support groups, we thus decompose the channel indices into $c_{\text{in}} := (\tilde{c}_{\text{in}}, g)$ and $c_{\text{out}} := (\tilde{c}_{\text{out}}, g)$. For the forward pass this yields the grouped convolution (without bias)

$$Y_{(g,\tilde{c}_{\text{out}}),o_1,o_2} = \sum_{i_1,i_2,\tilde{c}_{\text{in}},k_1,k_2} X_{(g,\tilde{c}_{\text{in}}),i_1,i_2} \Pi^{(1)}_{i_1,o_1,k_1} \Pi^{(2)}_{i_2,o_2,k_2} W_{(g,\tilde{c}_{\text{out}}),c_{\text{in}},k_1,k_2} . \tag{B8}$$

Figure B9a shows the batched version of Equation (B8) as TN. Applying the differentiation rule from §3 leads to the Jacobians and VJPs shown in the remaining panels of Figure B9.

## B.2   Exact Second-order Information

In Figure B12 we show the TNs for the GGN diagonal and the GGN Gram matrix (empirical NTK matrix) from Figure C16 extended by channel groups and a batch axis.

**Diagonal block extraction:**   Combined with index un-grouping, diagonal extraction generalizes to larger blocks: Let $\boldsymbol{A} \in \mathbb{R}^{KI \times KJ}$ be a matrix of $K$ horizontally and vertically concatenated blocks

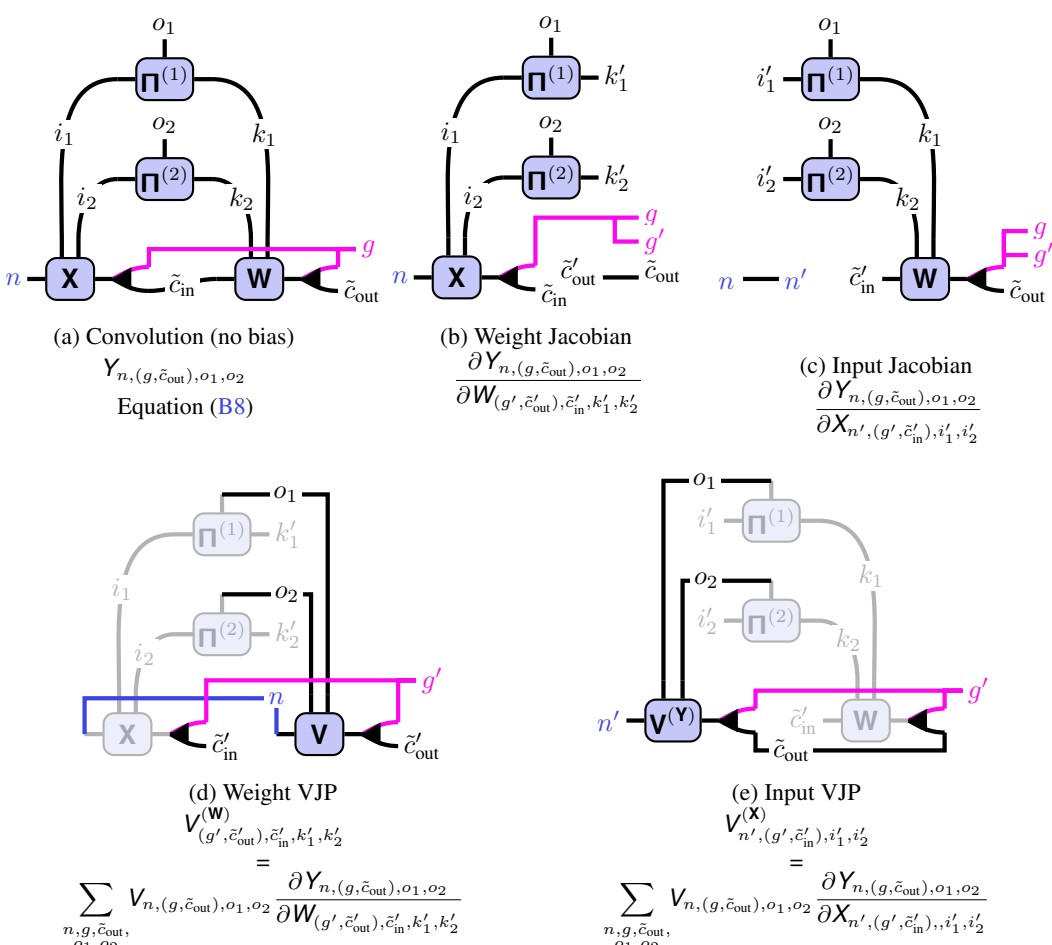

Figure B9: TNs of the (a) forward pass, (b, c) Jacobians, and (d, e) VJPs with batch axis and channel groups. They generalize Figures 2 and 3 from the main text. For the VJPs, the Jacobians are shaded.

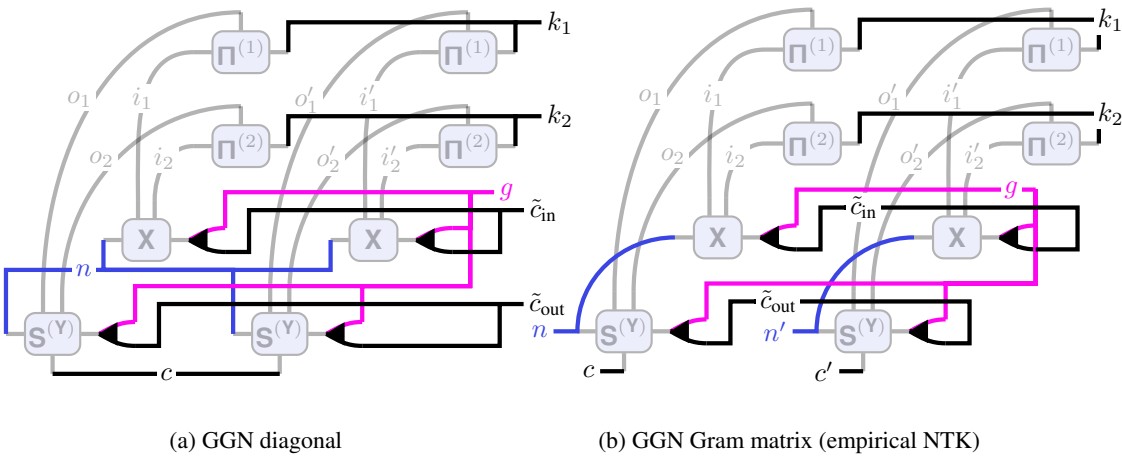

(a) GGN diagonal

(b) GGN Gram matrix (empirical NTK)

Figure B10: TNs of (a) the GGN diagonal and (b) the GGN Gram matrix with batching and channel groups. They extend Figures C16b and C16c from the main text.

$\boldsymbol{A}^{(k_1,k_2)} \in \mathbb{R}^{I \times J}, k_i = 1 \ldots, K$. We can extract the diagonal blocks by restoring the sub-structure,

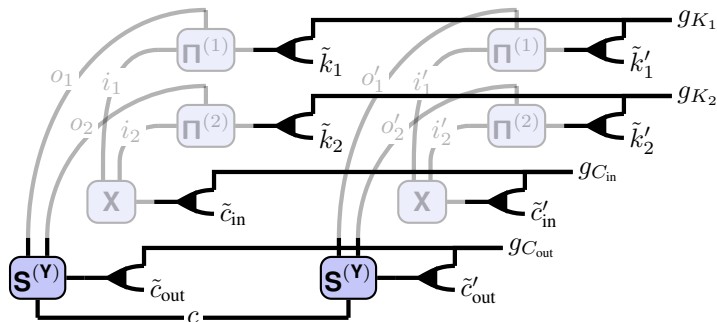

Figure B11: TN of a GGN mini-block diagonal without batching and channel groups.

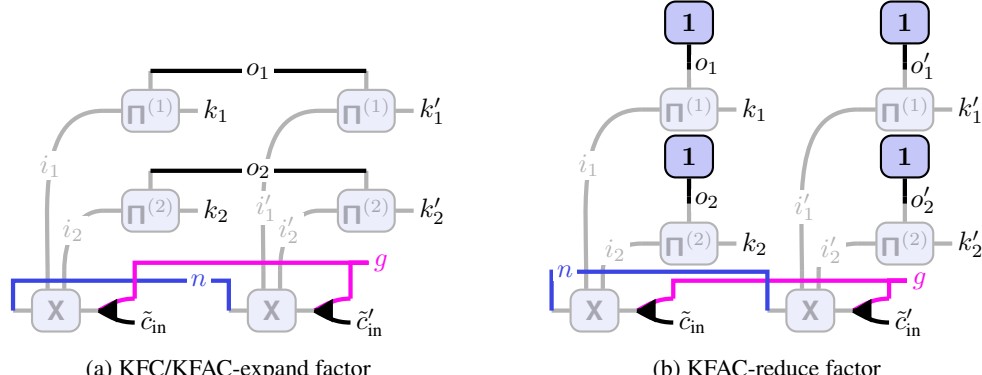

(a) KFC/KFAC-expand factor

(b) KFAC-reduce factor

Figure B12: TN diagrams of input-based factors in Kronecker approximations of the GGN for convolutions with batching and channel groups. They extend Figure 4 from the main text.

then taking the diagonal along the $K$-dimensional index,

$$k \underset{i}{=} \{ A^{(k,k)} \} - j \; = \; \underset{i}{\overset{k}{\searrow}} (k,i) - A - (k,j) \overset{}{\underset{j}{\nwarrow}} .$$

We can apply this procedure to the GGN from Figure C16a. Assume we want to divide the output channel, input channel, and spatial dimensions into $G_{C_{\text{out}}}, G_{C_{\text{in}}}, G_{K_1}, G_{K_2}$ groups. A group will thus be indexed with a tuple $(g_{C_{\text{out}}}, g_{C_{\text{in}}}, g_{K_1}, g_{K_2})$ and the corresponding GGN block will be of dimension $C_{\text{out}}/G_{C_{\text{out}}} \times C_{\text{in}}/G_{C_{\text{in}}} \times K_1/G_{K_1} \times K_2/G_{K_2} \times C_{\text{out}}/G_{C_{\text{out}}} \times C_{\text{in}}/G_{C_{\text{in}}} \times K_1/G_{K_1} \times K_2/G_{K_2}$ and correspond to the GGN for $[\mathbf{W}]_{(g_{C_{\text{out}}},:),(g_{C_{\text{in}}},:),(g_{K_1},:),(g_{K_2},:)}$. This process of un-grouping the output dimensions, then taking the diagonal along the group indices, is illustrated in Figure B11. Note that if we choose $G_{C_{\text{out}}} = C_{\text{out}}, G_{C_{\text{in}}} = C_{\text{in}}, G_{K_1} = K_1, G_{K_2} = K_2$, each block will be a single number and hence we recover the GGN diagonal from Figure C16b. If instead we $G_{C_{\text{out}}} = G_{C_{\text{in}}} G_{K_1} G_{K_2} = 1$, we obtain the full GGN from Figure C16a. The outlined schemes allows to extract mini-blocks of arbitrary size along the diagonal (subject to the total dimension).

## B.3 Kronecker-factored Approximate Curvature (KFAC) for Grouped Convolutions

We were unable to find a definition of KFAC for grouped convolutions. Hence, we derive it here and present the TN diagrams. We use the perspective that grouped convolutions are independent convolutions over channel groups which are then concatenated. For each of those convolutions, we can then apply established the KFAC approximation for convolutions without groups. For a group $g$ we have the kernel $\mathbf{W}_g = [\mathbf{W}]_{(g,:),:,:,:}$ and the unfolded input of its associated input channels, $[\![\mathbf{X}_g]\!] = [\![\mathbf{X}]\!]_{(g,:),:,:} = [\![\mathbf{X}]\!]_{(g,:),:,:,:}$ (or $[\![\mathbf{X}_{n,g}]\!] = [\![\mathbf{X}_n]\!]_{(g,:),:,:} = [\![\mathbf{X}]\!]_{n,(g,:),:,:}$ in the batched setting).

**KFC/KFAC-expand for grouped convolutions:** Applying the regular KFC approximation to the kernel of group $g$, this yields the Fisher approximation $\mathbf{\Omega}_g \otimes \mathbf{\Gamma}_g$ with $\mathbf{\Gamma}_g \in \mathbb{R}^{\tilde{C}_{\text{out}} \times \tilde{C}_{\text{out}}}$ and

$\mathbf{\Omega}_g = 1/N \sum_{n=1}^N [\![\mathbf{X}_{n,g}]\!][\![\mathbf{X}_{n,g}]\!]^\top \in \mathbb{R}^{\tilde{C}_{\text{in}} K_1 K_2 \times \tilde{C}_{\text{in}} K_1 K_2}$ where $\mathbf{X}_{n,g}$ is the input tensor for sample $n$ and group $g$ (remember the index structure $\mathbf{X}_{n,(g,\tilde{c}_{\text{in}}),i_1,i_2}$). Figure B12a shows the diagram for $\{N\mathbf{\Omega}_g\}_{g=1}^G$.

**KFAC-reduce for grouped convolutions:**    Proceeding in the same way, but using the unfolded input averaged over output locations, we obtain the Fisher approximation $\hat{\mathbf{\Omega}}_g \otimes \hat{\mathbf{\Gamma}}_g$ with $\hat{\mathbf{\Gamma}}_g \in \mathbb{R}^{\tilde{C}_{\text{out}} \times \tilde{C}_{\text{out}}}$ and $\hat{\mathbf{\Omega}}_g = 1/N(O_1 O_2)^2 \sum_{n=1}^N \mathbf{1}_{O_1 O_2}^\top [\![\mathbf{X}_{n,g}]\!](\mathbf{1}_{O_1 O_2}^\top [\![\mathbf{X}_{n,g}]\!])^\top \in \mathbb{R}^{\tilde{C}_{\text{in}} K_1 K_2 \times \tilde{C}_{\text{in}} K_1 K_2}$ for the kernel of group $g$. Figure B12b shows the diagram for $\{N(O_1 O_2)^2 \hat{\mathbf{\Omega}}_g\}_{g=1}^G$.

## B.4 Kronecker-factored Approximate Curvature (KFAC) for Transpose Convolution

Here we derive the KFAC approximation for transpose convolutions.

We describe transpose convolution in terms of its associated convolution from an input space $\mathcal{X} = \mathbb{R}^{C_{\text{in}} \times I_1 \times I_2}$ to an output space $\mathcal{Y} = \mathbb{R}^{C_{\text{out}} \times O_1 \times O_2}$. The convolution has hyper-parameters $K_{1,2}, S_{1,2}, P_{1,2}, D_{1,2}$ with index patterns $\mathbf{\Pi}^{(1)} = \mathbf{\Pi}(I_1, K_1, S_1, P_1, D_1) \in \mathbb{R}^{I_1 \times O_1 \times K_1}$ and $\mathbf{\Pi}^{(2)} = \mathbf{\Pi}(I_2, K_2, S_2, P_2, D_2) \in \mathbb{R}^{I_2 \times O_2 \times K_2}$.

**Transpose convolution as matrix multiplication:**    Transpose convolution maps a $\mathbf{Y} \in \mathcal{Y}$ into an $\mathbf{X} \in \mathcal{X}$. In ML frameworks like PyTorch, its kernel $\tilde{\mathbf{W}}$ is stored as $C_{\text{out}} \times C_{\text{in}} \times K_1 \times K_2$ tensor. The relation $\mathbf{X} = \tilde{\mathbf{W}} \star_T \mathbf{Y}$ where $\star_T$ denotes transpose convolution is given by Figure 3d,

$$X_{c_{\text{in}}, i_1, i_2} = \sum_{c_{\text{out}}=1}^{C_{\text{out}}} \sum_{k_1=1}^{K_1} \sum_{k_2=1}^{K_2} \sum_{o_1=1}^{O_1} \sum_{o_2=1}^{O_2} \Pi_{i_1,o_1,k_1}^{(1)} \Pi_{i_2,o_2,k_2}^{(2)} Y_{c_{\text{out}},k_1,k_2} \tilde{W}_{c_{\text{out}},c_{\text{in}},k_1,k_2} \tag{B9}$$

Our goal is to turn the express the above as matrix multiplication. To do that, we first define the matrix reshape $X$ of $\mathbf{X}$ via $X \in \mathbb{R}^{C_{\text{in}} \times I_1 I_2}$ such that $[X]_{c_{\text{in}},(i_1,i_2)} = X_{c_{\text{in}},i_1,i_2}$. Next, we consider a transposed kernel $\mathbf{W}$ of $\tilde{\mathbf{W}}$ with changed order of the first two indices, i.e. $\mathbf{W} \in \mathbb{R}^{C_{\text{in}} \times C_{\text{out}} \times K_1 \times K_2}$ such that

$$W_{c_{\text{in}},c_{\text{out}},k_1,k_2} = \tilde{W}_{c_{\text{out}},c_{\text{in}},k_1,k_2} . \tag{B10}$$

This transposition is necessary to convert the kernel's layout in the ML framework to a layout that admits Equation (B9) to be expressed as matrix multiplication. Using a matrix reshape $W$ of $\mathbf{W}$ via $W \in \mathbb{R}^{C_{\text{in}} \times C_{\text{out}} K_1 K_2}$ such that $[W]_{c_{\text{in}},(c_{\text{out}},k_1,k_2)} = W_{c_{\text{in}},c_{\text{out}},k_1,k_2}$, we can express Equation (B9) as matrix multiplication

$$X = W[\![\mathbf{Y}]\!]_T \tag{B11}$$

where $[\![\mathbf{Y}]\!]_T \in \mathbb{R}^{C_{\text{out}} K_1 K_2 \times I_1 I_2}$ is the *transpose-unfolded input* to the transpose convolution (note that $[\![\cdot]\!] \neq [\![\cdot]\!]_T$!)

$$[[\![\mathbf{Y}]\!]_T]_{(c_{\text{out}},k_1,k_2),(i_1,i_2)} = \sum_{o_1=1}^{O_1} \sum_{o_2=1}^{O_2} \Pi_{i_1,o_1,k_1}^{(1)} \Pi_{i_2,o_2,k_2}^{(2)} Y_{c_{\text{out}},o_1,o_2} . \tag{B12}$$

To the best of our knowledge there is no API for $[\![\cdot]\!]_T$ in existing ML frameworks. Our approach can provide a simple and efficient implementation of $[\![\cdot]\!]$ through the TN shown in Figure B13a which corresponds to Equation (B12). As Equation (B11) is of the same form as Equation (1), it is now straightforward to write down the KFAC approximations for transpose convolution.

**KFAC-expand:**    We will define the KFAC-expand approximation for the GGN w.r.t. the flattened kernel $w$ of $W$. Note that, in practise, this approximation must be properly transformed back to the layout $\tilde{\mathbf{W}}$ of the ML framework. We have $G(w) \approx \mathbf{\Omega} \otimes \mathbf{\Gamma}$, with $\mathbf{\Gamma} \in \mathbb{R}^{C_{\text{in}} \times C_{\text{in}}}$ computed from backpropagated gradients, and the input-based Kronecker factor

$$\mathbf{\Omega} = [\![\mathbf{Y}]\!]_T [\![\mathbf{Y}]\!]_T^\top \in \mathbb{R}^{C_{\text{out}} K_1 K_2 \times C_{\text{out}} K_1 K_2} . \tag{B13}$$

See Figure B13b for the corresponding TN.

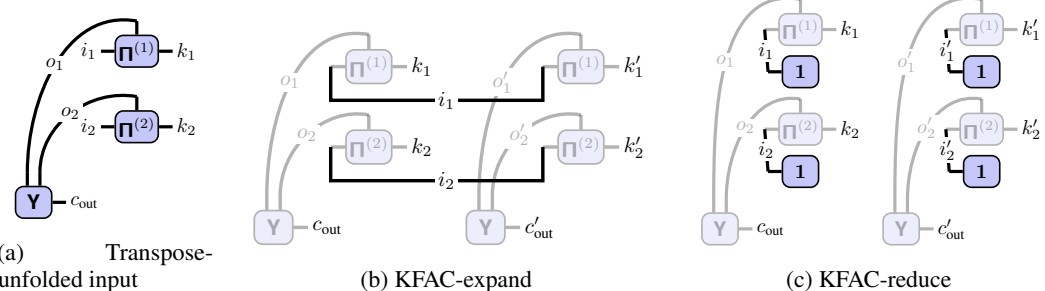

(a) Transpose-unfolded input          (b) KFAC-expand          (c) KFAC-reduce

Figure B13: TNs for extending KFAC to transpose convolutions (no batching and groups).

**KFAC-reduce:** For KFAC-reduce, we have $G(w) \approx \hat{\Omega} \otimes \hat{\Gamma}$, with $\hat{\Gamma} \in \mathbb{R}^{C_{\text{in}} \times C_{\text{in}}}$ computed from backpropagated gradients, and the input-based Kronecker factor

$$\hat{\Omega} = \frac{1}{(I_1 I_2)^2} \left( \mathbf{1}_{I_1 I_2}^\top [\![\mathbf{Y}]\!]_{\text{T}} \right) \left( \mathbf{1}_{I_1 I_2}^\top [\![\mathbf{Y}]\!]_{\text{T}} \right)^\top \in \mathbb{R}^{C_{\text{out}} K_1 K_2 \times C_{\text{out}} K_1 K_2} . \tag{B14}$$

See Figure B13c for the corresponding TN.

**With batching and groups:** In the presence of $G$ groups, we have per-group kernels $\tilde{\mathbf{W}}_g = [\tilde{\mathbf{W}}]_{(g,:),:,:,:} \in \mathbb{R}^{C_{\text{out}}/G \times C_{\text{in}}/G \times K_1 \times K_2}$ and $\mathbf{W}_g \in \mathbb{R}^{C_{\text{in}}/G \times C_{\text{out}}/G \times K_1 \times K_2}$, as well as per-group transpose-unfolded inputs $[\![\mathbf{Y}_g]\!]_{\text{T}} = [\![\mathbf{Y}]\!]_{\text{T}(g,:),:,:} = [\![\mathbf{Y}]_{(g,:),:,:}]\!]_{\text{T}} \in \mathbb{R}^{C_{\text{out}}/G K_1 K_2 \times I_1 I_2}$. Each group corresponds to a transpose convolution in itself. With batching, we have an additional leading batch dimension, i.e. $[\![\mathbf{Y}_{n,g}]\!]_{\text{T}}$. Applying the same steps from above, we can define the KFAC approximation for the GGN w.r.t. the flattened per-group kernel $w_g$ of $W_g$.

For KFAC-expand, we have $G(w_g) \approx \Omega_g \otimes \Gamma_g$, with $\Gamma_g \in \mathbb{R}^{C_{\text{in}}/G \times C_{\text{in}}/G}$ computed from backpropagated gradients, and the input-based Kronecker factor

$$\Omega_g = \frac{1}{N} \sum_{n=1}^{N} [\![\mathbf{Y}_{n,g}]\!]_{\text{T}} [\![\mathbf{Y}_{n,g}]\!]_{\text{T}}^\top \in \mathbb{R}^{C_{\text{out}}/G K_1 K_2 \times C_{\text{out}}/G K_1 K_2} .$$

For KFAC-reduce, we have $G(w_g) \approx \hat{\Omega}_g \otimes \hat{\Gamma}_g$, with $\hat{\Gamma}_g \in \mathbb{R}^{C_{\text{in}}/G \times C_{\text{in}}/G}$ computed from backpropagated gradients, and the input-based Kronecker factor

$$\hat{\Omega}_g = \frac{1}{N(O_1 O_2)^2} \sum_{n=1}^{N} \left( \mathbf{1}_{I_1 I_2}^\top [\![\mathbf{Y}_{n,g}]\!]_{\text{T}} \right) \left( \mathbf{1}_{I_1 I_2}^\top [\![\mathbf{Y}_{n,g}]\!]_{\text{T}} \right)^\top \in \mathbb{R}^{C_{\text{out}}/G K_1 K_2 \times C_{\text{out}}/G K_1 K_2} .$$

## B.5 Further Operations & Extensive Overview

**Consecutive convolutions:** We can chain two, or more, convolutions into a single TN diagram (Figure B14) to obtain a deep linear CNN [65] similar to deep linear networks which are popular for analytical studies.

**Convolution weight/input JVPs:** In the main text, we derived the Jacobians of convolution (§3.1) which can be used to derive the JVPs. A JVP propagates perturbations $\mathbf{V}^{(\mathbf{W})} \in \mathbb{R}^{C_{\text{out}} \times C_{\text{in}} \times K_1 \times K_2}$ and $\mathbf{V}^{(\mathbf{X})} \in \mathbb{R}^{C_{\text{in}} \times I_1 \times I_2}$ in the input space to perturbations in the output space by contracting the perturbation with the Jacobian. See Table B3 for the general `einsum` expressions.

**Batched convolution weight VJP:** To obtain per-sample gradients, the weight VJP must be carried out without summing over the batch axis which amounts to keeping the batch index in the output index tuple.

**VJPs and JVPs of `im2col`:** With the TN differentiation technique described in §3.1 we can compute the Jacobian of the unfolding operation, then contract it with perturbations $V^{(\mathbf{X})} \in \mathbb{R}^{C_{\text{in}} \times K_1 \times K_2}$ in input space to obtain the JVP, or with perturbations $V^{([\![\mathbf{X}]\!])} \in \mathbb{R}^{O_1 O_2 \times C_{\text{in}} K_1 K_2}$ to obtain the VJP.

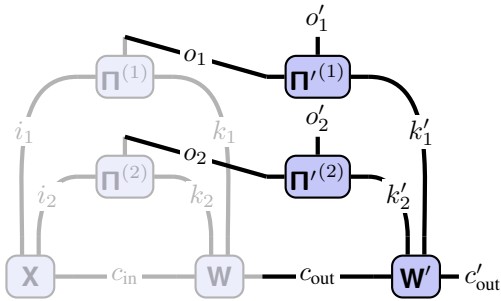

Figure B14: TN of two consecutive convolutions without groups and without batch axis.

**Approximate Hessian diagonals (HesScale/BL89):**  Becker & Lecun [4], Elsayed et al. [22] proposed approximate procedures for the Hessian diagonal which cost roughly a gradient. They can be understood as modifications of the Hessian backpropagation equations from Dangel et al. [16].

Consider a layer with input $x$, output $y$, and weights $w$ inside a sequential feedforward neural network (for a convolutional layer, these correspond to the flattened input, output, and kernel). To compute per-layer Hessians of a loss $\ell$, each layer backpropagates its incoming Hessian $\nabla_{\boldsymbol{y}}^2 \ell$ according to [16]

$$\nabla_{\boldsymbol{x}}^2 \ell = (\boldsymbol{J_x y})^\top \nabla_{\boldsymbol{y}}^2 \ell (\boldsymbol{J_x y}) + \sum_i \frac{\partial \ell}{\partial y_i} \nabla_{\boldsymbol{x}}^2 y_i \,,$$
$$\nabla_{\boldsymbol{w}}^2 \ell = (\boldsymbol{J_w y})^\top \nabla_{\boldsymbol{y}}^2 \ell (\boldsymbol{J_w y}) + \sum_i \frac{\partial \ell}{\partial y_i} \nabla_{\boldsymbol{w}}^2 y_i \,.$$
(B15)

The scheme of [4, 22] imposes diagonal structure on the backpropagated quantity. A layer receives a backpropagated diagonal $\boldsymbol{d}^{(\boldsymbol{y})}$ such that $\mathrm{diag}(\boldsymbol{d}^{(\boldsymbol{y})}) \approx \nabla_{\boldsymbol{y}}^2 \ell$, and backpropagates it according to Equation (B15), but with a post-processing step to obtain a diagonal backpropagated quantity,

$$\boldsymbol{d}^{(\boldsymbol{x})} = \mathrm{diag}\left((\boldsymbol{J_x y})^\top \mathrm{diag}(\boldsymbol{d}^{(\boldsymbol{y})})(\boldsymbol{J_x y})\right) + \mathrm{diag}\left(\sum_i \frac{\partial \ell}{\partial y_i} \nabla_{\boldsymbol{x}}^2 y_i\right) \,,$$
$$\boldsymbol{d}^{(\boldsymbol{w})} = \mathrm{diag}\left((\boldsymbol{J_w y})^\top \mathrm{diag}(\boldsymbol{d}^{(\boldsymbol{w})})(\boldsymbol{J_w y})\right) + \mathrm{diag}\left(\sum_i \frac{\partial \ell}{\partial y_i} \nabla_{\boldsymbol{w}}^2 y_i\right) \,,$$
(B16)

where $\mathrm{diag}(\boldsymbol{d}^{(\boldsymbol{x})}) \approx \nabla_{\boldsymbol{x}}^2 \ell$ and $\mathrm{diag}(\boldsymbol{d}^{(\boldsymbol{w})}) \approx \nabla_{\boldsymbol{w}}^2 \ell$ is an approximation to the Hessian diagonal.

For convolutional layers, which are linear in the input and weight, the second summands are zero due to $\nabla_{\boldsymbol{x}}^2 y_i = \boldsymbol{0} = \nabla_{\boldsymbol{w}}^2 y_i$. The first terms of Equation (B16) require (i) embedding a diagonal vector into a matrix, (ii) applying MJPs and JMPs, and (iii) extracting the result's diagonal. Those can be expressed as a single TN. We show the diagrams in Figure B15, using tensors rather than their flattened versions, that is $(\boldsymbol{x}, \boldsymbol{y}, \boldsymbol{w}, \boldsymbol{d}^{(\boldsymbol{x})}, \boldsymbol{d}^{(\boldsymbol{y})}, \boldsymbol{d}^{(\boldsymbol{w})}) \to (\mathbf{X}, \mathbf{Y}, \mathbf{W}, \mathbf{D}^{(\mathbf{X})}, \mathbf{D}^{(\mathbf{Y})}, \mathbf{D}^{(\mathbf{W})})$.

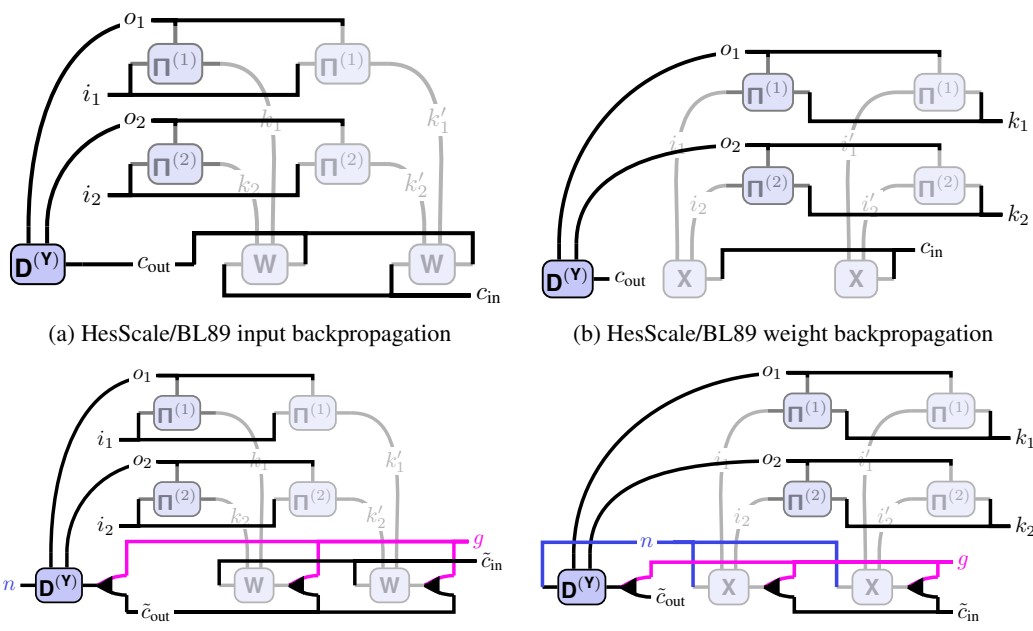

(a) HesScale/BL89 input backpropagation

(b) HesScale/BL89 weight backpropagation

(c) HesScale/BL89 input backpropagation (+ batch, groups)

(d) HesScale/BL89 weight backpropagation (+ batch, groups)

Figure B15: TN diagrams for HesScale/BL89 [4, 22] backpropagations through convolutional layers to approximate the Hessian diagonals $\mathbf{D}^{(\mathbf{X})}$, $\mathbf{D}^{(\mathbf{W})}$. JMPs and MJPs are shaded. (a, b) show the simple versions without batching and without channel groups. (c, d) include batching and channel groups.

Table B3: Extensive list of convolution and related operations (extension from Table 1 in the main text). All operations consider two spatial dimensions and support batching and channel groups. Generalization to other dimensions follow by introducing more spatial indices `i_3`, `o_3`, `...` and kernel indices `k_3`, `....`

| Operation | Operands | Contraction string (`einops` [59] convention) |
|---|---|---|
| Convolution (no bias) [29] | $\mathbf{X}, \mathbf{\Pi}^{(1)}, \mathbf{\Pi}^{(2)}, \mathbf{W}$ | `"n (g c_in) i1 i2, i1 o1 k1, i2 o2 k2, (g c_out) c_in k1 k2 -> n (g c_out) o1 o2"` |
| Unfolded input (`im2col`, $[\![\mathbf{X}]\!]$) | $\mathbf{X}, \mathbf{\Pi}^{(1)}, \mathbf{\Pi}^{(2)}$ | `"n c_in i1 i2, i1 o1 k1, i2 o2 k2 -> n (c_in k1 k2) (o1 o2)"` |
| Unfolded kernel (Toeplitz) | $\mathbf{\Pi}^{(1)}, \mathbf{\Pi}^{(2)}, \mathbf{W}$ | `"i1 o1 k1, i2 o2 k2, c_out c_in k1 k2 -> (c_out o1 o2) (c_in i1 i2)"` |
| Folded output (`col2im`) | $\mathbf{Y}, \mathbf{\Pi}^{(1)}, \mathbf{\Pi}^{(2)}$ | `"n (g c_out) o1 o2, i1 o1 k1, i2 o2 k2 -> n (g c_in) i1 i2"` |
| Transpose-unfolded input ($[\![\mathbf{Y}]\!]_\mathrm{T}$) | $\mathbf{Y}, \mathbf{\Pi}^{(1)}, \mathbf{\Pi}^{(2)}$ | `"n (g c_out) o1 o2, i1 o1 k1, i2 o2 k2 -> n (g c_in k1 k2) i1 i2"` |
| Convolution weight VJP | $\mathbf{X}, \mathbf{\Pi}^{(1)}, \mathbf{\Pi}^{(2)}, \mathbf{V}^{(\mathbf{Y})}$ | `"n (g c_in) i1 i2, i1 o1 k1, i2 o2 k2, n (g c_out) o1 o2 -> c_out c_in k1 k2"` |
| Convolution input VJP (transpose convolution) | $\mathbf{W}, \mathbf{\Pi}^{(1)}, \mathbf{\Pi}^{(2)}, \mathbf{V}^{(\mathbf{Y})}$ | `"(g c_out) c_in k1 k2, i1 o1 k1, i2 o2 k2, n (g c_out) o1 o2 -> n (g c_in) i1 i2"` |
| Convolution weight VJP (per-sample/batched) [58] | $\mathbf{X}, \mathbf{\Pi}^{(1)}, \mathbf{\Pi}^{(2)}, \mathbf{V}^{(\mathbf{Y})}$ | `"n (g c_in) i1 i2, i1 o1 k1, i2 o2 k2, n (g c_out) o1 o2 -> n (g c_out) c_in k1 k2"` |
| Convolution weight JVP | $\mathbf{X}, \mathbf{\Pi}^{(1)}, \mathbf{\Pi}^{(2)}, \mathbf{V}^{(\mathbf{W})}$ | `"n (g c_in) i1 i2, i1 o1 k1, i2 o2 k2, (g c_out) c_in k1 k2 -> n (g c_out) o1 o2"` |
| Convolution input JVP | $\mathbf{W}, \mathbf{\Pi}^{(1)}, \mathbf{\Pi}^{(2)}, \mathbf{V}^{(\mathbf{X})}$ | `"(g c_out) c_in i1 i2, i1 o1 k1, i2 o2 k2, n (g c_in) i1 i2 -> n (g c_out) o1 o2"` |
| `im2col` VJP | $\mathbf{\Pi}^{(1)}, \mathbf{\Pi}^{(2)}, \mathbf{V}^{([\![\mathbf{X}]\!])}$ | `"i1 o1 k1, i2 o2 k2, n (c_in k1 k2) (o1 o2) -> n c_in i1 i2"` |
| `im2col` JVP | $\mathbf{\Pi}^{(1)}, \mathbf{\Pi}^{(2)}, \mathbf{V}^{(\mathbf{X})}$ | `"i1 o1 k1, i2 o2 k2, n c_in i1 i2 -> n (c_in k1 k2) (o1 o2)"` |
| KFC/KFAC-expand [27, 23] | $\mathbf{X}, \mathbf{\Pi}^{(1)}, \mathbf{\Pi}^{(2)}, \mathbf{X}, \mathbf{\Pi}^{(1)}, \mathbf{\Pi}^{(2)}$ | `"n (g c_in) i1 i2, i1 o1 k1, i2 o2 k2, n (g c_in_) i1_ i2_, i1_ o1 k1_, i2_ o2 k2_ -> g (c_in k1 k2) (c_in_ k1_ k2_)"` |
| KFAC-reduce [23] | $\mathbf{X}, \mathbf{\Pi}^{(1)}, \mathbf{\Pi}^{(2)}, \mathbf{X}, \mathbf{\Pi}^{(1)}, \mathbf{\Pi}^{(2)}$ | `"n (g c_in) i1 i2, i1 o1 k1, i2 o2 k2, n (g c_in_) i1_ i2_, i1_ o1_ k1_, i2_ o2_ k2_ -> g (c k1 k2) (c_ k1_ k2_)"` |
| KFC/KFAC-expand for transpose convolution | $\mathbf{Y}, \mathbf{\Pi}^{(1)}, \mathbf{\Pi}^{(2)}, \mathbf{Y}, \mathbf{\Pi}^{(1)}, \mathbf{\Pi}^{(2)}$ | `"n (g c_out) o1 o2, i1 o1 k1, i2 o2 k2, n (g c_out_) o1_ o2_, i1_ o1 k1_, i2_ o2 k2_ -> g (c_out k1 k2) (c_out_ k1_ k2_)"` |
| KFAC-reduce for transpose convolution | $\mathbf{Y}, \mathbf{\Pi}^{(1)}, \mathbf{\Pi}^{(2)}, \mathbf{Y}, \mathbf{\Pi}^{(1)}, \mathbf{\Pi}^{(2)}$ | `"n (g c_out) o1 o2, i1 o1 k1, i2 o2 k2, n (g c_out_) o1_ o2_, i1_ o1_ k1_, i2_ o2_ k2_ -> g (c_out k1 k2) (c_out_ k1_ k2_)"` |
| GGN Gram/empirical NTK matrix [18, 51, 49] | $\mathbf{X}, \mathbf{\Pi}^{(1)}, \mathbf{\Pi}^{(2)}, \mathbf{S}^{(\mathbf{Y})}, \mathbf{X}, \mathbf{\Pi}^{(1)}, \mathbf{\Pi}^{(2)}, \mathbf{S}^{(\mathbf{Y})}$ | `"n (g c_in) i1 i2, i1 o1 k1, i2 o2 k2, c n (g c_out) o1 o2, n_ (g c_in) i1_ i2_, i1_ o1_ k1, i2_ o2_ k2, c_ n_ (g c_out) o1_ o2_ -> (c n) (c_ n_)"` |
| GGN/Fisher diagonal [17, 51] | $\mathbf{X}, \mathbf{\Pi}^{(1)}, \mathbf{\Pi}^{(2)}, \mathbf{S}^{(\mathbf{Y})}, \mathbf{X}, \mathbf{\Pi}^{(1)}, \mathbf{\Pi}^{(2)}, \mathbf{S}^{(\mathbf{Y})}$ | `"n (g c_in) i1 i2, i1 o1 k1, i2 o2 k2, c n (g c_out) o1 o2, n (g c_in) i1_ i2_, i1_ o1_ k1, i2_ o2_ k2, c n (g c_out) o1_ o2_ -> (g c_out) c_in k1 k2"` |
| GGN/Fisher diagonal (per-sample/batched) | $\mathbf{X}, \mathbf{\Pi}^{(1)}, \mathbf{\Pi}^{(2)}, \mathbf{S}^{(\mathbf{Y})}, \mathbf{X}, \mathbf{\Pi}^{(1)}, \mathbf{\Pi}^{(2)}, \mathbf{S}^{(\mathbf{Y})}$ | `"n (g c_in) i1 i2, i1 o1 k1, i2 o2 k2, c n (g c_out) o1 o2, n (g c_in) i1_ i2_, i1_ o1_ k1, i2_ o2_ k2, c n (g c_out) o1_ o2_ -> n (g c_out) c_in k1 k2"` |
| Approximate weight Hessian diagonal [4, 22] | $\mathbf{X}, \mathbf{\Pi}^{(1)}, \mathbf{\Pi}^{(2)}, \mathbf{D}^{(\mathbf{Y})}, \mathbf{X}, \mathbf{\Pi}^{(1)}, \mathbf{\Pi}^{(2)}$ | `"n (g c_in) i1 i2, i1 o1 k1, i2 o2 k2, n (g c_out) o1 o2, n (g c_in) i1_ i2_, i1_ o1 k1, i2_ o2 k2 -> (g c_out) c_in k1 k2"` |
| Approximate input Hessian diagonal [4, 22] | $\mathbf{W}, \mathbf{\Pi}^{(1)}, \mathbf{\Pi}^{(2)}, \mathbf{D}^{(\mathbf{Y})}, \mathbf{W}, \mathbf{\Pi}^{(1)}, \mathbf{\Pi}^{(2)}$ | `"(g c_out) c_in k1 k2, i1 o1 k1, i2 o2 k2, n (g c_out) o1 o2, (g c_out) c_in k1_ k2_, i1 o1 k1_, i2 o2 k2_ -> n (g c_in) i1 i2"` |
| Approximate weight Hessian diagonal (per-sample/batched) | $\mathbf{X}, \mathbf{\Pi}^{(1)}, \mathbf{\Pi}^{(2)}, \mathbf{D}^{(\mathbf{Y})}, \mathbf{X}, \mathbf{\Pi}^{(1)}, \mathbf{\Pi}^{(2)}$ | `"n (g c_in) i1 i2, i1 o1 k1, i2 o2 k2, n (g c_out) o1 o2, n (g c_in) i1_ i2_, i1_ o1 k1, i2_ o2 k2 -> n (g c_out) c_in k1 k2"` |

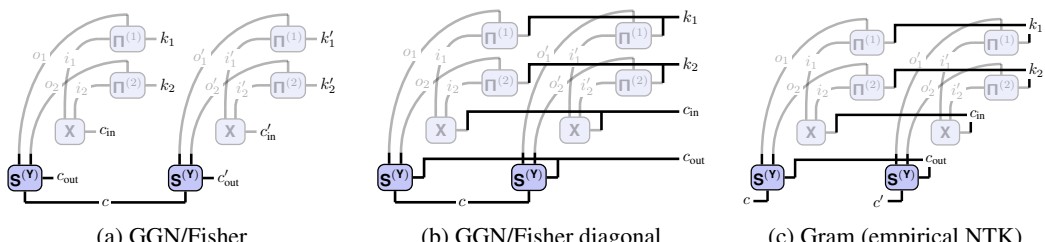

(a) GGN/Fisher      (b) GGN/Fisher diagonal      (c) Gram (empirical NTK)

Figure C16: TN composition and sub-tensor extraction for second-order information. Weight MJPs from Figure 3c are shaded. (a) exact and (b) diagonal of the kernel's GGN (the same applies to structurally similar matrices like the gradient covariance [35]). (c) TN of the GGN Gram matrix.

## C   Exact Second-Order Information

Here, we look at computing second-order information of a loss w.r.t. to the kernel of a convolution. Its computation can be phrased as backpropagation with a final extraction step [19] which contains less standard operations like Jacobian-matrix products (JMPs) and sub-tensor extraction. TNs can express this extraction step in a single diagram.

Consider a datum $(\boldsymbol{x}, \boldsymbol{t})$ and its loss $\ell(\boldsymbol{w}) = \ell(\boldsymbol{f}, \boldsymbol{t})$ where $\boldsymbol{f} := f_{\boldsymbol{w}}(\boldsymbol{x}) \in \mathbb{R}^C$ is the prediction of a CNN with a convolution with flattened kernel $\boldsymbol{w}$ and flattened output $\boldsymbol{y}$ (derivations carry over to a batch loss). The kernel's generalized Gauss-Newton (GGN) matrix [63] $\boldsymbol{G}(\boldsymbol{w}) = (\boldsymbol{J}_{\boldsymbol{w}}\boldsymbol{f})^{\top}\nabla_{\boldsymbol{f}}^2 \ell(\boldsymbol{J}_{\boldsymbol{w}}\boldsymbol{f}) \in \mathbb{R}^{C_{\text{out}}C_{\text{in}}K_1K_2 \times C_{\text{out}}C_{\text{in}}K_1K_2}$ is a positive semi-definite Hessian proxy preferred by many applications [e.g. 20, 45] and coincides with the Fisher information matrix for many common losses [46]. It is the self-outer product of a backpropagated symmetric factorization $\boldsymbol{S}^{(\boldsymbol{y})} = (\boldsymbol{J}_{\boldsymbol{y}}\boldsymbol{f})^{\top}\boldsymbol{S}^{(\boldsymbol{f})} \in \mathbb{R}^{C_{\text{out}}O_1O_2 \times C}$ of the loss Hessian, $\nabla_{\boldsymbol{f}}^2 \ell(\boldsymbol{f}, \boldsymbol{y}) = \boldsymbol{S}^{(\boldsymbol{f})}(\boldsymbol{S}^{(\boldsymbol{f})})^{\top}$. During backpropagation, the convolution extracts information about $\boldsymbol{G}(\boldsymbol{w}) = (\boldsymbol{J}_{\boldsymbol{w}}\boldsymbol{y})^{\top}\boldsymbol{S}^{(\boldsymbol{y})}(\boldsymbol{S}^{(\boldsymbol{y})})^{\top}\boldsymbol{J}_{\boldsymbol{w}}\boldsymbol{y}$.

In TN notation, this is easy to express without flattening: We simply compose two VJP diagrams from Figure 3c with an extra leg (MJP) and add the outer-product contraction to obtain the tensor version $\mathbf{G}(\mathbf{W}) \in \mathbb{R}^{C_{\text{out}} \times C_{\text{in}} \times K_1 \times K_2 \times C_{\text{out}} \times C_{\text{in}} \times K_1 \times K_2}$ of $\boldsymbol{G}(\boldsymbol{w})$ (Figure C16a). The GGN is often further approximated by sub-tensors as it is too large. These slicing operations are also easy to integrate into the diagrams, e.g. to extract diagonal elements (Figure C16b [17, 51]), or mini-block diagonals (Figure B11 [16, 3]). This also removes redundant computations compared to computing, then slicing, the matrix. The same ideas apply to the GGN Gram matrix $(\boldsymbol{S}^{(\boldsymbol{w})})^{\top}\boldsymbol{S}^{(\boldsymbol{w})} \in \mathbb{R}^{C \times C}$ (Figure C16c). It contains the GGN spectrum [18] and is related to the empirical NTK for square loss [49].

# D Implementation Details

Here we present details on the index pattern computation, and additional transformations.

## D.1 Index Pattern Tensor Computation for Convolutions

Algorithm D1 lists pseudo-code for the index pattern computation from the convolution hyper-parameters $K, S, P, D$, and the spatial input dimension $I$, that is $\mathbf{\Pi}(I, K, S, P, D)$. *Unlike in the main text, we use 0-based indexing which is more common in numerical libraries.* For self-consistency, we re-state the relation of the hyper-parameters to output dimension from [21, Relationship 15],

$$O(I, K, S, P, D) = 1 + \left\lfloor \frac{I + 2P - K - (K-1)(D-1)}{S} \right\rfloor . \tag{D17}$$

---

**Algorithm D1** Computing the convolution index pattern tensor $\mathbf{\Pi}$ for a spatial dimension.

---

**Require:** Input size $I \in \mathbb{N}^+$, kernel size $K \in \mathbb{N}^+$, stride $S \in \mathbb{N}^+$, padding $P \in \mathbb{N}_0^+$, dilation $D \in \mathbb{N}^+$

$O \leftarrow 1 + \left\lfloor \frac{I+2P-K-(K-1)(D-1)}{S} \right\rfloor$      ▷ Compute output dimension [21, Relationship 15]

$\mathbf{\Pi} \leftarrow \mathbf{0}_{I \times O \times K}$      ▷ Initialize index pattern tensor

**for** $o = 0, \dots, O-1, k = 0, \dots, K-1$ **do**      ▷ Use 0-based indexing!

     $i \leftarrow kD + oS - P$      ▷ Reconstruct contributing input element

     **if** $0 \leq i \leq I-1$ **then**      ▷ Check in bounds

         $\mathbf{\Pi}_{i,o,k} \leftarrow 1$

     **end if**

**end for**

**return** Index pattern tensor $\mathbf{\Pi} \in \{0, 1\}^{I \times O \times K}$

---

## D.2 Index Pattern Tensor for Standalone Transpose Convolution

Although a transpose convolution is defined w.r.t. a reference convolution with hyper-parameters $K, S, P, D$, most libraries offer standalone implementations of transpose convolution. We describe the transpose convolution by its associated convolution, that is as a mapping from $\mathbb{R}^{C_{\text{out}} \times O_1 \times O_2}$ (the convolution's output space) to $\mathbb{R}^{C_{\text{in}} \times I_1 \times I_2}$ (the convolution's input space). For convolution with $S > 1$, we cannot infer $I$ from $O, K, S, P, D$, as multiple $I$s map to the same $O$ if $(I + 2P - K - (K-1)(D-1)) \mod S \neq 0$ (see the floor operation in Algorithm D1). We need to either supply $I$ directly, or the remainder

$$A = I + 2P - K - (K-1)(D-1) - S(O-1)$$

(often called `output_padding`) to make $I$ unambiguous. Then, we compute

$$I = (O-1)S - 2P + K + (K-1)(D-1) + A . \tag{D18}$$

to get $I(O, A)$ and call Algorithm D1 to obtain $\mathbf{\Pi}(I(O, A), K, S, P, D)$.

## D.3 Details on Index Pattern Simplifications

In the following, we will assume the absence of boundary pixels that don't overlap with the kernel, that is

$$I + 2P - (K + (K-1)(D-1)) \mod S = 0 , \tag{D19}$$

where the floor operation in $O(I, K, S, P, D)$ is obsolete. This can always be assured by narrowing $\mathbf{X}$ before a convolution. Based on our hyper-parameter analysis of real-world CNNs (§E), we identify:

**Transformation D1 (Dense convolutions)** *Assume Equation* (D19). *For $K = S$ with default padding and dilation ($P = 0$, $D = 1$), patches are adjacent non-overlapping tiles, accessible by un-grouping the input index $i$ into a tuple index $(\tilde{i}, \tilde{k})$ of size $I/K \times K$:*

$$[\mathbf{\Pi}(I, K, K, 0, 1)]_{i,o,k} = [\mathbf{\Pi}(I, K, K, 0, 1)]_{(\tilde{i}, \tilde{k}),o,k} = \delta_{\tilde{i},o} \delta_{\tilde{k},k} .$$

*Point-wise convolutions ($K = S = 1$) are a special case with pattern $[\mathbf{\Pi}(I, 1, 1, 0, 1)]_{i,o,k} = \delta_{i,o}$.*

Point-wise convolutions with $K = S = 1$ are common in DenseNets [33], MobileNets [31, 60] and ResNets [30]. InceptionV3 [69] has 2d 'mixed dense' convolutions that are point-wise along one spatial dimension. ConvNeXt [42] uses dense convolutions with $K = S \in \{2, 4\}$.

**Transformation D2 (Down-sampling convolutions)** *For $S > K$ with default padding and dilation ($P = 0$, $D = 1$), some elements do not overlap with the kernel. If the input dimension $i$ is summed, all participating tensors can be pruned to remove the explicit zeros. Assume $I \mod S = 0$. Then, pruning amounts to un-grouping $i$ into $(i', s)$ of size $I/S \times S$, narrowing $s$ to $K$ entries, and grouping back into an index $\tilde{i}$ of size $KI/S$. After pruning, the index pattern represents a dense convolution with input size $KI/S$, kernel size $K$, and stride $K$. In a contraction with some tensor $\mathbf{V}$,*

$$\sum_{i=1}^{I} [\mathbf{V}]_{\ldots,i,\ldots} [\mathbf{\Pi}(I, K, S > K, 0, 1)]_{i,o,k} = \sum_{\tilde{i}=1}^{I/S} [\tilde{\mathbf{V}}]_{\ldots,\tilde{i},\ldots} [\mathbf{\Pi}(KI/S, K, K, 0, 1)]_{\tilde{i},o,k}$$

*with sub-tensor $[\tilde{\mathbf{V}}]_{\ldots,\tilde{i},\ldots} = [[\mathbf{V}]_{\ldots,(i',s),\ldots}]_{\ldots,(:,:K),\ldots}$ where $:K$ means narrowing to $K$ elements.*

Transformation D2 converts down-sampling convolutions to dense convolutions, which can be further simplified with Transformation D1. We find down-sampling convolutions with $S = 2 > K = 1$ in ResNet18 [30], ResNext101 [72], and WideResNet101 [73]. Those convolutions discard 75 % of their input! Knowledge that an operation only consumes a fraction of its input could be used to eliminate those 'dead' computations in preceding operations, reducing FLOPS and memory.

**Transformation D3 (Kernel-output dimension swap)** *Assume Equation* (D19). *Transposing kernel and output dimensions in an index pattern yields another index pattern with same input size, kernel size $O(I, K, S, P, D)$, and swapped stride and dilation:*

$$[\mathbf{\Pi}(I, K, S, P, D)]_{i,o,k} = [\mathbf{\Pi}(I, O, D, P, S)]_{i,k,o} \ .$$

This transformation is easy to see from the symmetry of $(k, D)$ and $(o, S)$ in Equation (7) and $O(I, K, S, P, D)$. It converts index pattern contractions over output into kernel dimensions, like in convolutions. An example is the weight VJP from Figure 3c, which—after swapping kernel and output dimensions—resembles the TN for convolution from Figure 2 with kernel $\mathbf{V}$. Rochette et al. [58] use this to phrase the computation of per-example gradients as convolution.

§D.3 presents more properties of $\mathbf{\Pi}$ based on the sub-sampling interpretation of stride and dilation along the output and kernel dimensions. We also provide a transformation for swapping input and output dimensions, relating convolution and transpose convolution as described in [21].

For completeness, we state additional index pattern tensor properties here (using 1-based indexing):

**Transformation D4 (Sub-sampling interpretation of stride)** *Strided convolutions ($S > 1$) sub-sample non-strided convolutions along the output dimension, ignoring all but every $S$th output [21]. In other words, $[\mathbf{\Pi}(I, K, S, P, D)]_{i,o,k} = [\mathbf{\Pi}(I, K, 1, P, D)]_{i,1+S(o-1),k}$ or, in tensor notation ($[\cdot]_{::S}$ denotes slicing with steps of $S$),*

$$\mathbf{\Pi}(I, K, S, P, D) = [\mathbf{\Pi}(I, K, 1, P, D)]_{:,::S,:} \ .$$

**Transformation D5 (Sub-sampling interpretation of dilation)** *Dilated convolutions ($D > 1$) with kernel size $K$ sub-sample the kernel of a non-dilated convolution of kernel size $K + (D - 1)(K - 1)$, ignoring all but every $D$th kernel element. In other words, $[\mathbf{\Pi}(I, K, S, P, D)]_{i,o,k} = [\mathbf{\Pi}(I, K + (K - 1)(D - 1), S, P, 1)]_{i,o,1+D(k-1)}$ or, in tensor notation,*

$$\mathbf{\Pi}(I, K, S, P, D) = [\mathbf{\Pi}(I, K + (K - 1)(D - 1), S, P, 1)]_{:,:,::D} \ .$$

**Transformation D6 (Transpose convolution as convolution)** *Assume Equation* (D19). *Consider a non-strided ($S = 1$), non-dilated ($D = 1$) convolution with index pattern $\mathbf{\Pi}(I, K, 1, P, 1)$ and output dimension $O(I, K, 1, P, 1)$. Transposing the spatial dimensions and flipping the kernel dimension yields another index pattern with modified padding $P' = K - P - 1$. In other words, for all $i = 1, \ldots, I$, $k = 1, \ldots, K$, $o = 1, \ldots, O$*

$$[\mathbf{\Pi}(I, K, 1, P, 1)]_{i,o,k} = [\mathbf{\Pi}(O, K, 1, P', 1)]_{o,i,K+1-k} \ .$$

# E    Convolution Layer Hyper-parameter Analysis

Here we give an overview of and characterize convolutions in popular architectures (see Table E4). We include moderately deep CNNs on Fashion MNIST, CIFAR-10, and CIFAR-100 from the DeepOBS benchmark [62], and deep CNNs on ImageNet (AlexNet, ResNet18, InceptionV3, MobileNetV2, ResNext101). Regarding the hyper-parameters, we make the following observations:

- Many CNNs do not use a bias term. This is because the output of those layers feeds directly into a batch normalization layer, which is invariant under the addition of a bias term.

- All investigated convolutions use default dilation.

- Group convolutions are rarely used. MobileNetV2 and ConvNeXt-base (Tables E4g and E4i) use group convolutions that interpret each individual channel as a group. ResNext101 (Table E4f) uses group convolutions that interpret a collection of channels as a group. ConvNeXt-base (Table E4g) uses dense convolutions with $P = 0$ and $S = K \in \{2, 4\}$.

- Many networks use dense convolutions, that is convolutions with unit kernel size ($K = 1$), unit stride ($S = 1$), and no padding ($P = 0$). These convolutions have a trivial index pattern and can therefore be simplified.

- InceptionV3 (Table E4h) uses two-dimensional convolutions with one trivial dimension ('mixed dense') with unit kernel size, unit stride, and no padding along one direction. For this spatial dimension, the index pattern can be simplified.

- ResNet18 (Table E4e) and ResNext101 (Table E4f) use convolutions with $S > K$ for down-sampling whose kernel only overlaps with a fraction of the input. The index pattern can be simplified.

Table E4: Hyper-parameters of convolutions in different CNNs. For convolutions with identical hyper-parameters, we only show one instance and its multiplicity.

### (a) 3c3d, CIFAR-10 (3, 32, 32)

| Name (count) | Input shape | Output shape | Kernel | Stride | Padding | Dilation | Groups | Bias | Type |
|---|---|---|---|---|---|---|---|---|---|
| conv1.0 (1) | (3, 32, 32) | (64, 28, 28) | (5, 5) | (1, 1) | (0, 0) | (1, 1) | 1 | Yes | General |
| conv2.0 (1) | (64, 14, 14) | (96, 12, 12) | (3, 3) | (1, 1) | (0, 0) | (1, 1) | 1 | Yes | General |
| conv3.1 (1) | (96, 8, 8) | (128, 6, 6) | (3, 3) | (1, 1) | (0, 0) | (1, 1) | 1 | Yes | General |

### (b) 2c2d, Fashion MNIST (1, 28, 28)

| Name (count) | Input shape | Output shape | Kernel | Stride | Padding | Dilation | Groups | Bias | Type |
|---|---|---|---|---|---|---|---|---|---|
| conv1.1 (1) | (1, 32, 32) | (32, 28, 28) | (5, 5) | (1, 1) | (0, 0) | (1, 1) | 1 | Yes | General |
| conv2.1 (1) | (32, 18, 18) | (64, 14, 14) | (5, 5) | (1, 1) | (0, 0) | (1, 1) | 1 | Yes | General |

### (c) All-CNN-C, CIFAR-100 (3, 32, 32)

| Name (count) | Input shape | Output shape | Kernel | Stride | Padding | Dilation | Groups | Bias | Type |
|---|---|---|---|---|---|---|---|---|---|
| conv1.1 (1) | (3, 34, 34) | (96, 32, 32) | (3, 3) | (1, 1) | (0, 0) | (1, 1) | 1 | Yes | General |
| conv2.1 (1) | (96, 34, 34) | (96, 32, 32) | (3, 3) | (1, 1) | (0, 0) | (1, 1) | 1 | Yes | General |
| conv3.1 (1) | (96, 33, 33) | (96, 16, 16) | (3, 3) | (2, 2) | (0, 0) | (1, 1) | 1 | Yes | General |
| conv4.1 (1) | (96, 18, 18) | (192, 16, 16) | (3, 3) | (1, 1) | (0, 0) | (1, 1) | 1 | Yes | General |
| conv5.1 (1) | (192, 18, 18) | (192, 16, 16) | (3, 3) | (1, 1) | (0, 0) | (1, 1) | 1 | Yes | General |
| conv6.1 (1) | (192, 17, 17) | (192, 8, 8) | (3, 3) | (2, 2) | (0, 0) | (1, 1) | 1 | Yes | General |
| conv7.0 (1) | (192, 8, 8) | (192, 6, 6) | (3, 3) | (1, 1) | (0, 0) | (1, 1) | 1 | Yes | General |
| conv8.1 (1) | (192, 6, 6) | (192, 6, 6) | (1, 1) | (1, 1) | (0, 0) | (1, 1) | 1 | Yes | Dense |
| conv9.1 (1) | (192, 6, 6) | (100, 6, 6) | (1, 1) | (1, 1) | (0, 0) | (1, 1) | 1 | Yes | Dense |

### (d) AlexNet, ImageNet (3, 256, 256)

| Name (count) | Input shape | Output shape | Kernel | Stride | Padding | Dilation | Groups | Bias | Type |
|---|---|---|---|---|---|---|---|---|---|
| features.0 (1) | (3, 256, 256) | (64, 63, 63) | (11, 11) | (4, 4) | (2, 2) | (1, 1) | 1 | Yes | General |
| features.3 (1) | (64, 31, 31) | (192, 31, 31) | (5, 5) | (1, 1) | (2, 2) | (1, 1) | 1 | Yes | General |
| features.6 (1) | (192, 15, 15) | (384, 15, 15) | (3, 3) | (1, 1) | (1, 1) | (1, 1) | 1 | Yes | General |
| features.8 (1) | (384, 15, 15) | (256, 15, 15) | (3, 3) | (1, 1) | (1, 1) | (1, 1) | 1 | Yes | General |
| features.10 (1) | (256, 15, 15) | (256, 15, 15) | (3, 3) | (1, 1) | (1, 1) | (1, 1) | 1 | Yes | General |

### (e) ResNet18, ImageNet (3, 256, 256)

| Name (count) | Input shape | Output shape | Kernel | Stride | Padding | Dilation | Groups | Bias | Type |
|---|---|---|---|---|---|---|---|---|---|
| conv1 (1) | (3, 256, 256) | (64, 128, 128) | (7, 7) | (2, 2) | (3, 3) | (1, 1) | 1 | No | General |
| layer1.0.conv1 (4) | (64, 64, 64) | (64, 64, 64) | (3, 3) | (1, 1) | (1, 1) | (1, 1) | 1 | No | General |
| layer2.0.conv1 (1) | (64, 64, 64) | (128, 32, 32) | (3, 3) | (2, 2) | (1, 1) | (1, 1) | 1 | No | General |
| layer2.0.conv2 (3) | (128, 32, 32) | (128, 32, 32) | (3, 3) | (1, 1) | (1, 1) | (1, 1) | 1 | No | General |
| layer2.0.downsample.0 (1) | (64, 64, 64) | (128, 32, 32) | (1, 1) | (2, 2) | (0, 0) | (1, 1) | 1 | No | Down |
| layer3.0.conv1 (1) | (128, 32, 32) | (256, 16, 16) | (3, 3) | (2, 2) | (1, 1) | (1, 1) | 1 | No | General |
| layer3.0.conv2 (3) | (256, 16, 16) | (256, 16, 16) | (3, 3) | (1, 1) | (1, 1) | (1, 1) | 1 | No | General |
| layer3.0.downsample.0 (1) | (128, 32, 32) | (256, 16, 16) | (1, 1) | (2, 2) | (0, 0) | (1, 1) | 1 | No | Down |
| layer4.0.conv1 (1) | (256, 16, 16) | (512, 8, 8) | (3, 3) | (2, 2) | (1, 1) | (1, 1) | 1 | No | General |
| layer4.0.conv2 (3) | (512, 8, 8) | (512, 8, 8) | (3, 3) | (1, 1) | (1, 1) | (1, 1) | 1 | No | General |
| layer4.0.downsample.0 (1) | (256, 16, 16) | (512, 8, 8) | (1, 1) | (2, 2) | (0, 0) | (1, 1) | 1 | No | Down |

### (f) ResNext101_32x8d, ImageNet (3, 256, 256)

| Name (count) | Input shape | Output shape | Kernel | Stride | Padding | Dilation | Groups | Bias | Type |
|---|---|---|---|---|---|---|---|---|---|
| conv1 (1) | (3, 256, 256) | (64, 128, 128) | (7, 7) | (2, 2) | (3, 3) | (1, 1) | 1 | No | General |
| layer1.0.conv1 (2) | (64, 64, 64) | (256, 64, 64) | (1, 1) | (1, 1) | (0, 0) | (1, 1) | 1 | No | Dense |
| layer1.0.conv2 (3) | (256, 64, 64) | (256, 64, 64) | (3, 3) | (1, 1) | (1, 1) | (1, 1) | 32 | No | General |
| layer1.0.conv3 (5) | (256, 64, 64) | (256, 64, 64) | (1, 1) | (1, 1) | (0, 0) | (1, 1) | 1 | No | Dense |
| layer2.0.conv1 (1) | (256, 64, 64) | (512, 64, 64) | (1, 1) | (1, 1) | (0, 0) | (1, 1) | 1 | No | Dense |
| layer2.0.conv2 (1) | (512, 64, 64) | (512, 32, 32) | (3, 3) | (2, 2) | (1, 1) | (1, 1) | 32 | No | General |
| layer2.0.conv3 (7) | (512, 32, 32) | (512, 32, 32) | (1, 1) | (1, 1) | (0, 0) | (1, 1) | 1 | No | Dense |
| layer2.0.downsample.0 (1) | (256, 64, 64) | (512, 32, 32) | (1, 1) | (2, 2) | (0, 0) | (1, 1) | 1 | No | Down |
| layer2.1.conv2 (3) | (512, 32, 32) | (512, 32, 32) | (3, 3) | (1, 1) | (1, 1) | (1, 1) | 32 | No | General |
| layer3.0.conv1 (1) | (512, 32, 32) | (1024, 32, 32) | (1, 1) | (1, 1) | (0, 0) | (1, 1) | 1 | No | Dense |
| layer3.0.conv2 (1) | (1024, 32, 32) | (1024, 16, 16) | (3, 3) | (2, 2) | (1, 1) | (1, 1) | 32 | No | General |
| layer3.0.conv3 (45) | (1024, 16, 16) | (1024, 16, 16) | (1, 1) | (1, 1) | (0, 0) | (1, 1) | 1 | No | Dense |
| layer3.0.downsample.0 (1) | (512, 32, 32) | (1024, 16, 16) | (1, 1) | (2, 2) | (0, 0) | (1, 1) | 1 | No | Down |
| layer3.1.conv2 (22) | (1024, 16, 16) | (1024, 16, 16) | (3, 3) | (1, 1) | (1, 1) | (1, 1) | 32 | No | General |
| layer4.0.conv1 (1) | (1024, 16, 16) | (2048, 16, 16) | (1, 1) | (1, 1) | (0, 0) | (1, 1) | 1 | No | Dense |
| layer4.0.conv2 (1) | (2048, 16, 16) | (2048, 8, 8) | (3, 3) | (2, 2) | (1, 1) | (1, 1) | 32 | No | General |
| layer4.0.conv3 (5) | (2048, 8, 8) | (2048, 8, 8) | (1, 1) | (1, 1) | (0, 0) | (1, 1) | 1 | No | Dense |
| layer4.0.downsample.0 (1) | (1024, 16, 16) | (2048, 8, 8) | (1, 1) | (2, 2) | (0, 0) | (1, 1) | 1 | No | Down |
| layer4.1.conv2 (2) | (2048, 8, 8) | (2048, 8, 8) | (3, 3) | (1, 1) | (1, 1) | (1, 1) | 32 | No | General |

### (g) ConvNeXt-base, ImageNet (3, 256, 256)

| Name (count) | Input shape | Output shape | Kernel | Stride | Padding | Dilation | Groups | Bias | Type |
|---|---|---|---|---|---|---|---|---|---|
| features.0.0 (1) | (3, 256, 256) | (128, 64, 64) | (4, 4) | (4, 4) | (0, 0) | (1, 1) | 1 | Yes | Dense |
| features.1.0.block.0 (3) | (128, 64, 64) | (128, 64, 64) | (7, 7) | (1, 1) | (3, 3) | (1, 1) | 128 | Yes | General |
| features.2.1 (1) | (128, 64, 64) | (256, 32, 32) | (2, 2) | (2, 2) | (0, 0) | (1, 1) | 1 | Yes | Dense |
| features.3.0.block.0 (3) | (256, 32, 32) | (256, 32, 32) | (7, 7) | (1, 1) | (3, 3) | (1, 1) | 256 | Yes | General |
| features.4.1 (1) | (256, 32, 32) | (512, 16, 16) | (2, 2) | (2, 2) | (0, 0) | (1, 1) | 1 | Yes | Dense |
| features.5.0.block.0 (27) | (512, 16, 16) | (512, 16, 16) | (7, 7) | (1, 1) | (3, 3) | (1, 1) | 512 | Yes | General |
| features.6.1 (1) | (512, 16, 16) | (1024, 8, 8) | (2, 2) | (2, 2) | (0, 0) | (1, 1) | 1 | Yes | Dense |
| features.7.0.block.0 (3) | (1024, 8, 8) | (1024, 8, 8) | (7, 7) | (1, 1) | (3, 3) | (1, 1) | 1024 | Yes | General |

(h) InceptionV3, ImageNet (3, 299, 299)

| Name (count) | Input shape | Output shape | Kernel | Stride | Padding | Dilation | Groups | Bias | Type |
|---|---|---|---|---|---|---|---|---|---|
| Conv2d_1a_3x3.conv (1) | (3, 299, 299) | (32, 149, 149) | (3, 3) | (2, 2) | (0, 0) | (1, 1) | 1 | No | General |
| Conv2d_2a_3x3.conv (1) | (32, 149, 149) | (32, 147, 147) | (3, 3) | (1, 1) | (0, 0) | (1, 1) | 1 | No | General |
| Conv2d_2b_3x3.conv (1) | (32, 147, 147) | (64, 147, 147) | (3, 3) | (1, 1) | (1, 1) | (1, 1) | 1 | No | General |
| Conv2d_3b_1x1.conv (1) | (64, 73, 73) | (80, 73, 73) | (1, 1) | (1, 1) | (0, 0) | (1, 1) | 1 | No | Dense |
| Conv2d_4a_3x3.conv (1) | (80, 73, 73) | (192, 71, 71) | (3, 3) | (1, 1) | (0, 0) | (1, 1) | 1 | No | General |
| Mixed_5b.branch1x1.conv (2) | (192, 35, 35) | (64, 35, 35) | (1, 1) | (1, 1) | (0, 0) | (1, 1) | 1 | No | Dense |
| Mixed_5b.branch5x5_1.conv (1) | (192, 35, 35) | (48, 35, 35) | (1, 1) | (1, 1) | (0, 0) | (1, 1) | 1 | No | Dense |
| Mixed_5b.branch5x5_2.conv (3) | (48, 35, 35) | (64, 35, 35) | (5, 5) | (1, 1) | (2, 2) | (1, 1) | 1 | No | General |
| Mixed_5b.branch3x3dbl_2.conv (4) | (64, 35, 35) | (96, 35, 35) | (3, 3) | (1, 1) | (1, 1) | (1, 1) | 1 | No | General |
| Mixed_5b.branch3x3dbl_3.conv (3) | (96, 35, 35) | (96, 35, 35) | (3, 3) | (1, 1) | (1, 1) | (1, 1) | 1 | No | General |
| Mixed_5b.branch_pool.conv (1) | (192, 35, 35) | (32, 35, 35) | (1, 1) | (1, 1) | (0, 0) | (1, 1) | 1 | No | Dense |
| Mixed_5c.branch1x1.conv (3) | (256, 35, 35) | (64, 35, 35) | (1, 1) | (1, 1) | (0, 0) | (1, 1) | 1 | No | Dense |
| Mixed_5c.branch5x5_1.conv (1) | (256, 35, 35) | (48, 35, 35) | (1, 1) | (1, 1) | (0, 0) | (1, 1) | 1 | No | Dense |
| Mixed_5d.branch1x1.conv (4) | (288, 35, 35) | (64, 35, 35) | (1, 1) | (1, 1) | (0, 0) | (1, 1) | 1 | No | Dense |
| Mixed_5d.branch5x5_1.conv (1) | (288, 35, 35) | (48, 35, 35) | (1, 1) | (1, 1) | (0, 0) | (1, 1) | 1 | No | Dense |
| Mixed_6a.branch3x3.conv (1) | (288, 35, 35) | (384, 17, 17) | (3, 3) | (2, 2) | (0, 0) | (1, 1) | 1 | No | General |
| Mixed_6a.branch3x3dbl_3.conv (1) | (96, 35, 35) | (96, 17, 17) | (3, 3) | (2, 2) | (0, 0) | (1, 1) | 1 | No | General |
| Mixed_6b.branch1x1.conv (12) | (768, 17, 17) | (192, 17, 17) | (1, 1) | (1, 1) | (0, 0) | (1, 1) | 1 | No | Dense |
| Mixed_6b.branch7x7_1.conv (2) | (768, 17, 17) | (128, 17, 17) | (1, 1) | (1, 1) | (0, 0) | (1, 1) | 1 | No | Dense |
| Mixed_6b.branch7x7_2.conv (2) | (128, 17, 17) | (128, 17, 17) | (1, 7) | (1, 1) | (0, 3) | (1, 1) | 1 | No | Dense mix |
| Mixed_6b.branch7x7_3.conv (1) | (128, 17, 17) | (192, 17, 17) | (7, 1) | (1, 1) | (3, 0) | (1, 1) | 1 | No | Dense mix |
| Mixed_6b.branch7x7dbl_2.conv (2) | (128, 17, 17) | (128, 17, 17) | (7, 1) | (1, 1) | (3, 0) | (1, 1) | 1 | No | Dense mix |
| Mixed_6b.branch7x7dbl_5.conv (1) | (128, 17, 17) | (192, 17, 17) | (1, 7) | (1, 1) | (0, 3) | (1, 1) | 1 | No | Dense mix |
| Mixed_6c.branch7x7_1.conv (4) | (768, 17, 17) | (160, 17, 17) | (1, 1) | (1, 1) | (0, 0) | (1, 1) | 1 | No | Dense |
| Mixed_6c.branch7x7_2.conv (4) | (160, 17, 17) | (160, 17, 17) | (1, 7) | (1, 1) | (0, 3) | (1, 1) | 1 | No | Dense mix |
| Mixed_6c.branch7x7_3.conv (2) | (160, 17, 17) | (192, 17, 17) | (7, 1) | (1, 1) | (3, 0) | (1, 1) | 1 | No | Dense mix |
| Mixed_6c.branch7x7dbl_2.conv (4) | (160, 17, 17) | (160, 17, 17) | (7, 1) | (1, 1) | (3, 0) | (1, 1) | 1 | No | Dense mix |
| Mixed_6c.branch7x7dbl_5.conv (2) | (160, 17, 17) | (192, 17, 17) | (1, 7) | (1, 1) | (0, 3) | (1, 1) | 1 | No | Dense mix |
| Mixed_6e.branch7x7_2.conv (4) | (192, 17, 17) | (192, 17, 17) | (1, 7) | (1, 1) | (0, 3) | (1, 1) | 1 | No | Dense mix |
| Mixed_6e.branch7x7_3.conv (4) | (192, 17, 17) | (192, 17, 17) | (7, 1) | (1, 1) | (3, 0) | (1, 1) | 1 | No | Dense mix |
| AuxLogits.conv0.conv (1) | (768, 5, 5) | (128, 5, 5) | (1, 1) | (1, 1) | (0, 0) | (1, 1) | 1 | No | Dense |
| AuxLogits.conv1.conv (1) | (128, 5, 5) | (768, 1, 1) | (5, 5) | (1, 1) | (0, 0) | (1, 1) | 1 | No | General |
| Mixed_7a.branch3x3_2.conv (1) | (192, 17, 17) | (320, 8, 8) | (3, 3) | (2, 2) | (0, 0) | (1, 1) | 1 | No | General |
| Mixed_7a.branch7x7x3_4.conv (1) | (192, 17, 17) | (192, 8, 8) | (3, 3) | (2, 2) | (0, 0) | (1, 1) | 1 | No | General |
| Mixed_7b.branch1x1.conv (1) | (1280, 8, 8) | (320, 8, 8) | (1, 1) | (1, 1) | (0, 0) | (1, 1) | 1 | No | Dense |
| Mixed_7b.branch3x3_1.conv (1) | (1280, 8, 8) | (384, 8, 8) | (1, 1) | (1, 1) | (0, 0) | (1, 1) | 1 | No | Dense |
| Mixed_7b.branch3x3_2a.conv (4) | (384, 8, 8) | (384, 8, 8) | (1, 3) | (1, 1) | (0, 1) | (1, 1) | 1 | No | Dense mix |
| Mixed_7b.branch3x3_2b.conv (4) | (384, 8, 8) | (384, 8, 8) | (3, 1) | (1, 1) | (1, 0) | (1, 1) | 1 | No | Dense mix |
| Mixed_7b.branch3x3dbl_1.conv (1) | (1280, 8, 8) | (448, 8, 8) | (1, 1) | (1, 1) | (0, 0) | (1, 1) | 1 | No | Dense |
| Mixed_7b.branch3x3dbl_2.conv (2) | (448, 8, 8) | (384, 8, 8) | (3, 3) | (1, 1) | (1, 1) | (1, 1) | 1 | No | General |
| Mixed_7b.branch_pool.conv (1) | (1280, 8, 8) | (192, 8, 8) | (1, 1) | (1, 1) | (0, 0) | (1, 1) | 1 | No | Dense |
| Mixed_7c.branch1x1.conv (1) | (2048, 8, 8) | (320, 8, 8) | (1, 1) | (1, 1) | (0, 0) | (1, 1) | 1 | No | Dense |
| Mixed_7c.branch3x3_1.conv (1) | (2048, 8, 8) | (384, 8, 8) | (1, 1) | (1, 1) | (0, 0) | (1, 1) | 1 | No | Dense |
| Mixed_7c.branch3x3dbl_1.conv (1) | (2048, 8, 8) | (448, 8, 8) | (1, 1) | (1, 1) | (0, 0) | (1, 1) | 1 | No | Dense |
| Mixed_7c.branch_pool.conv (1) | (2048, 8, 8) | (192, 8, 8) | (1, 1) | (1, 1) | (0, 0) | (1, 1) | 1 | No | Dense |

(i) MobileNetV2, ImageNet (3, 256, 256)

| Name (count) | Input shape | Output shape | Kernel | Stride | Padding | Dilation | Groups | Bias | Type |
|---|---|---|---|---|---|---|---|---|---|
| features.0.0 (1) | (3, 256, 256) | (32, 128, 128) | (3, 3) | (2, 2) | (1, 1) | (1, 1) | 1 | No | General |
| features.1.conv.0.0 (1) | (32, 128, 128) | (32, 128, 128) | (3, 3) | (1, 1) | (1, 1) | (1, 1) | 32 | No | General |
| features.1.conv.1 (1) | (32, 128, 128) | (16, 128, 128) | (1, 1) | (1, 1) | (0, 0) | (1, 1) | 1 | No | Dense |
| features.2.conv.0.0 (1) | (16, 128, 128) | (96, 128, 128) | (1, 1) | (1, 1) | (0, 0) | (1, 1) | 1 | No | Dense |
| features.2.conv.1.0 (1) | (96, 128, 128) | (96, 64, 64) | (3, 3) | (2, 2) | (1, 1) | (1, 1) | 96 | No | General |
| features.2.conv.2 (1) | (96, 64, 64) | (24, 64, 64) | (1, 1) | (1, 1) | (0, 0) | (1, 1) | 1 | No | Dense |
| features.3.conv.0.0 (2) | (24, 64, 64) | (144, 64, 64) | (1, 1) | (1, 1) | (0, 0) | (1, 1) | 1 | No | Dense |
| features.3.conv.1.0 (1) | (144, 64, 64) | (144, 64, 64) | (3, 3) | (1, 1) | (1, 1) | (1, 1) | 144 | No | General |
| features.3.conv.2 (1) | (144, 64, 64) | (24, 64, 64) | (1, 1) | (1, 1) | (0, 0) | (1, 1) | 1 | No | Dense |
| features.4.conv.1.0 (1) | (144, 64, 64) | (144, 32, 32) | (3, 3) | (2, 2) | (1, 1) | (1, 1) | 144 | No | General |
| features.4.conv.2 (1) | (144, 32, 32) | (32, 32, 32) | (1, 1) | (1, 1) | (0, 0) | (1, 1) | 1 | No | Dense |
| features.5.conv.0.0 (3) | (32, 32, 32) | (192, 32, 32) | (1, 1) | (1, 1) | (0, 0) | (1, 1) | 1 | No | Dense |
| features.5.conv.1.0 (2) | (192, 32, 32) | (192, 32, 32) | (3, 3) | (1, 1) | (1, 1) | (1, 1) | 192 | No | General |
| features.5.conv.2 (2) | (192, 32, 32) | (32, 32, 32) | (1, 1) | (1, 1) | (0, 0) | (1, 1) | 1 | No | Dense |
| features.7.conv.1.0 (1) | (192, 32, 32) | (192, 16, 16) | (3, 3) | (2, 2) | (1, 1) | (1, 1) | 192 | No | General |
| features.7.conv.2 (1) | (192, 16, 16) | (64, 16, 16) | (1, 1) | (1, 1) | (0, 0) | (1, 1) | 1 | No | Dense |
| features.8.conv.0.0 (4) | (64, 16, 16) | (384, 16, 16) | (1, 1) | (1, 1) | (0, 0) | (1, 1) | 1 | No | Dense |
| features.8.conv.1.0 (4) | (384, 16, 16) | (384, 16, 16) | (3, 3) | (1, 1) | (1, 1) | (1, 1) | 384 | No | General |
| features.8.conv.2 (3) | (384, 16, 16) | (64, 16, 16) | (1, 1) | (1, 1) | (0, 0) | (1, 1) | 1 | No | Dense |
| features.11.conv.2 (1) | (384, 16, 16) | (96, 16, 16) | (1, 1) | (1, 1) | (0, 0) | (1, 1) | 1 | No | Dense |
| features.12.conv.0.0 (3) | (96, 16, 16) | (576, 16, 16) | (1, 1) | (1, 1) | (0, 0) | (1, 1) | 1 | No | Dense |
| features.12.conv.1.0 (2) | (576, 16, 16) | (576, 16, 16) | (3, 3) | (1, 1) | (1, 1) | (1, 1) | 576 | No | General |
| features.12.conv.2 (2) | (576, 16, 16) | (96, 16, 16) | (1, 1) | (1, 1) | (0, 0) | (1, 1) | 1 | No | Dense |
| features.14.conv.1.0 (1) | (576, 16, 16) | (576, 8, 8) | (3, 3) | (2, 2) | (1, 1) | (1, 1) | 576 | No | General |
| features.14.conv.2 (1) | (576, 8, 8) | (160, 8, 8) | (1, 1) | (1, 1) | (0, 0) | (1, 1) | 1 | No | Dense |
| features.15.conv.0.0 (3) | (160, 8, 8) | (960, 8, 8) | (1, 1) | (1, 1) | (0, 0) | (1, 1) | 1 | No | Dense |
| features.15.conv.1.0 (3) | (960, 8, 8) | (960, 8, 8) | (3, 3) | (1, 1) | (1, 1) | (1, 1) | 960 | No | General |
| features.15.conv.2 (2) | (960, 8, 8) | (160, 8, 8) | (1, 1) | (1, 1) | (0, 0) | (1, 1) | 1 | No | Dense |
| features.17.conv.2 (1) | (960, 8, 8) | (320, 8, 8) | (1, 1) | (1, 1) | (0, 0) | (1, 1) | 1 | No | Dense |
| features.18.0 (1) | (320, 8, 8) | (1280, 8, 8) | (1, 1) | (1, 1) | (0, 0) | (1, 1) | 1 | No | Dense |

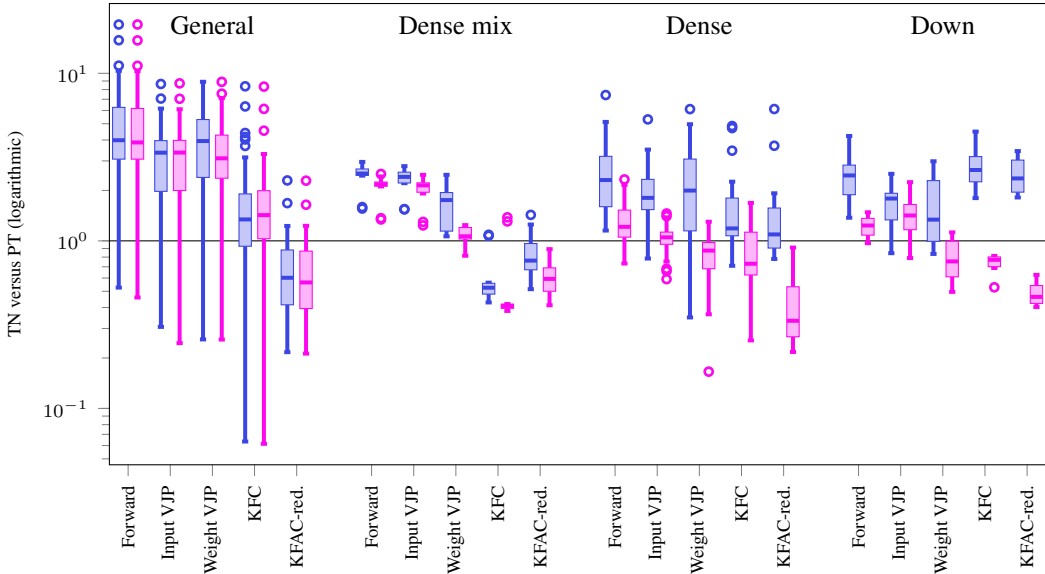

Figure F17: Benchmark overview. We measure the performance ratios of our TN implementation w.r.t. a base line in PyTorch (PT). Blue boxes show the performance ratios of TN versus PT, second-color boxes show the performance ratios of TN+opt versus PT.

# F   Run Time Evaluation Details (GPU)

Here we provide all details on the run time evaluation from the main text. We consider the convolutions from the CNNs from §E. Experiments were carried out on an Nvidia Tesla T4 (16 GB memory). We use a batch size of 32 for the ImageNet architectures, and 128 for the others.

## F.1   Protocol & Overview

We compare different implementations of the same operations in PyTorch. The base line (referenced by 'PT') uses PyTorch's built-in functionalities for convolutions and related operations, such as `torch.nn.functional.conv2d` (forward), `torch.nn.functional.unfold` (KFC, KFAC-reduce), and PyTorch's built-in automatic differentiation `torch.autograd.grad` (VJPs).

Our TN implementation (referenced by 'TN') sets up operands and the string-valued equation for each routine. Optionally, we can apply the simplifications from §4 as a post-processing step before contraction, which yields a modified equation and operand list ('TN + opt'). Finally, we determine the contraction path using `opt_einsum.contract_path` and perform the contraction with its PyTorch back-end (`opt_einsum.contract`). We only measure the contraction time as in practical settings, the contraction path search would be executed once, then cached. We also exclude final operations to obtain the correct shape or scale (flattening, reshaping, scaling by constant) in all implementations (including the base line).

For each operation and each convolution layer, we perform 50 independent repetitions and report the minimum time in tables. To summarize those tables, we extract the performance ratios, that is the TN implementation's run time divided by the base line's. Ratios larger than 1 mean that the TN implementation is slower, ratios smaller than 1 indicate that it is faster than the base line. We collect those ratios for the different convolution types (general, mixed dense, dense, sub-sampling) and display them separately using box plots. Each operation has two boxes, corresponding to the un-simplified (TN), and the simplified (TN + opt) implementation. For the box plots, we use `matplotlib`'s default settings (a box extends from the first quartile to the third quartile of the data, with a line at the median; whiskers extend from the box by 1.5x the inter-quartile range; flier points are those past the end of the whiskers). Figure F17 summarizes the entire GPU benchmark. Figure F18 shows the same information with each convolution type as an individual plot.

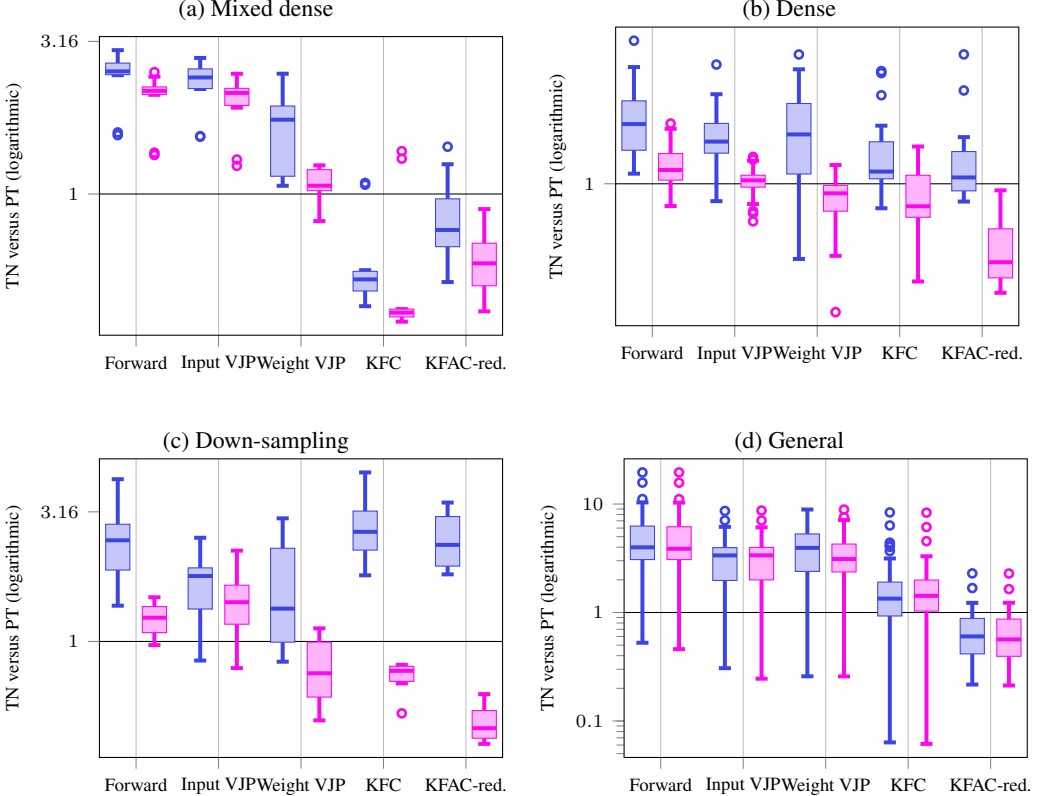

Figure F18: Impact of TN simplifications (non-simplified performance ratios shown in blue). TN simplifications improve performance on (a) mixed dense, (b) dense, and (c) down-sampling convolutions. (d) General convolutions are not affected by TN simplifications.

## F.2 Forward Pass

We compare TN and TN+opt with PyTorch's `torch.nn.functional.conv2d`. Figure F19 visualizes the performance ratios for different convolution categories. Table F5 contains the detailed run times and performance factors.

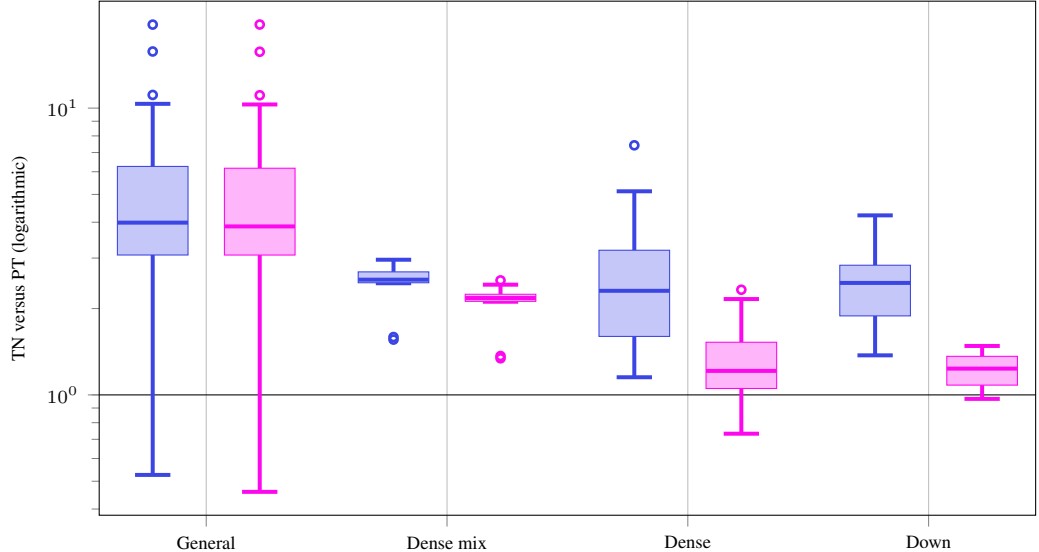

Figure F19: Forward pass performance ratios of TN versus PT and TN+opt versus PT for different convolution types on GPU.

Table F5: Forward pass performance comparison on GPU.

**(a) 3c3d, CIFAR-10, input shape (128, 3, 32, 32)**

| Name | TN [s] | PT [s] | Factor | TN + opt [s] | PT [s] | Factor |
|---|---|---|---|---|---|---|
| conv1.0 | $1.26 \cdot 10^{-3}$ | $4.63 \cdot 10^{-4}$ | 2.73 x | $1.23 \cdot 10^{-3}$ | $4.64 \cdot 10^{-4}$ | 2.64 x |
| conv2.0 | $1.91 \cdot 10^{-3}$ | $4.52 \cdot 10^{-4}$ | 4.22 x | $1.79 \cdot 10^{-3}$ | $4.53 \cdot 10^{-4}$ | 3.95 x |
| conv3.1 | $1.21 \cdot 10^{-3}$ | $4.11 \cdot 10^{-4}$ | 2.94 x | $1.16 \cdot 10^{-3}$ | $4.10 \cdot 10^{-4}$ | 2.83 x |

**(b) F-MNIST 2c2d, input shape (128, 1, 28, 28)**

| Name | TN [s] | PT [s] | Factor | TN + opt [s] | PT [s] | Factor |
|---|---|---|---|---|---|---|
| conv1.1 | $8.21 \cdot 10^{-4}$ | $2.25 \cdot 10^{-4}$ | 3.65 x | $7.67 \cdot 10^{-4}$ | $2.25 \cdot 10^{-4}$ | 3.41 x |
| conv2.1 | $3.56 \cdot 10^{-3}$ | $7.43 \cdot 10^{-4}$ | 4.79 x | $3.24 \cdot 10^{-3}$ | $7.83 \cdot 10^{-4}$ | 4.14 x |

**(c) CIFAR-100 All-CNN-C, input shape (128, 3, 32, 32)**

| Name | TN [s] | PT [s] | Factor | TN + opt [s] | PT [s] | Factor |
|---|---|---|---|---|---|---|
| conv1.1 | $1.01 \cdot 10^{-3}$ | $4.20 \cdot 10^{-4}$ | 2.41 x | $9.45 \cdot 10^{-4}$ | $4.19 \cdot 10^{-4}$ | 2.25 x |
| conv2.1 | $1.94 \cdot 10^{-2}$ | $3.09 \cdot 10^{-3}$ | 6.26 x | $1.88 \cdot 10^{-2}$ | $3.10 \cdot 10^{-3}$ | 6.08 x |
| conv3.1 | $8.56 \cdot 10^{-3}$ | $2.86 \cdot 10^{-3}$ | 3.00 x | $7.77 \cdot 10^{-3}$ | $2.86 \cdot 10^{-3}$ | 2.72 x |
| conv4.1 | $8.58 \cdot 10^{-3}$ | $1.75 \cdot 10^{-3}$ | 4.91 x | $7.77 \cdot 10^{-3}$ | $1.75 \cdot 10^{-3}$ | 4.45 x |
| conv5.1 | $1.67 \cdot 10^{-2}$ | $2.91 \cdot 10^{-3}$ | 5.74 x | $1.51 \cdot 10^{-2}$ | $2.91 \cdot 10^{-3}$ | 5.19 x |
| conv6.1 | $5.13 \cdot 10^{-3}$ | $2.24 \cdot 10^{-3}$ | 2.29 x | $5.08 \cdot 10^{-3}$ | $2.24 \cdot 10^{-3}$ | 2.27 x |
| conv7.0 | $2.58 \cdot 10^{-3}$ | $8.26 \cdot 10^{-4}$ | 3.12 x | $2.51 \cdot 10^{-3}$ | $8.27 \cdot 10^{-4}$ | 3.03 x |
| conv8.1 | $8.20 \cdot 10^{-4}$ | $2.96 \cdot 10^{-4}$ | 2.77 x | $3.42 \cdot 10^{-4}$ | $2.97 \cdot 10^{-4}$ | 1.15 x |
| conv9.1 | $7.52 \cdot 10^{-4}$ | $2.35 \cdot 10^{-4}$ | 3.19 x | $3.01 \cdot 10^{-4}$ | $2.35 \cdot 10^{-4}$ | 1.28 x |

**(d) Alexnet, input shape (32, 3, 256, 256)**

| Name | TN [s] | PT [s] | Factor | TN + opt [s] | PT [s] | Factor |
|---|---|---|---|---|---|---|
| features.0 | $1.83 \cdot 10^{-2}$ | $2.45 \cdot 10^{-3}$ | 7.47 x | $1.79 \cdot 10^{-2}$ | $2.44 \cdot 10^{-3}$ | 7.35 x |
| features.3 | $7.43 \cdot 10^{-3}$ | $2.67 \cdot 10^{-3}$ | 2.79 x | $7.27 \cdot 10^{-3}$ | $2.85 \cdot 10^{-3}$ | 2.55 x |
| features.6 | $4.68 \cdot 10^{-3}$ | $1.04 \cdot 10^{-3}$ | 4.52 x | $3.22 \cdot 10^{-3}$ | $1.02 \cdot 10^{-3}$ | 3.14 x |
| features.8 | $6.15 \cdot 10^{-3}$ | $1.86 \cdot 10^{-3}$ | 3.31 x | $6.16 \cdot 10^{-3}$ | $1.84 \cdot 10^{-3}$ | 3.34 x |
| features.10 | $4.41 \cdot 10^{-3}$ | $1.31 \cdot 10^{-3}$ | 3.36 x | $4.38 \cdot 10^{-3}$ | $1.31 \cdot 10^{-3}$ | 3.35 x |

**(e) ResNet18, input shape (32, 3, 256, 256)**

| Name | TN [s] | PT [s] | Factor | TN + opt [s] | PT [s] | Factor |
|---|---|---|---|---|---|---|
| conv1 | $1.44 \cdot 10^{-2}$ | $4.07 \cdot 10^{-3}$ | 3.53 x | $1.44 \cdot 10^{-2}$ | $4.08 \cdot 10^{-3}$ | 3.53 x |
| layer1.0.conv1 | $1.05 \cdot 10^{-2}$ | $1.78 \cdot 10^{-3}$ | 5.91 x | $1.05 \cdot 10^{-2}$ | $1.79 \cdot 10^{-3}$ | 5.87 x |
| layer2.0.conv1 | $6.44 \cdot 10^{-3}$ | $1.89 \cdot 10^{-3}$ | 3.41 x | $6.46 \cdot 10^{-3}$ | $1.89 \cdot 10^{-3}$ | 3.42 x |
| layer2.0.conv2 | $6.88 \cdot 10^{-3}$ | $1.51 \cdot 10^{-3}$ | 4.54 x | $6.91 \cdot 10^{-3}$ | $1.52 \cdot 10^{-3}$ | 4.54 x |
| layer2.0.downsample.0 | $1.60 \cdot 10^{-3}$ | $3.79 \cdot 10^{-4}$ | 4.23 x | $5.19 \cdot 10^{-4}$ | $3.80 \cdot 10^{-4}$ | 1.37 x |
| layer3.0.conv1 | $3.82 \cdot 10^{-3}$ | $2.00 \cdot 10^{-3}$ | 1.91 x | $3.56 \cdot 10^{-3}$ | $2.01 \cdot 10^{-3}$ | 1.77 x |
| layer3.0.conv2 | $5.02 \cdot 10^{-3}$ | $1.30 \cdot 10^{-3}$ | 3.85 x | $5.05 \cdot 10^{-3}$ | $1.31 \cdot 10^{-3}$ | 3.87 x |
| layer3.0.downsample.0 | $1.10 \cdot 10^{-3}$ | $3.78 \cdot 10^{-4}$ | 2.91 x | $5.61 \cdot 10^{-4}$ | $3.79 \cdot 10^{-4}$ | 1.48 x |
| layer4.0.conv1 | $2.87 \cdot 10^{-3}$ | $2.36 \cdot 10^{-3}$ | 1.21 x | $2.86 \cdot 10^{-3}$ | $2.36 \cdot 10^{-3}$ | 1.21 x |
| layer4.0.conv2 | $4.47 \cdot 10^{-3}$ | $1.40 \cdot 10^{-3}$ | 3.18 x | $4.51 \cdot 10^{-3}$ | $1.40 \cdot 10^{-3}$ | 3.21 x |
| layer4.0.downsample.0 | $9.90 \cdot 10^{-4}$ | $3.81 \cdot 10^{-4}$ | 2.60 x | $5.16 \cdot 10^{-4}$ | $3.83 \cdot 10^{-4}$ | 1.35 x |

**(f) ResNext101, input shape (32, 3, 256, 256)**

| Name | TN [s] | PT [s] | Factor | TN + opt [s] | PT [s] | Factor |
|---|---|---|---|---|---|---|
| conv1 | $1.45 \cdot 10^{-2}$ | $4.07 \cdot 10^{-3}$ | 3.57 x | $1.44 \cdot 10^{-2}$ | $4.07 \cdot 10^{-3}$ | 3.54 x |
| layer1.0.conv1 | $4.31 \cdot 10^{-3}$ | $1.22 \cdot 10^{-3}$ | 3.54 x | $2.26 \cdot 10^{-3}$ | $1.22 \cdot 10^{-3}$ | 1.85 x |
| layer1.0.conv2 | $3.03 \cdot 10^{-2}$ | $9.86 \cdot 10^{-3}$ | 3.07 x | $3.03 \cdot 10^{-2}$ | $9.86 \cdot 10^{-3}$ | 3.08 x |
| layer1.0.conv3 | $1.51 \cdot 10^{-2}$ | $6.54 \cdot 10^{-3}$ | 2.31 x | $7.49 \cdot 10^{-3}$ | $6.54 \cdot 10^{-3}$ | 1.15 x |
| layer2.0.conv1 | $2.08 \cdot 10^{-2}$ | $1.29 \cdot 10^{-2}$ | 1.61 x | $1.36 \cdot 10^{-2}$ | $1.29 \cdot 10^{-2}$ | 1.05 x |
| layer2.0.conv2 | $3.33 \cdot 10^{-2}$ | $4.93 \cdot 10^{-3}$ | 6.75 x | $3.33 \cdot 10^{-2}$ | $4.93 \cdot 10^{-3}$ | 6.75 x |
| layer2.0.conv3 | $1.05 \cdot 10^{-2}$ | $6.24 \cdot 10^{-3}$ | 1.69 x | $6.84 \cdot 10^{-3}$ | $6.24 \cdot 10^{-3}$ | 1.10 x |
| layer2.0.downsample.0 | $7.65 \cdot 10^{-3}$ | $3.30 \cdot 10^{-3}$ | 2.31 x | $3.71 \cdot 10^{-3}$ | $3.31 \cdot 10^{-3}$ | 1.12 x |
| layer2.1.conv2 | $1.50 \cdot 10^{-2}$ | $4.59 \cdot 10^{-3}$ | 3.27 x | $1.50 \cdot 10^{-2}$ | $4.59 \cdot 10^{-3}$ | 3.27 x |
| layer3.0.conv1 | $1.67 \cdot 10^{-2}$ | $1.23 \cdot 10^{-2}$ | 1.35 x | $1.28 \cdot 10^{-2}$ | $1.23 \cdot 10^{-2}$ | 1.04 x |
| layer3.0.conv2 | $1.76 \cdot 10^{-2}$ | $2.65 \cdot 10^{-3}$ | 6.65 x | $1.76 \cdot 10^{-2}$ | $2.66 \cdot 10^{-3}$ | 6.65 x |
| layer3.0.conv3 | $8.27 \cdot 10^{-3}$ | $6.14 \cdot 10^{-3}$ | 1.35 x | $6.44 \cdot 10^{-3}$ | $6.14 \cdot 10^{-3}$ | 1.05 x |
| layer3.0.downsample.0 | $5.58 \cdot 10^{-3}$ | $3.20 \cdot 10^{-3}$ | 1.74 x | $3.42 \cdot 10^{-3}$ | $3.20 \cdot 10^{-3}$ | 1.07 x |
| layer3.1.conv2 | $7.64 \cdot 10^{-3}$ | $2.49 \cdot 10^{-3}$ | 3.07 x | $7.64 \cdot 10^{-3}$ | $2.48 \cdot 10^{-3}$ | 3.07 x |
| layer4.0.conv1 | $1.43 \cdot 10^{-2}$ | $1.22 \cdot 10^{-2}$ | 1.18 x | $1.24 \cdot 10^{-2}$ | $1.22 \cdot 10^{-2}$ | 1.02 x |
| layer4.0.conv2 | $8.07 \cdot 10^{-3}$ | $2.02 \cdot 10^{-3}$ | 3.99 x | $8.08 \cdot 10^{-3}$ | $2.02 \cdot 10^{-3}$ | 4.00 x |
| layer4.0.conv3 | $7.85 \cdot 10^{-3}$ | $6.28 \cdot 10^{-3}$ | 1.25 x | $6.33 \cdot 10^{-3}$ | $6.28 \cdot 10^{-3}$ | 1.01 x |
| layer4.0.downsample.0 | $4.73 \cdot 10^{-3}$ | $3.44 \cdot 10^{-3}$ | 1.37 x | $3.34 \cdot 10^{-3}$ | $3.44 \cdot 10^{-3}$ | **0.97 x** |
| layer4.1.conv2 | $4.76 \cdot 10^{-3}$ | $1.36 \cdot 10^{-3}$ | 3.51 x | $4.77 \cdot 10^{-3}$ | $1.35 \cdot 10^{-3}$ | 3.52 x |

**(g) ConvNeXt-base, input shape (32, 3, 256, 256)**

| Name | TN [s] | PT [s] | Factor | TN + opt [s] | PT [s] | Factor |
|---|---|---|---|---|---|---|
| features.0.0 | $4.26 \cdot 10^{-3}$ | $9.88 \cdot 10^{-4}$ | 4.31 x | $1.20 \cdot 10^{-3}$ | $9.94 \cdot 10^{-4}$ | 1.21 x |
| features.1.0.block.0 | $5.07 \cdot 10^{-2}$ | $7.61 \cdot 10^{-3}$ | 6.66 x | $5.07 \cdot 10^{-2}$ | $7.61 \cdot 10^{-3}$ | 6.66 x |
| features.2.1 | $7.60 \cdot 10^{-3}$ | $3.21 \cdot 10^{-3}$ | 2.37 x | $3.89 \cdot 10^{-3}$ | $3.20 \cdot 10^{-3}$ | 1.21 x |
| features.3.0.block.0 | $2.36 \cdot 10^{-2}$ | $3.81 \cdot 10^{-3}$ | 6.18 x | $2.35 \cdot 10^{-2}$ | $3.81 \cdot 10^{-3}$ | 6.17 x |
| features.4.1 | $5.41 \cdot 10^{-3}$ | $3.38 \cdot 10^{-3}$ | 1.60 x | $3.52 \cdot 10^{-3}$ | $3.38 \cdot 10^{-3}$ | 1.04 x |
| features.5.0.block.0 | $1.11 \cdot 10^{-2}$ | $1.94 \cdot 10^{-3}$ | 5.70 x | $1.10 \cdot 10^{-2}$ | $1.94 \cdot 10^{-3}$ | 5.69 x |
| features.6.1 | $4.54 \cdot 10^{-3}$ | $3.69 \cdot 10^{-3}$ | 1.23 x | $3.44 \cdot 10^{-3}$ | $3.70 \cdot 10^{-3}$ | **0.93 x** |
| features.7.0.block.0 | $1.06 \cdot 10^{-3}$ | $1.01 \cdot 10^{-3}$ | 1.05 x | $1.02 \cdot 10^{-3}$ | $1.01 \cdot 10^{-3}$ | 1.01 x |

## (h) InceptionV3, input shape (32, 3, 299, 299)

| Name | TN [s] | PT [s] | Factor | TN + opt [s] | PT [s] | Factor |
|---|---|---|---|---|---|---|
| Conv2d_1a_3x3.conv | $1.02 \cdot 10^{-2}$ | $9.85 \cdot 10^{-4}$ | 10.35 x | $1.01 \cdot 10^{-2}$ | $9.79 \cdot 10^{-4}$ | 10.30 x |
| Conv2d_2a_3x3.conv | $3.23 \cdot 10^{-2}$ | $5.14 \cdot 10^{-3}$ | 6.30 x | $3.19 \cdot 10^{-2}$ | $5.16 \cdot 10^{-3}$ | 6.18 x |
| Conv2d_2b_3x3.conv | $4.83 \cdot 10^{-2}$ | $8.14 \cdot 10^{-3}$ | 5.93 x | $4.78 \cdot 10^{-2}$ | $8.14 \cdot 10^{-3}$ | 5.87 x |
| Conv2d_3b_1x1.conv | $4.96 \cdot 10^{-3}$ | $1.17 \cdot 10^{-3}$ | 4.24 x | $1.72 \cdot 10^{-3}$ | $1.17 \cdot 10^{-3}$ | 1.48 x |
| Conv2d_4a_3x3.conv | $3.69 \cdot 10^{-2}$ | $7.64 \cdot 10^{-3}$ | 4.83 x | $3.65 \cdot 10^{-2}$ | $7.64 \cdot 10^{-3}$ | 4.77 x |
| Mixed_5b.branch1x1.conv | $1.85 \cdot 10^{-3}$ | $5.04 \cdot 10^{-4}$ | 3.68 x | $8.17 \cdot 10^{-4}$ | $5.03 \cdot 10^{-4}$ | 1.62 x |
| Mixed_5b.branch5x5_1.conv | $1.64 \cdot 10^{-3}$ | $4.97 \cdot 10^{-4}$ | 3.30 x | $8.11 \cdot 10^{-4}$ | $4.99 \cdot 10^{-4}$ | 1.63 x |
| Mixed_5b.branch5x5_2.conv | $5.01 \cdot 10^{-3}$ | $1.23 \cdot 10^{-3}$ | 4.07 x | $4.83 \cdot 10^{-3}$ | $1.23 \cdot 10^{-3}$ | 3.94 x |
| Mixed_5b.branch3x3dbl_2.conv | $4.40 \cdot 10^{-3}$ | $1.31 \cdot 10^{-3}$ | 3.38 x | $4.31 \cdot 10^{-3}$ | $1.31 \cdot 10^{-3}$ | 3.30 x |
| Mixed_5b.branch3x3dbl_3.conv | $5.82 \cdot 10^{-3}$ | $1.66 \cdot 10^{-3}$ | 3.50 x | $5.66 \cdot 10^{-3}$ | $1.66 \cdot 10^{-3}$ | 3.40 x |
| Mixed_5b.branch_pool.conv | $1.33 \cdot 10^{-3}$ | $3.26 \cdot 10^{-4}$ | 4.09 x | $7.04 \cdot 10^{-4}$ | $3.27 \cdot 10^{-4}$ | 2.15 x |
| Mixed_5c.branch1x1.conv | $2.08 \cdot 10^{-3}$ | $6.41 \cdot 10^{-4}$ | 3.24 x | $1.03 \cdot 10^{-3}$ | $6.40 \cdot 10^{-4}$ | 1.61 x |
| Mixed_5c.branch5x5_1.conv | $1.87 \cdot 10^{-3}$ | $6.29 \cdot 10^{-4}$ | 2.97 x | $1.03 \cdot 10^{-3}$ | $6.30 \cdot 10^{-4}$ | 1.63 x |
| Mixed_5d.branch1x1.conv | $2.18 \cdot 10^{-3}$ | $6.99 \cdot 10^{-4}$ | 3.12 x | $1.13 \cdot 10^{-3}$ | $6.98 \cdot 10^{-4}$ | 1.62 x |
| Mixed_5d.branch5x5_1.conv | $1.96 \cdot 10^{-3}$ | $6.91 \cdot 10^{-4}$ | 2.84 x | $1.13 \cdot 10^{-3}$ | $6.88 \cdot 10^{-4}$ | 1.64 x |
| Mixed_6a.branch3x3.conv | $1.15 \cdot 10^{-2}$ | $7.12 \cdot 10^{-3}$ | 1.61 x | $1.07 \cdot 10^{-2}$ | $7.13 \cdot 10^{-3}$ | 1.51 x |
| Mixed_6a.branch3x3dbl_3.conv | $2.61 \cdot 10^{-3}$ | $8.99 \cdot 10^{-4}$ | 2.90 x | $2.36 \cdot 10^{-3}$ | $9.00 \cdot 10^{-4}$ | 2.62 x |
| Mixed_6b.branch1x1.conv | $2.16 \cdot 10^{-3}$ | $1.22 \cdot 10^{-3}$ | 1.77 x | $1.41 \cdot 10^{-3}$ | $1.22 \cdot 10^{-3}$ | 1.15 x |
| Mixed_6b.branch7x7_1.conv | $1.67 \cdot 10^{-3}$ | $8.15 \cdot 10^{-4}$ | 2.05 x | $1.10 \cdot 10^{-3}$ | $8.16 \cdot 10^{-4}$ | 1.35 x |
| Mixed_6b.branch7x7_2.conv | $2.14 \cdot 10^{-3}$ | $8.04 \cdot 10^{-4}$ | 2.66 x | $1.76 \cdot 10^{-3}$ | $8.05 \cdot 10^{-4}$ | 2.19 x |
| Mixed_6b.branch7x7_3.conv | $2.59 \cdot 10^{-3}$ | $1.06 \cdot 10^{-3}$ | 2.45 x | $2.27 \cdot 10^{-3}$ | $1.06 \cdot 10^{-3}$ | 2.15 x |
| Mixed_6b.branch7x7dbl_2.conv | $2.17 \cdot 10^{-3}$ | $7.88 \cdot 10^{-4}$ | 2.76 x | $1.78 \cdot 10^{-3}$ | $7.88 \cdot 10^{-4}$ | 2.26 x |
| Mixed_6b.branch7x7dbl_5.conv | $2.63 \cdot 10^{-3}$ | $1.07 \cdot 10^{-3}$ | 2.46 x | $2.25 \cdot 10^{-3}$ | $1.07 \cdot 10^{-3}$ | 2.11 x |
| Mixed_6c.branch7x7_1.conv | $2.05 \cdot 10^{-3}$ | $1.16 \cdot 10^{-3}$ | 1.77 x | $1.41 \cdot 10^{-3}$ | $1.16 \cdot 10^{-3}$ | 1.21 x |
| Mixed_6c.branch7x7_2.conv | $3.19 \cdot 10^{-3}$ | $1.12 \cdot 10^{-3}$ | 2.84 x | $2.72 \cdot 10^{-3}$ | $1.12 \cdot 10^{-3}$ | 2.42 x |
| Mixed_6c.branch7x7_3.conv | $3.12 \cdot 10^{-3}$ | $1.25 \cdot 10^{-3}$ | 2.50 x | $2.76 \cdot 10^{-3}$ | $1.25 \cdot 10^{-3}$ | 2.21 x |
| Mixed_6c.branch7x7dbl_2.conv | $3.25 \cdot 10^{-3}$ | $1.10 \cdot 10^{-3}$ | 2.96 x | $2.75 \cdot 10^{-3}$ | $1.10 \cdot 10^{-3}$ | 2.51 x |
| Mixed_6c.branch7x7dbl_5.conv | $3.19 \cdot 10^{-3}$ | $1.28 \cdot 10^{-3}$ | 2.49 x | $2.73 \cdot 10^{-3}$ | $1.29 \cdot 10^{-3}$ | 2.12 x |
| Mixed_6e.branch7x7_2.conv | $3.78 \cdot 10^{-3}$ | $1.48 \cdot 10^{-3}$ | 2.54 x | $3.21 \cdot 10^{-3}$ | $1.48 \cdot 10^{-3}$ | 2.16 x |
| Mixed_6e.branch7x7_3.conv | $3.87 \cdot 10^{-3}$ | $1.45 \cdot 10^{-3}$ | 2.66 x | $3.26 \cdot 10^{-3}$ | $1.46 \cdot 10^{-3}$ | 2.24 x |
| AuxLogits.conv0.conv | $6.40 \cdot 10^{-4}$ | $2.38 \cdot 10^{-4}$ | 2.69 x | $3.20 \cdot 10^{-4}$ | $2.39 \cdot 10^{-4}$ | 1.34 x |
| AuxLogits.conv1.conv | $8.06 \cdot 10^{-4}$ | $1.53 \cdot 10^{-3}$ | **0.53 x** | $6.98 \cdot 10^{-4}$ | $1.52 \cdot 10^{-3}$ | **0.46 x** |
| Mixed_7a.branch3x3_2.conv | $1.08 \cdot 10^{-3}$ | $4.37 \cdot 10^{-4}$ | 2.48 x | $1.09 \cdot 10^{-3}$ | $5.01 \cdot 10^{-4}$ | 2.18 x |
| Mixed_7a.branch7x7x3_4.conv | $1.54 \cdot 10^{-3}$ | $8.89 \cdot 10^{-4}$ | 1.73 x | $1.52 \cdot 10^{-3}$ | $8.88 \cdot 10^{-4}$ | 1.71 x |
| Mixed_7b.branch1x1.conv | $1.29 \cdot 10^{-3}$ | $7.43 \cdot 10^{-4}$ | 1.73 x | $8.76 \cdot 10^{-4}$ | $7.43 \cdot 10^{-4}$ | 1.18 x |
| Mixed_7b.branch3x3_1.conv | $1.47 \cdot 10^{-3}$ | $1.03 \cdot 10^{-3}$ | 1.42 x | $1.02 \cdot 10^{-3}$ | $1.03 \cdot 10^{-3}$ | **0.99 x** |
| Mixed_7b.branch3x3_2a.conv | $1.49 \cdot 10^{-3}$ | $9.36 \cdot 10^{-4}$ | 1.59 x | $1.26 \cdot 10^{-3}$ | $9.38 \cdot 10^{-4}$ | 1.34 x |
| Mixed_7b.branch3x3_2b.conv | $1.46 \cdot 10^{-3}$ | $9.37 \cdot 10^{-4}$ | 1.56 x | $1.28 \cdot 10^{-3}$ | $9.37 \cdot 10^{-4}$ | 1.37 x |
| Mixed_7b.branch3x3dbl_1.conv | $1.67 \cdot 10^{-3}$ | $1.04 \cdot 10^{-3}$ | 1.61 x | $1.17 \cdot 10^{-3}$ | $1.04 \cdot 10^{-3}$ | 1.13 x |
| Mixed_7b.branch3x3dbl_2.conv | $3.18 \cdot 10^{-3}$ | $9.82 \cdot 10^{-4}$ | 3.23 x | $3.21 \cdot 10^{-3}$ | $9.83 \cdot 10^{-4}$ | 3.26 x |
| Mixed_7b.branch_pool.conv | $9.54 \cdot 10^{-4}$ | $6.76 \cdot 10^{-4}$ | 1.41 x | $6.30 \cdot 10^{-4}$ | $6.75 \cdot 10^{-4}$ | **0.93 x** |
| Mixed_7c.branch1x1.conv | $1.68 \cdot 10^{-3}$ | $1.08 \cdot 10^{-3}$ | 1.56 x | $1.27 \cdot 10^{-3}$ | $1.08 \cdot 10^{-3}$ | 1.18 x |
| Mixed_7c.branch3x3_1.conv | $1.98 \cdot 10^{-3}$ | $1.60 \cdot 10^{-3}$ | 1.23 x | $1.51 \cdot 10^{-3}$ | $1.60 \cdot 10^{-3}$ | **0.94 x** |
| Mixed_7c.branch3x3dbl_1.conv | $2.25 \cdot 10^{-3}$ | $1.56 \cdot 10^{-3}$ | 1.44 x | $1.73 \cdot 10^{-3}$ | $1.56 \cdot 10^{-3}$ | 1.11 x |
| Mixed_7c.branch_pool.conv | $1.25 \cdot 10^{-3}$ | $1.04 \cdot 10^{-3}$ | 1.20 x | $9.35 \cdot 10^{-4}$ | $1.04 \cdot 10^{-3}$ | **0.90 x** |

## (i) MobileNetV2, input shape (32, 3, 256, 256)

| Name | TN [s] | PT [s] | Factor | TN + opt [s] | PT [s] | Factor |
|---|---|---|---|---|---|---|
| features.0.0 | $6.91 \cdot 10^{-3}$ | $7.23 \cdot 10^{-4}$ | 9.55 x | $6.92 \cdot 10^{-3}$ | $7.24 \cdot 10^{-4}$ | 9.56 x |
| features.1.conv.0.0 | $2.28 \cdot 10^{-2}$ | $2.05 \cdot 10^{-3}$ | 11.11 x | $2.28 \cdot 10^{-2}$ | $2.06 \cdot 10^{-3}$ | 11.09 x |
| features.1.conv.1 | $5.64 \cdot 10^{-3}$ | $7.61 \cdot 10^{-4}$ | 7.42 x | $1.56 \cdot 10^{-3}$ | $7.57 \cdot 10^{-4}$ | 2.06 x |
| features.2.conv.0.0 | $4.27 \cdot 10^{-3}$ | $1.74 \cdot 10^{-3}$ | 2.45 x | $2.02 \cdot 10^{-3}$ | $1.74 \cdot 10^{-3}$ | 1.16 x |
| features.2.conv.1.0 | $3.31 \cdot 10^{-2}$ | $1.69 \cdot 10^{-3}$ | 19.55 x | $3.31 \cdot 10^{-2}$ | $1.69 \cdot 10^{-3}$ | 19.56 x |
| features.2.conv.2 | $2.53 \cdot 10^{-3}$ | $4.93 \cdot 10^{-4}$ | 5.13 x | $1.08 \cdot 10^{-3}$ | $4.99 \cdot 10^{-4}$ | 2.16 x |
| features.3.conv.0.0 | $1.78 \cdot 10^{-3}$ | $7.88 \cdot 10^{-4}$ | 2.26 x | $9.63 \cdot 10^{-4}$ | $7.88 \cdot 10^{-4}$ | 1.22 x |
| features.3.conv.1.0 | $2.09 \cdot 10^{-2}$ | $2.30 \cdot 10^{-3}$ | 9.07 x | $2.09 \cdot 10^{-2}$ | $2.30 \cdot 10^{-3}$ | 9.06 x |
| features.3.conv.2 | $2.93 \cdot 10^{-3}$ | $6.33 \cdot 10^{-4}$ | 4.63 x | $1.47 \cdot 10^{-3}$ | $6.34 \cdot 10^{-4}$ | 2.33 x |
| features.4.conv.1.0 | $1.04 \cdot 10^{-2}$ | $6.62 \cdot 10^{-4}$ | 15.76 x | $1.04 \cdot 10^{-2}$ | $6.63 \cdot 10^{-4}$ | 15.72 x |
| features.4.conv.2 | $1.10 \cdot 10^{-3}$ | $2.61 \cdot 10^{-4}$ | 4.23 x | $5.03 \cdot 10^{-4}$ | $2.61 \cdot 10^{-4}$ | 1.92 x |
| features.5.conv.0.0 | $9.24 \cdot 10^{-4}$ | $3.32 \cdot 10^{-4}$ | 2.78 x | $5.07 \cdot 10^{-4}$ | $3.33 \cdot 10^{-4}$ | 1.52 x |
| features.5.conv.1.0 | $5.44 \cdot 10^{-3}$ | $7.87 \cdot 10^{-4}$ | 6.91 x | $5.42 \cdot 10^{-3}$ | $7.88 \cdot 10^{-4}$ | 6.88 x |
| features.5.conv.2 | $1.22 \cdot 10^{-3}$ | $3.11 \cdot 10^{-4}$ | 3.93 x | $6.16 \cdot 10^{-4}$ | $3.11 \cdot 10^{-4}$ | 1.98 x |
| features.7.conv.1.0 | $2.38 \cdot 10^{-3}$ | $2.49 \cdot 10^{-4}$ | 9.58 x | $2.37 \cdot 10^{-3}$ | $2.51 \cdot 10^{-4}$ | 9.44 x |
| features.7.conv.2 | $7.49 \cdot 10^{-4}$ | $2.09 \cdot 10^{-4}$ | 3.58 x | $3.20 \cdot 10^{-4}$ | $2.10 \cdot 10^{-4}$ | 1.53 x |
| features.8.conv.0.0 | $8.05 \cdot 10^{-4}$ | $2.91 \cdot 10^{-4}$ | 2.77 x | $4.42 \cdot 10^{-4}$ | $2.92 \cdot 10^{-4}$ | 1.51 x |
| features.8.conv.1.0 | $2.29 \cdot 10^{-3}$ | $4.14 \cdot 10^{-4}$ | 5.53 x | $2.27 \cdot 10^{-3}$ | $4.15 \cdot 10^{-4}$ | 5.48 x |
| features.8.conv.2 | $7.98 \cdot 10^{-4}$ | $3.07 \cdot 10^{-4}$ | 2.60 x | $4.63 \cdot 10^{-4}$ | $3.06 \cdot 10^{-4}$ | 1.51 x |
| features.11.conv.2 | $9.88 \cdot 10^{-4}$ | $4.08 \cdot 10^{-4}$ | 2.42 x | $5.67 \cdot 10^{-4}$ | $4.07 \cdot 10^{-4}$ | 1.39 x |
| features.12.conv.0.0 | $1.06 \cdot 10^{-3}$ | $4.92 \cdot 10^{-4}$ | 2.16 x | $5.64 \cdot 10^{-4}$ | $4.92 \cdot 10^{-4}$ | 1.14 x |
| features.12.conv.1.0 | $4.18 \cdot 10^{-3}$ | $6.04 \cdot 10^{-4}$ | 6.91 x | $4.16 \cdot 10^{-3}$ | $6.05 \cdot 10^{-4}$ | 6.87 x |
| features.12.conv.2 | $1.16 \cdot 10^{-3}$ | $5.53 \cdot 10^{-4}$ | 2.10 x | $7.40 \cdot 10^{-4}$ | $5.55 \cdot 10^{-4}$ | 1.33 x |
| features.14.conv.1.0 | $1.73 \cdot 10^{-3}$ | $2.29 \cdot 10^{-4}$ | 7.57 x | $1.72 \cdot 10^{-3}$ | $2.28 \cdot 10^{-4}$ | 7.53 x |
| features.14.conv.2 | $6.95 \cdot 10^{-4}$ | $3.90 \cdot 10^{-4}$ | 1.78 x | $4.10 \cdot 10^{-4}$ | $3.90 \cdot 10^{-4}$ | 1.05 x |
| features.15.conv.0.0 | $9.24 \cdot 10^{-4}$ | $3.53 \cdot 10^{-4}$ | 2.62 x | $4.36 \cdot 10^{-4}$ | $3.53 \cdot 10^{-4}$ | 1.23 x |
| features.15.conv.1.0 | $1.49 \cdot 10^{-3}$ | $2.72 \cdot 10^{-4}$ | 5.46 x | $1.47 \cdot 10^{-3}$ | $2.73 \cdot 10^{-4}$ | 5.39 x |
| features.15.conv.2 | $8.32 \cdot 10^{-4}$ | $5.80 \cdot 10^{-4}$ | 1.43 x | $5.44 \cdot 10^{-4}$ | $5.80 \cdot 10^{-4}$ | **0.94 x** |
| features.17.conv.2 | $1.12 \cdot 10^{-3}$ | $9.74 \cdot 10^{-4}$ | 1.15 x | $7.14 \cdot 10^{-4}$ | $9.75 \cdot 10^{-4}$ | **0.73 x** |
| features.18.0 | $1.25 \cdot 10^{-3}$ | $7.31 \cdot 10^{-4}$ | 1.71 x | $8.01 \cdot 10^{-4}$ | $7.31 \cdot 10^{-4}$ | 1.10 x |

## F.3 Input VJP

We compare TN and TN+opt with a PyTorch implementation of the input VJP via `torch.autograd.grad`. Figure F20 visualizes the performance ratios for different convolution categories. Table F6 contains the detailed run times and performance factors.

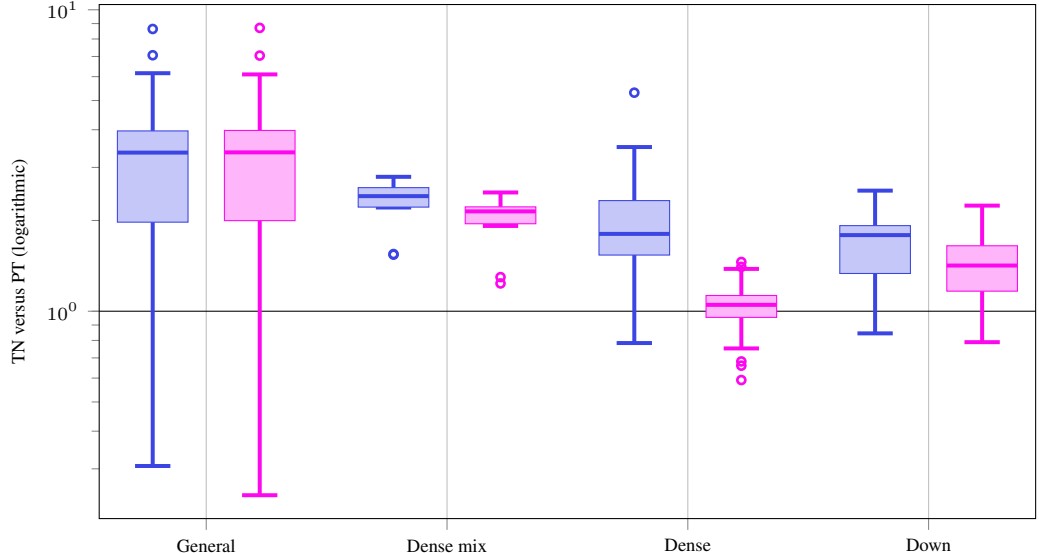

Figure F20: Input VJP performance ratios of TN versus PT and TN+opt versus PT for different convolution types on GPU.

## Table F6: Input VJP performance comparison on GPU.

### (a) 3c3d, CIFAR-10, input shape (128, 3, 32, 32)

| Name | TN [s] | PT [s] | Factor | TN + opt [s] | PT [s] | Factor |
|---|---|---|---|---|---|---|
| conv1.0 | $2.24 \cdot 10^{-3}$ | $1.39 \cdot 10^{-3}$ | 1.61 x | $2.19 \cdot 10^{-3}$ | $1.34 \cdot 10^{-3}$ | 1.63 x |
| conv2.0 | $2.61 \cdot 10^{-3}$ | $8.29 \cdot 10^{-4}$ | 3.15 x | $2.55 \cdot 10^{-3}$ | $7.86 \cdot 10^{-4}$ | 3.25 x |
| conv3.1 | $1.46 \cdot 10^{-3}$ | $5.04 \cdot 10^{-4}$ | 2.90 x | $1.42 \cdot 10^{-3}$ | $4.69 \cdot 10^{-4}$ | 3.02 x |

### (b) F-MNIST 2c2d, input shape (128, 1, 28, 28)

| Name | TN [s] | PT [s] | Factor | TN + opt [s] | PT [s] | Factor |
|---|---|---|---|---|---|---|
| conv1.1 | $9.47 \cdot 10^{-4}$ | $4.36 \cdot 10^{-4}$ | 2.17 x | $8.86 \cdot 10^{-4}$ | $4.40 \cdot 10^{-4}$ | 2.02 x |
| conv2.1 | $3.67 \cdot 10^{-3}$ | $9.83 \cdot 10^{-4}$ | 3.74 x | $3.62 \cdot 10^{-3}$ | $9.83 \cdot 10^{-4}$ | 3.69 x |

### (c) CIFAR-100 All-CNN-C, input shape (128, 3, 32, 32)

| Name | TN [s] | PT [s] | Factor | TN + opt [s] | PT [s] | Factor |
|---|---|---|---|---|---|---|
| conv1.1 | $1.91 \cdot 10^{-3}$ | $9.84 \cdot 10^{-4}$ | 1.94 x | $1.87 \cdot 10^{-3}$ | $9.37 \cdot 10^{-4}$ | 2.00 x |
| conv2.1 | $2.00 \cdot 10^{-2}$ | $5.95 \cdot 10^{-3}$ | 3.35 x | $2.00 \cdot 10^{-2}$ | $5.92 \cdot 10^{-3}$ | 3.37 x |
| conv3.1 | $7.82 \cdot 10^{-3}$ | $5.05 \cdot 10^{-3}$ | 1.55 x | $7.77 \cdot 10^{-3}$ | $5.01 \cdot 10^{-3}$ | 1.55 x |
| conv4.1 | $8.23 \cdot 10^{-3}$ | $3.11 \cdot 10^{-3}$ | 2.65 x | $8.17 \cdot 10^{-3}$ | $3.10 \cdot 10^{-3}$ | 2.63 x |
| conv5.1 | $1.56 \cdot 10^{-2}$ | $4.36 \cdot 10^{-3}$ | 3.57 x | $1.55 \cdot 10^{-2}$ | $4.36 \cdot 10^{-3}$ | 3.56 x |
| conv6.1 | $4.58 \cdot 10^{-3}$ | $3.96 \cdot 10^{-3}$ | 1.16 x | $4.53 \cdot 10^{-3}$ | $3.96 \cdot 10^{-3}$ | 1.14 x |
| conv7.0 | $2.86 \cdot 10^{-3}$ | $8.32 \cdot 10^{-4}$ | 3.44 x | $2.81 \cdot 10^{-3}$ | $8.68 \cdot 10^{-4}$ | 3.24 x |
| conv8.1 | $8.31 \cdot 10^{-4}$ | $2.91 \cdot 10^{-4}$ | 2.85 x | $3.47 \cdot 10^{-4}$ | $3.32 \cdot 10^{-4}$ | 1.04 x |
| conv9.1 | $7.76 \cdot 10^{-4}$ | $2.21 \cdot 10^{-4}$ | 3.51 x | $2.90 \cdot 10^{-4}$ | $2.61 \cdot 10^{-4}$ | 1.11 x |

### (d) Alexnet, input shape (32, 3, 256, 256)

| Name | TN [s] | PT [s] | Factor | TN + opt [s] | PT [s] | Factor |
|---|---|---|---|---|---|---|
| features.0 | $1.92 \cdot 10^{-2}$ | $5.48 \cdot 10^{-3}$ | 3.50 x | $1.92 \cdot 10^{-2}$ | $5.54 \cdot 10^{-3}$ | 3.46 x |
| features.3 | $1.15 \cdot 10^{-2}$ | $4.16 \cdot 10^{-3}$ | 2.77 x | $1.15 \cdot 10^{-2}$ | $4.20 \cdot 10^{-3}$ | 2.75 x |
| features.6 | $5.36 \cdot 10^{-3}$ | $1.49 \cdot 10^{-3}$ | 3.60 x | $5.36 \cdot 10^{-3}$ | $1.49 \cdot 10^{-3}$ | 3.60 x |
| features.8 | $6.26 \cdot 10^{-3}$ | $1.86 \cdot 10^{-3}$ | 3.36 x | $6.25 \cdot 10^{-3}$ | $1.86 \cdot 10^{-3}$ | 3.37 x |
| features.10 | $4.41 \cdot 10^{-3}$ | $1.32 \cdot 10^{-3}$ | 3.35 x | $4.40 \cdot 10^{-3}$ | $1.35 \cdot 10^{-3}$ | 3.26 x |

### (e) ResNet18, input shape (32, 3, 256, 256)

| Name | TN [s] | PT [s] | Factor | TN + opt [s] | PT [s] | Factor |
|---|---|---|---|---|---|---|
| conv1 | $3.38 \cdot 10^{-2}$ | $8.56 \cdot 10^{-3}$ | 3.96 x | $3.38 \cdot 10^{-2}$ | $8.49 \cdot 10^{-3}$ | 3.98 x |
| layer1.0.conv1 | $1.06 \cdot 10^{-2}$ | $2.17 \cdot 10^{-3}$ | 4.87 x | $1.05 \cdot 10^{-2}$ | $2.11 \cdot 10^{-3}$ | 4.99 x |
| layer2.0.conv1 | $7.03 \cdot 10^{-3}$ | $3.72 \cdot 10^{-3}$ | 1.89 x | $6.95 \cdot 10^{-3}$ | $3.66 \cdot 10^{-3}$ | 1.90 x |
| layer2.0.conv2 | $6.91 \cdot 10^{-3}$ | $1.55 \cdot 10^{-3}$ | 4.47 x | $6.90 \cdot 10^{-3}$ | $1.51 \cdot 10^{-3}$ | 4.56 x |
| layer2.0.downsample.0 | $2.02 \cdot 10^{-3}$ | $8.02 \cdot 10^{-4}$ | 2.51 x | $1.71 \cdot 10^{-3}$ | $7.64 \cdot 10^{-4}$ | 2.24 x |
| layer3.0.conv1 | $3.94 \cdot 10^{-3}$ | $3.05 \cdot 10^{-3}$ | 1.29 x | $3.88 \cdot 10^{-3}$ | $3.01 \cdot 10^{-3}$ | 1.29 x |
| layer3.0.conv2 | $5.07 \cdot 10^{-3}$ | $1.31 \cdot 10^{-3}$ | 3.87 x | $5.07 \cdot 10^{-3}$ | $1.36 \cdot 10^{-3}$ | 3.74 x |
| layer3.0.downsample.0 | $1.15 \cdot 10^{-3}$ | $5.96 \cdot 10^{-4}$ | 1.94 x | $9.54 \cdot 10^{-4}$ | $6.40 \cdot 10^{-4}$ | 1.49 x |
| layer4.0.conv1 | $2.89 \cdot 10^{-3}$ | $3.08 \cdot 10^{-3}$ | **0.94 x** | $2.84 \cdot 10^{-3}$ | $3.12 \cdot 10^{-3}$ | **0.91 x** |
| layer4.0.conv2 | $4.50 \cdot 10^{-3}$ | $1.40 \cdot 10^{-3}$ | 3.21 x | $4.49 \cdot 10^{-3}$ | $1.44 \cdot 10^{-3}$ | 3.12 x |
| layer4.0.downsample.0 | $9.35 \cdot 10^{-4}$ | $5.51 \cdot 10^{-4}$ | 1.70 x | $7.93 \cdot 10^{-4}$ | $5.90 \cdot 10^{-4}$ | 1.34 x |

### (f) ResNext101, input shape (32, 3, 256, 256)

| Name | TN [s] | PT [s] | Factor | TN + opt [s] | PT [s] | Factor |
|---|---|---|---|---|---|---|
| conv1 | $3.38 \cdot 10^{-2}$ | $8.52 \cdot 10^{-3}$ | 3.97 x | $3.38 \cdot 10^{-2}$ | $8.48 \cdot 10^{-3}$ | 3.98 x |
| layer1.0.conv1 | $6.18 \cdot 10^{-3}$ | $1.96 \cdot 10^{-3}$ | 3.15 x | $2.86 \cdot 10^{-3}$ | $1.96 \cdot 10^{-3}$ | 1.46 x |
| layer1.0.conv2 | $3.04 \cdot 10^{-2}$ | $1.17 \cdot 10^{-2}$ | 2.60 x | $3.05 \cdot 10^{-2}$ | $1.17 \cdot 10^{-2}$ | 2.61 x |
| layer1.0.conv3 | $1.46 \cdot 10^{-2}$ | $6.57 \cdot 10^{-3}$ | 2.22 x | $7.39 \cdot 10^{-3}$ | $6.58 \cdot 10^{-3}$ | 1.12 x |
| layer2.0.conv1 | $2.40 \cdot 10^{-2}$ | $1.14 \cdot 10^{-2}$ | 2.10 x | $1.44 \cdot 10^{-2}$ | $1.17 \cdot 10^{-2}$ | 1.23 x |
| layer2.0.conv2 | $2.75 \cdot 10^{-2}$ | $1.96 \cdot 10^{-2}$ | 1.40 x | $2.75 \cdot 10^{-2}$ | $1.95 \cdot 10^{-2}$ | 1.41 x |
| layer2.0.conv3 | $1.04 \cdot 10^{-2}$ | $6.43 \cdot 10^{-3}$ | 1.61 x | $6.74 \cdot 10^{-3}$ | $6.43 \cdot 10^{-3}$ | 1.05 x |
| layer2.0.downsample.0 | $8.99 \cdot 10^{-3}$ | $4.78 \cdot 10^{-3}$ | 1.88 x | $8.06 \cdot 10^{-3}$ | $4.74 \cdot 10^{-3}$ | 1.70 x |
| layer2.1.conv2 | $1.51 \cdot 10^{-2}$ | $4.46 \cdot 10^{-3}$ | 3.38 x | $1.51 \cdot 10^{-2}$ | $4.45 \cdot 10^{-3}$ | 3.39 x |
| layer3.0.conv1 | $1.94 \cdot 10^{-2}$ | $1.25 \cdot 10^{-2}$ | 1.55 x | $1.32 \cdot 10^{-2}$ | $1.25 \cdot 10^{-2}$ | 1.06 x |
| layer3.0.conv2 | $1.76 \cdot 10^{-2}$ | $8.33 \cdot 10^{-3}$ | 2.11 x | $1.76 \cdot 10^{-2}$ | $8.34 \cdot 10^{-3}$ | 2.11 x |
| layer3.0.conv3 | $8.21 \cdot 10^{-3}$ | $6.32 \cdot 10^{-3}$ | 1.30 x | $6.39 \cdot 10^{-3}$ | $6.32 \cdot 10^{-3}$ | 1.01 x |
| layer3.0.downsample.0 | $5.51 \cdot 10^{-3}$ | $4.54 \cdot 10^{-3}$ | 1.21 x | $5.00 \cdot 10^{-3}$ | $4.52 \cdot 10^{-3}$ | 1.11 x |
| layer3.1.conv2 | $7.60 \cdot 10^{-3}$ | $1.97 \cdot 10^{-3}$ | 3.85 x | $7.60 \cdot 10^{-3}$ | $1.98 \cdot 10^{-3}$ | 3.84 x |
| layer4.0.conv1 | $1.51 \cdot 10^{-2}$ | $1.24 \cdot 10^{-2}$ | 1.22 x | $1.26 \cdot 10^{-2}$ | $1.24 \cdot 10^{-2}$ | 1.02 x |
| layer4.0.conv2 | $8.24 \cdot 10^{-3}$ | $5.43 \cdot 10^{-3}$ | 1.52 x | $8.24 \cdot 10^{-3}$ | $5.44 \cdot 10^{-3}$ | 1.51 x |
| layer4.0.conv3 | $7.65 \cdot 10^{-3}$ | $6.72 \cdot 10^{-3}$ | 1.14 x | $6.25 \cdot 10^{-3}$ | $6.73 \cdot 10^{-3}$ | **0.93 x** |
| layer4.0.downsample.0 | $4.61 \cdot 10^{-3}$ | $5.45 \cdot 10^{-3}$ | **0.84 x** | $4.31 \cdot 10^{-3}$ | $5.45 \cdot 10^{-3}$ | **0.79 x** |
| layer4.1.conv2 | $4.79 \cdot 10^{-3}$ | $1.34 \cdot 10^{-3}$ | 3.57 x | $4.79 \cdot 10^{-3}$ | $1.39 \cdot 10^{-3}$ | 3.44 x |

### (g) ConvNeXt-base, input shape (32, 3, 256, 256)

| Name | TN [s] | PT [s] | Factor | TN + opt [s] | PT [s] | Factor |
|---|---|---|---|---|---|---|
| features.0.0 | $5.36 \cdot 10^{-3}$ | $1.79 \cdot 10^{-3}$ | 2.99 x | $1.57 \cdot 10^{-3}$ | $1.79 \cdot 10^{-3}$ | **0.88 x** |
| features.1.0.block.0 | $4.63 \cdot 10^{-2}$ | $8.60 \cdot 10^{-3}$ | 5.38 x | $4.63 \cdot 10^{-2}$ | $8.58 \cdot 10^{-3}$ | 5.40 x |
| features.2.1 | $8.85 \cdot 10^{-3}$ | $5.37 \cdot 10^{-3}$ | 1.65 x | $3.55 \cdot 10^{-3}$ | $5.38 \cdot 10^{-3}$ | **0.66 x** |
| features.3.0.block.0 | $2.14 \cdot 10^{-2}$ | $4.21 \cdot 10^{-3}$ | 5.09 x | $2.14 \cdot 10^{-2}$ | $4.21 \cdot 10^{-3}$ | 5.09 x |
| features.4.1 | $5.64 \cdot 10^{-3}$ | $4.43 \cdot 10^{-3}$ | 1.27 x | $3.34 \cdot 10^{-3}$ | $4.43 \cdot 10^{-3}$ | **0.75 x** |
| features.5.0.block.0 | $1.05 \cdot 10^{-2}$ | $2.16 \cdot 10^{-3}$ | 4.87 x | $1.05 \cdot 10^{-2}$ | $2.16 \cdot 10^{-3}$ | 4.86 x |
| features.6.1 | $4.31 \cdot 10^{-3}$ | $5.50 \cdot 10^{-3}$ | **0.78 x** | $3.25 \cdot 10^{-3}$ | $5.50 \cdot 10^{-3}$ | **0.59 x** |
| features.7.0.block.0 | $1.09 \cdot 10^{-3}$ | $1.17 \cdot 10^{-3}$ | **0.93 x** | $1.06 \cdot 10^{-3}$ | $1.15 \cdot 10^{-3}$ | **0.92 x** |

(h) InceptionV3, input shape (32, 3, 299, 299)

| Name | TN [s] | PT [s] | Factor | TN + opt [s] | PT [s] | Factor |
|---|---|---|---|---|---|---|
| Conv2d_1a_3x3.conv | $1.27 \cdot 10^{-2}$ | $3.19 \cdot 10^{-3}$ | 3.97 x | $1.26 \cdot 10^{-2}$ | $3.21 \cdot 10^{-3}$ | 3.92 x |
| Conv2d_2a_3x3.conv | $3.16 \cdot 10^{-2}$ | $5.13 \cdot 10^{-3}$ | 6.16 x | $3.16 \cdot 10^{-2}$ | $5.17 \cdot 10^{-3}$ | 6.10 x |
| Conv2d_2b_3x3.conv | $4.32 \cdot 10^{-2}$ | $8.11 \cdot 10^{-3}$ | 5.33 x | $4.24 \cdot 10^{-2}$ | $8.17 \cdot 10^{-3}$ | 5.19 x |
| Conv2d_3b_1x1.conv | $5.76 \cdot 10^{-3}$ | $1.09 \cdot 10^{-3}$ | 5.31 x | $1.37 \cdot 10^{-3}$ | $1.09 \cdot 10^{-3}$ | 1.25 x |
| Conv2d_4a_3x3.conv | $3.71 \cdot 10^{-2}$ | $1.12 \cdot 10^{-2}$ | 3.30 x | $3.71 \cdot 10^{-2}$ | $1.12 \cdot 10^{-2}$ | 3.30 x |
| Mixed_5b.branch1x1.conv | $1.37 \cdot 10^{-3}$ | $6.89 \cdot 10^{-4}$ | 1.99 x | $6.67 \cdot 10^{-4}$ | $6.88 \cdot 10^{-4}$ | **0.97 x** |
| Mixed_5b.branch5x5_1.conv | $1.13 \cdot 10^{-3}$ | $5.87 \cdot 10^{-4}$ | 1.92 x | $5.70 \cdot 10^{-4}$ | $5.88 \cdot 10^{-4}$ | **0.97 x** |
| Mixed_5b.branch5x5_2.conv | $4.97 \cdot 10^{-3}$ | $1.39 \cdot 10^{-3}$ | 3.58 x | $4.97 \cdot 10^{-3}$ | $1.39 \cdot 10^{-3}$ | 3.57 x |
| Mixed_5b.branch3x3dbl_2.conv | $4.23 \cdot 10^{-3}$ | $1.07 \cdot 10^{-3}$ | 3.98 x | $4.23 \cdot 10^{-3}$ | $1.06 \cdot 10^{-3}$ | 3.98 x |
| Mixed_5b.branch3x3dbl_3.conv | $5.68 \cdot 10^{-3}$ | $1.66 \cdot 10^{-3}$ | 3.41 x | $5.68 \cdot 10^{-3}$ | $1.66 \cdot 10^{-3}$ | 3.41 x |
| Mixed_5b.branch_pool.conv | $9.70 \cdot 10^{-4}$ | $5.10 \cdot 10^{-4}$ | 1.90 x | $4.77 \cdot 10^{-4}$ | $5.13 \cdot 10^{-4}$ | **0.93 x** |
| Mixed_5c.branch1x1.conv | $1.48 \cdot 10^{-3}$ | $8.10 \cdot 10^{-4}$ | 1.82 x | $8.07 \cdot 10^{-4}$ | $8.10 \cdot 10^{-4}$ | **1.00 x** |
| Mixed_5c.branch5x5_1.conv | $1.23 \cdot 10^{-3}$ | $6.85 \cdot 10^{-4}$ | 1.79 x | $6.81 \cdot 10^{-4}$ | $6.87 \cdot 10^{-4}$ | **0.99 x** |
| Mixed_5d.branch1x1.conv | $1.68 \cdot 10^{-3}$ | $1.04 \cdot 10^{-3}$ | 1.61 x | $9.40 \cdot 10^{-4}$ | $1.05 \cdot 10^{-3}$ | **0.90 x** |
| Mixed_5d.branch5x5_1.conv | $1.38 \cdot 10^{-3}$ | $8.69 \cdot 10^{-4}$ | 1.59 x | $7.84 \cdot 10^{-4}$ | $8.12 \cdot 10^{-4}$ | **0.96 x** |
| Mixed_6a.branch3x3.conv | $1.14 \cdot 10^{-2}$ | $1.32 \cdot 10^{-2}$ | **0.86 x** | $1.14 \cdot 10^{-2}$ | $1.32 \cdot 10^{-2}$ | **0.86 x** |
| Mixed_6a.branch3x3dbl_3.conv | $2.52 \cdot 10^{-3}$ | $1.70 \cdot 10^{-3}$ | 1.48 x | $2.46 \cdot 10^{-3}$ | $1.70 \cdot 10^{-3}$ | 1.45 x |
| Mixed_6b.branch1x1.conv | $1.78 \cdot 10^{-3}$ | $1.18 \cdot 10^{-3}$ | 1.50 x | $1.24 \cdot 10^{-3}$ | $1.19 \cdot 10^{-3}$ | 1.05 x |
| Mixed_6b.branch7x7_1.conv | $1.37 \cdot 10^{-3}$ | $8.69 \cdot 10^{-4}$ | 1.58 x | $9.27 \cdot 10^{-4}$ | $8.70 \cdot 10^{-4}$ | 1.07 x |
| Mixed_6b.branch7x7_2.conv | $2.13 \cdot 10^{-3}$ | $8.27 \cdot 10^{-4}$ | 2.58 x | $1.79 \cdot 10^{-3}$ | $8.27 \cdot 10^{-4}$ | 2.16 x |
| Mixed_6b.branch7x7_3.conv | $2.54 \cdot 10^{-3}$ | $1.08 \cdot 10^{-3}$ | 2.36 x | $2.22 \cdot 10^{-3}$ | $1.08 \cdot 10^{-3}$ | 2.05 x |
| Mixed_6b.branch7x7dbl_2.conv | $2.08 \cdot 10^{-3}$ | $8.10 \cdot 10^{-4}$ | 2.57 x | $1.80 \cdot 10^{-3}$ | $8.09 \cdot 10^{-4}$ | 2.22 x |
| Mixed_6b.branch7x7dbl_5.conv | $2.45 \cdot 10^{-3}$ | $1.11 \cdot 10^{-3}$ | 2.21 x | $2.14 \cdot 10^{-3}$ | $1.11 \cdot 10^{-3}$ | 1.92 x |
| Mixed_6c.branch7x7_1.conv | $1.56 \cdot 10^{-3}$ | $1.03 \cdot 10^{-3}$ | 1.52 x | $1.09 \cdot 10^{-3}$ | $1.03 \cdot 10^{-3}$ | 1.06 x |
| Mixed_6c.branch7x7_2.conv | $3.19 \cdot 10^{-3}$ | $1.14 \cdot 10^{-3}$ | 2.79 x | $2.76 \cdot 10^{-3}$ | $1.13 \cdot 10^{-3}$ | 2.43 x |
| Mixed_6c.branch7x7_3.conv | $3.06 \cdot 10^{-3}$ | $1.28 \cdot 10^{-3}$ | 2.40 x | $2.72 \cdot 10^{-3}$ | $1.28 \cdot 10^{-3}$ | 2.12 x |
| Mixed_6c.branch7x7dbl_2.conv | $3.10 \cdot 10^{-3}$ | $1.12 \cdot 10^{-3}$ | 2.78 x | $2.77 \cdot 10^{-3}$ | $1.12 \cdot 10^{-3}$ | 2.48 x |
| Mixed_6c.branch7x7dbl_5.conv | $2.96 \cdot 10^{-3}$ | $1.33 \cdot 10^{-3}$ | 2.22 x | $2.61 \cdot 10^{-3}$ | $1.33 \cdot 10^{-3}$ | 1.96 x |
| Mixed_6e.branch7x7_2.conv | $3.77 \cdot 10^{-3}$ | $1.54 \cdot 10^{-3}$ | 2.45 x | $3.26 \cdot 10^{-3}$ | $1.50 \cdot 10^{-3}$ | 2.17 x |
| Mixed_6e.branch7x7_3.conv | $3.65 \cdot 10^{-3}$ | $1.51 \cdot 10^{-3}$ | 2.42 x | $3.27 \cdot 10^{-3}$ | $1.47 \cdot 10^{-3}$ | 2.22 x |
| AuxLogits.conv0.conv | $5.53 \cdot 10^{-4}$ | $2.74 \cdot 10^{-4}$ | 2.02 x | $3.03 \cdot 10^{-4}$ | $2.34 \cdot 10^{-4}$ | 1.30 x |
| AuxLogits.conv1.conv | $6.27 \cdot 10^{-4}$ | $2.04 \cdot 10^{-3}$ | **0.31 x** | $4.94 \cdot 10^{-4}$ | $2.02 \cdot 10^{-3}$ | **0.25 x** |
| Mixed_7a.branch3x3_2.conv | $1.56 \cdot 10^{-3}$ | $7.08 \cdot 10^{-4}$ | 2.21 x | $1.47 \cdot 10^{-3}$ | $6.64 \cdot 10^{-4}$ | 2.22 x |
| Mixed_7a.branch7x7x3_4.conv | $1.46 \cdot 10^{-3}$ | $1.10 \cdot 10^{-3}$ | 1.33 x | $1.42 \cdot 10^{-3}$ | $1.14 \cdot 10^{-3}$ | 1.25 x |
| Mixed_7b.branch1x1.conv | $1.31 \cdot 10^{-3}$ | $7.40 \cdot 10^{-4}$ | 1.77 x | $8.47 \cdot 10^{-4}$ | $7.89 \cdot 10^{-4}$ | 1.07 x |
| Mixed_7b.branch3x3_1.conv | $1.44 \cdot 10^{-3}$ | $8.55 \cdot 10^{-4}$ | 1.69 x | $9.54 \cdot 10^{-4}$ | $9.00 \cdot 10^{-4}$ | 1.06 x |
| Mixed_7b.branch3x3_2a.conv | $1.51 \cdot 10^{-3}$ | $9.77 \cdot 10^{-4}$ | 1.55 x | $1.26 \cdot 10^{-3}$ | $1.02 \cdot 10^{-3}$ | 1.24 x |
| Mixed_7b.branch3x3_2b.conv | $1.50 \cdot 10^{-3}$ | $9.77 \cdot 10^{-4}$ | 1.54 x | $1.27 \cdot 10^{-3}$ | $9.78 \cdot 10^{-4}$ | 1.30 x |
| Mixed_7b.branch3x3dbl_1.conv | $1.56 \cdot 10^{-3}$ | $1.02 \cdot 10^{-3}$ | 1.54 x | $1.07 \cdot 10^{-3}$ | $9.72 \cdot 10^{-4}$ | 1.10 x |
| Mixed_7b.branch3x3dbl_2.conv | $3.32 \cdot 10^{-3}$ | $1.02 \cdot 10^{-3}$ | 3.24 x | $3.28 \cdot 10^{-3}$ | $9.91 \cdot 10^{-4}$ | 3.31 x |
| Mixed_7b.branch_pool.conv | $1.01 \cdot 10^{-3}$ | $5.57 \cdot 10^{-4}$ | 1.81 x | $6.18 \cdot 10^{-4}$ | $5.10 \cdot 10^{-4}$ | 1.21 x |
| Mixed_7c.branch1x1.conv | $1.69 \cdot 10^{-3}$ | $1.25 \cdot 10^{-3}$ | 1.35 x | $1.21 \cdot 10^{-3}$ | $1.22 \cdot 10^{-3}$ | **0.99 x** |
| Mixed_7c.branch3x3_1.conv | $1.86 \cdot 10^{-3}$ | $1.45 \cdot 10^{-3}$ | 1.28 x | $1.39 \cdot 10^{-3}$ | $1.45 \cdot 10^{-3}$ | **0.95 x** |
| Mixed_7c.branch3x3dbl_1.conv | $2.05 \cdot 10^{-3}$ | $1.66 \cdot 10^{-3}$ | 1.23 x | $1.57 \cdot 10^{-3}$ | $1.66 \cdot 10^{-3}$ | **0.95 x** |
| Mixed_7c.branch_pool.conv | $1.27 \cdot 10^{-3}$ | $8.35 \cdot 10^{-4}$ | 1.53 x | $8.32 \cdot 10^{-4}$ | $8.35 \cdot 10^{-4}$ | **1.00 x** |

(i) MobileNetV2, input shape (32, 3, 256, 256)

| Name | TN [s] | PT [s] | Factor | TN + opt [s] | PT [s] | Factor |
|---|---|---|---|---|---|---|
| features.0.0 | $8.32 \cdot 10^{-3}$ | $2.08 \cdot 10^{-3}$ | 4.01 x | $8.26 \cdot 10^{-3}$ | $2.02 \cdot 10^{-3}$ | 4.09 x |
| features.1.conv.0.0 | $2.27 \cdot 10^{-2}$ | $2.63 \cdot 10^{-3}$ | 8.64 x | $2.27 \cdot 10^{-2}$ | $2.60 \cdot 10^{-3}$ | 8.71 x |
| features.1.conv.1 | $3.22 \cdot 10^{-3}$ | $1.17 \cdot 10^{-3}$ | 2.75 x | $1.02 \cdot 10^{-3}$ | $1.16 \cdot 10^{-3}$ | **0.87 x** |
| features.2.conv.0.0 | $7.67 \cdot 10^{-3}$ | $2.61 \cdot 10^{-3}$ | 2.94 x | $3.66 \cdot 10^{-3}$ | $2.61 \cdot 10^{-3}$ | 1.40 x |
| features.2.conv.1.0 | $2.79 \cdot 10^{-2}$ | $8.11 \cdot 10^{-3}$ | 3.44 x | $2.79 \cdot 10^{-2}$ | $8.11 \cdot 10^{-3}$ | 3.43 x |
| features.2.conv.2 | $1.48 \cdot 10^{-3}$ | $6.38 \cdot 10^{-4}$ | 2.33 x | $7.53 \cdot 10^{-4}$ | $6.40 \cdot 10^{-4}$ | 1.18 x |
| features.3.conv.0.0 | $2.86 \cdot 10^{-3}$ | $1.05 \cdot 10^{-3}$ | 2.73 x | $1.45 \cdot 10^{-3}$ | $1.05 \cdot 10^{-3}$ | 1.38 x |
| features.3.conv.1.0 | $2.08 \cdot 10^{-2}$ | $2.95 \cdot 10^{-3}$ | 7.07 x | $2.08 \cdot 10^{-2}$ | $2.95 \cdot 10^{-3}$ | 7.05 x |
| features.3.conv.2 | $1.77 \cdot 10^{-3}$ | $1.04 \cdot 10^{-3}$ | 1.70 x | $9.75 \cdot 10^{-4}$ | $1.04 \cdot 10^{-3}$ | **0.94 x** |
| features.4.conv.1.0 | $7.63 \cdot 10^{-3}$ | $3.15 \cdot 10^{-3}$ | 2.42 x | $7.62 \cdot 10^{-3}$ | $3.15 \cdot 10^{-3}$ | 2.42 x |
| features.4.conv.2 | $9.49 \cdot 10^{-4}$ | $4.32 \cdot 10^{-4}$ | 2.20 x | $4.38 \cdot 10^{-4}$ | $3.88 \cdot 10^{-4}$ | 1.13 x |
| features.5.conv.0.0 | $1.20 \cdot 10^{-3}$ | $5.26 \cdot 10^{-4}$ | 2.29 x | $6.09 \cdot 10^{-4}$ | $4.83 \cdot 10^{-4}$ | 1.26 x |
| features.5.conv.1.0 | $5.41 \cdot 10^{-3}$ | $1.02 \cdot 10^{-3}$ | 5.29 x | $5.39 \cdot 10^{-3}$ | $1.02 \cdot 10^{-3}$ | 5.27 x |
| features.5.conv.2 | $9.53 \cdot 10^{-4}$ | $4.00 \cdot 10^{-4}$ | 2.38 x | $4.35 \cdot 10^{-4}$ | $3.98 \cdot 10^{-4}$ | 1.09 x |
| features.7.conv.1.0 | $2.11 \cdot 10^{-3}$ | $1.07 \cdot 10^{-3}$ | 1.97 x | $2.10 \cdot 10^{-3}$ | $1.07 \cdot 10^{-3}$ | 1.97 x |
| features.7.conv.2 | $7.77 \cdot 10^{-4}$ | $2.33 \cdot 10^{-4}$ | 3.34 x | $3.04 \cdot 10^{-4}$ | $2.33 \cdot 10^{-4}$ | 1.30 x |
| features.8.conv.0.0 | $8.13 \cdot 10^{-4}$ | $3.41 \cdot 10^{-4}$ | 2.38 x | $4.63 \cdot 10^{-4}$ | $3.40 \cdot 10^{-4}$ | 1.36 x |
| features.8.conv.1.0 | $2.09 \cdot 10^{-3}$ | $5.48 \cdot 10^{-4}$ | 3.81 x | $2.07 \cdot 10^{-3}$ | $5.47 \cdot 10^{-4}$ | 3.79 x |
| features.8.conv.2 | $8.65 \cdot 10^{-4}$ | $3.16 \cdot 10^{-4}$ | 2.74 x | $4.04 \cdot 10^{-4}$ | $3.16 \cdot 10^{-4}$ | 1.28 x |
| features.11.conv.2 | $9.34 \cdot 10^{-4}$ | $4.22 \cdot 10^{-4}$ | 2.21 x | $4.74 \cdot 10^{-4}$ | $4.24 \cdot 10^{-4}$ | 1.12 x |
| features.12.conv.0.0 | $1.16 \cdot 10^{-3}$ | $7.11 \cdot 10^{-4}$ | 1.64 x | $7.37 \cdot 10^{-4}$ | $7.10 \cdot 10^{-4}$ | 1.04 x |
| features.12.conv.1.0 | $3.84 \cdot 10^{-3}$ | $7.91 \cdot 10^{-4}$ | 4.85 x | $3.82 \cdot 10^{-3}$ | $7.91 \cdot 10^{-4}$ | 4.83 x |
| features.12.conv.2 | $1.08 \cdot 10^{-3}$ | $5.71 \cdot 10^{-4}$ | 1.90 x | $6.13 \cdot 10^{-4}$ | $5.73 \cdot 10^{-4}$ | 1.07 x |
| features.14.conv.1.0 | $1.61 \cdot 10^{-3}$ | $8.26 \cdot 10^{-4}$ | 1.95 x | $1.60 \cdot 10^{-3}$ | $8.26 \cdot 10^{-4}$ | 1.93 x |
| features.14.conv.2 | $8.14 \cdot 10^{-4}$ | $2.87 \cdot 10^{-4}$ | 2.83 x | $3.84 \cdot 10^{-4}$ | $2.87 \cdot 10^{-4}$ | 1.34 x |
| features.15.conv.0.0 | $8.46 \cdot 10^{-4}$ | $6.29 \cdot 10^{-4}$ | 1.34 x | $5.55 \cdot 10^{-4}$ | $6.08 \cdot 10^{-4}$ | **0.91 x** |
| features.15.conv.1.0 | $1.52 \cdot 10^{-3}$ | $3.62 \cdot 10^{-4}$ | 4.21 x | $1.50 \cdot 10^{-3}$ | $3.61 \cdot 10^{-4}$ | 4.17 x |
| features.15.conv.2 | $9.64 \cdot 10^{-4}$ | $4.43 \cdot 10^{-4}$ | 2.18 x | $4.82 \cdot 10^{-4}$ | $4.44 \cdot 10^{-4}$ | 1.09 x |
| features.17.conv.2 | $1.23 \cdot 10^{-3}$ | $7.30 \cdot 10^{-4}$ | 1.69 x | $6.98 \cdot 10^{-4}$ | $7.32 \cdot 10^{-4}$ | **0.95 x** |
| features.18.0 | $1.29 \cdot 10^{-3}$ | $1.28 \cdot 10^{-3}$ | 1.00 x | $8.76 \cdot 10^{-4}$ | $1.28 \cdot 10^{-3}$ | **0.68 x** |

### F.4 Weight VJP

We compare TN and TN+opt with a PyTorch implementation of the weight VJP via `torch.autograd.grad`. Figure F21 visualizes the performance ratios for different convolution categories. Table F7 contains the detailed run times and performance factors.

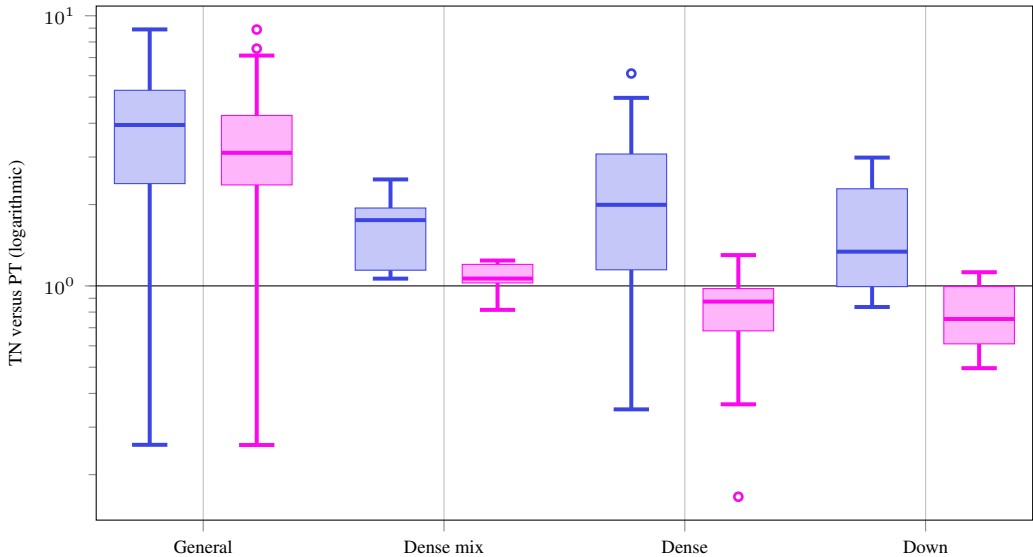

Figure F21: Weight VJP performance ratios of TN versus PT and TN+opt versus PT for different convolution types on GPU.

### Table F7: Weight VJP performance comparison on GPU.

#### (a) 3c3d, CIFAR-10, input shape (128, 3, 32, 32)

| Name | TN [s] | PT [s] | Factor | TN + opt [s] | PT [s] | Factor |
|---|---|---|---|---|---|---|
| conv1.0 | $2.27 \cdot 10^{-3}$ | $1.50 \cdot 10^{-3}$ | 1.52 x | $2.27 \cdot 10^{-3}$ | $1.50 \cdot 10^{-3}$ | 1.51 x |
| conv2.0 | $3.00 \cdot 10^{-3}$ | $1.12 \cdot 10^{-3}$ | 2.68 x | $2.99 \cdot 10^{-3}$ | $1.07 \cdot 10^{-3}$ | 2.78 x |
| conv3.1 | $1.29 \cdot 10^{-3}$ | $5.46 \cdot 10^{-4}$ | 2.37 x | $1.25 \cdot 10^{-3}$ | $5.08 \cdot 10^{-4}$ | 2.47 x |

#### (b) F-MNIST 2c2d, input shape (128, 1, 28, 28)

| Name | TN [s] | PT [s] | Factor | TN + opt [s] | PT [s] | Factor |
|---|---|---|---|---|---|---|
| conv1.1 | $1.08 \cdot 10^{-3}$ | $3.81 \cdot 10^{-4}$ | 2.83 x | $1.03 \cdot 10^{-3}$ | $4.05 \cdot 10^{-4}$ | 2.54 x |
| conv2.1 | $4.12 \cdot 10^{-3}$ | $1.02 \cdot 10^{-3}$ | 4.02 x | $4.09 \cdot 10^{-3}$ | $1.03 \cdot 10^{-3}$ | 3.98 x |

#### (c) CIFAR-100 All-CNN-C, input shape (128, 3, 32, 32)

| Name | TN [s] | PT [s] | Factor | TN + opt [s] | PT [s] | Factor |
|---|---|---|---|---|---|---|
| conv1.1 | $2.43 \cdot 10^{-3}$ | $1.02 \cdot 10^{-3}$ | 2.39 x | $2.42 \cdot 10^{-3}$ | $1.02 \cdot 10^{-3}$ | 2.37 x |
| conv2.1 | $3.83 \cdot 10^{-2}$ | $5.62 \cdot 10^{-3}$ | 6.81 x | $1.97 \cdot 10^{-2}$ | $5.62 \cdot 10^{-3}$ | 3.51 x |
| conv3.1 | $8.30 \cdot 10^{-3}$ | $4.14 \cdot 10^{-3}$ | 2.00 x | $8.33 \cdot 10^{-3}$ | $4.21 \cdot 10^{-3}$ | 1.98 x |
| conv4.1 | $8.66 \cdot 10^{-3}$ | $2.64 \cdot 10^{-3}$ | 3.28 x | $8.68 \cdot 10^{-3}$ | $2.68 \cdot 10^{-3}$ | 3.24 x |
| conv5.1 | $1.60 \cdot 10^{-2}$ | $3.38 \cdot 10^{-3}$ | 4.75 x | $1.61 \cdot 10^{-2}$ | $3.42 \cdot 10^{-3}$ | 4.70 x |
| conv6.1 | $5.23 \cdot 10^{-3}$ | $2.80 \cdot 10^{-3}$ | 1.87 x | $5.17 \cdot 10^{-3}$ | $2.81 \cdot 10^{-3}$ | 1.84 x |
| conv7.0 | $2.68 \cdot 10^{-3}$ | $9.97 \cdot 10^{-4}$ | 2.68 x | $2.59 \cdot 10^{-3}$ | $1.04 \cdot 10^{-3}$ | 2.49 x |
| conv8.1 | $9.13 \cdot 10^{-4}$ | $2.62 \cdot 10^{-3}$ | **0.35 x** | $4.33 \cdot 10^{-4}$ | $2.62 \cdot 10^{-3}$ | **0.17 x** |
| conv9.1 | $8.78 \cdot 10^{-4}$ | $3.54 \cdot 10^{-4}$ | 2.48 x | $3.93 \cdot 10^{-4}$ | $3.50 \cdot 10^{-4}$ | 1.12 x |

#### (d) Alexnet, input shape (32, 3, 256, 256)

| Name | TN [s] | PT [s] | Factor | TN + opt [s] | PT [s] | Factor |
|---|---|---|---|---|---|---|
| features.0 | $1.82 \cdot 10^{-2}$ | $3.31 \cdot 10^{-3}$ | 5.50 x | $1.82 \cdot 10^{-2}$ | $3.33 \cdot 10^{-3}$ | 5.46 x |
| features.3 | $2.02 \cdot 10^{-2}$ | $2.57 \cdot 10^{-3}$ | 7.85 x | $1.14 \cdot 10^{-2}$ | $2.58 \cdot 10^{-3}$ | 4.44 x |
| features.6 | $6.98 \cdot 10^{-3}$ | $1.67 \cdot 10^{-3}$ | 4.19 x | $5.17 \cdot 10^{-3}$ | $1.67 \cdot 10^{-3}$ | 3.10 x |
| features.8 | $8.16 \cdot 10^{-3}$ | $1.97 \cdot 10^{-3}$ | 4.15 x | $6.13 \cdot 10^{-3}$ | $1.97 \cdot 10^{-3}$ | 3.11 x |
| features.10 | $5.80 \cdot 10^{-3}$ | $1.47 \cdot 10^{-3}$ | 3.94 x | $4.34 \cdot 10^{-3}$ | $1.47 \cdot 10^{-3}$ | 2.95 x |

#### (e) ResNet18, input shape (32, 3, 256, 256)

| Name | TN [s] | PT [s] | Factor | TN + opt [s] | PT [s] | Factor |
|---|---|---|---|---|---|---|
| conv1 | $3.00 \cdot 10^{-2}$ | $8.23 \cdot 10^{-3}$ | 3.65 x | $2.99 \cdot 10^{-2}$ | $7.83 \cdot 10^{-3}$ | 3.82 x |
| layer1.0.conv1 | $2.34 \cdot 10^{-2}$ | $3.18 \cdot 10^{-3}$ | 7.37 x | $1.10 \cdot 10^{-2}$ | $3.22 \cdot 10^{-3}$ | 3.43 x |
| layer2.0.conv1 | $5.88 \cdot 10^{-3}$ | $2.97 \cdot 10^{-3}$ | 1.98 x | $5.87 \cdot 10^{-3}$ | $2.96 \cdot 10^{-3}$ | 1.98 x |
| layer2.0.conv2 | $1.17 \cdot 10^{-2}$ | $1.66 \cdot 10^{-3}$ | 7.03 x | $6.98 \cdot 10^{-3}$ | $1.66 \cdot 10^{-3}$ | 4.20 x |
| layer2.0.downsample.0 | $1.85 \cdot 10^{-3}$ | $7.39 \cdot 10^{-4}$ | 2.51 x | $7.60 \cdot 10^{-4}$ | $7.33 \cdot 10^{-4}$ | 1.04 x |
| layer3.0.conv1 | $3.79 \cdot 10^{-3}$ | $2.71 \cdot 10^{-3}$ | 1.40 x | $3.76 \cdot 10^{-3}$ | $2.71 \cdot 10^{-3}$ | 1.39 x |
| layer3.0.conv2 | $6.62 \cdot 10^{-3}$ | $1.48 \cdot 10^{-3}$ | 4.46 x | $4.87 \cdot 10^{-3}$ | $1.48 \cdot 10^{-3}$ | 3.28 x |
| layer3.0.downsample.0 | $1.61 \cdot 10^{-3}$ | $5.39 \cdot 10^{-4}$ | 2.99 x | $6.12 \cdot 10^{-4}$ | $5.45 \cdot 10^{-4}$ | 1.12 x |
| layer4.0.conv1 | $2.85 \cdot 10^{-3}$ | $2.46 \cdot 10^{-3}$ | 1.16 x | $2.82 \cdot 10^{-3}$ | $2.46 \cdot 10^{-3}$ | 1.15 x |
| layer4.0.conv2 | $4.83 \cdot 10^{-3}$ | $1.72 \cdot 10^{-3}$ | 2.80 x | $4.31 \cdot 10^{-3}$ | $1.72 \cdot 10^{-3}$ | 2.50 x |
| layer4.0.downsample.0 | $1.00 \cdot 10^{-3}$ | $1.02 \cdot 10^{-3}$ | **0.98 x** | $5.07 \cdot 10^{-4}$ | $1.02 \cdot 10^{-3}$ | **0.50 x** |

#### (f) ResNext101, input shape (32, 3, 256, 256)

| Name | TN [s] | PT [s] | Factor | TN + opt [s] | PT [s] | Factor |
|---|---|---|---|---|---|---|
| conv1 | $3.00 \cdot 10^{-2}$ | $8.22 \cdot 10^{-3}$ | 3.65 x | $2.99 \cdot 10^{-2}$ | $7.83 \cdot 10^{-3}$ | 3.82 x |
| layer1.0.conv1 | $7.08 \cdot 10^{-3}$ | $2.92 \cdot 10^{-3}$ | 2.42 x | $3.75 \cdot 10^{-3}$ | $2.89 \cdot 10^{-3}$ | 1.30 x |
| layer1.0.conv2 | $6.70 \cdot 10^{-2}$ | $2.53 \cdot 10^{-2}$ | 2.65 x | $6.72 \cdot 10^{-2}$ | $1.91 \cdot 10^{-2}$ | 3.51 x |
| layer1.0.conv3 | $3.12 \cdot 10^{-2}$ | $8.78 \cdot 10^{-3}$ | 3.55 x | $1.04 \cdot 10^{-2}$ | $1.02 \cdot 10^{-2}$ | 1.02 x |
| layer2.0.conv1 | $2.29 \cdot 10^{-2}$ | $1.80 \cdot 10^{-2}$ | 1.28 x | $1.77 \cdot 10^{-2}$ | $1.76 \cdot 10^{-2}$ | 1.01 x |
| layer2.0.conv2 | $6.64 \cdot 10^{-2}$ | $1.23 \cdot 10^{-2}$ | 5.40 x | $6.63 \cdot 10^{-2}$ | $1.23 \cdot 10^{-2}$ | 5.39 x |
| layer2.0.conv3 | $1.82 \cdot 10^{-2}$ | $5.90 \cdot 10^{-3}$ | 3.08 x | $8.44 \cdot 10^{-3}$ | $6.50 \cdot 10^{-3}$ | 1.30 x |
| layer2.0.downsample.0 | $8.57 \cdot 10^{-3}$ | $5.24 \cdot 10^{-3}$ | 1.64 x | $4.55 \cdot 10^{-3}$ | $5.24 \cdot 10^{-3}$ | **0.87 x** |
| layer2.1.conv2 | $4.04 \cdot 10^{-2}$ | $1.21 \cdot 10^{-2}$ | 3.33 x | $4.04 \cdot 10^{-2}$ | $1.22 \cdot 10^{-2}$ | 3.32 x |
| layer3.0.conv1 | $1.84 \cdot 10^{-2}$ | $2.03 \cdot 10^{-2}$ | **0.91 x** | $1.48 \cdot 10^{-2}$ | $2.03 \cdot 10^{-2}$ | **0.73 x** |
| layer3.0.conv2 | $1.63 \cdot 10^{-2}$ | $5.77 \cdot 10^{-3}$ | 2.83 x | $1.63 \cdot 10^{-2}$ | $5.82 \cdot 10^{-3}$ | 2.81 x |
| layer3.0.conv3 | $1.17 \cdot 10^{-2}$ | $1.07 \cdot 10^{-2}$ | 1.10 x | $7.19 \cdot 10^{-3}$ | $1.07 \cdot 10^{-2}$ | **0.67 x** |
| layer3.0.downsample.0 | $6.19 \cdot 10^{-3}$ | $5.95 \cdot 10^{-3}$ | 1.04 x | $3.85 \cdot 10^{-3}$ | $6.01 \cdot 10^{-3}$ | **0.64 x** |
| layer3.1.conv2 | $1.47 \cdot 10^{-2}$ | $3.17 \cdot 10^{-3}$ | 4.65 x | $1.47 \cdot 10^{-2}$ | $3.14 \cdot 10^{-3}$ | 4.67 x |
| layer4.0.conv1 | $1.55 \cdot 10^{-2}$ | $2.10 \cdot 10^{-2}$ | **0.74 x** | $1.33 \cdot 10^{-2}$ | $2.11 \cdot 10^{-2}$ | **0.63 x** |
| layer4.0.conv2 | $8.07 \cdot 10^{-3}$ | $3.13 \cdot 10^{-2}$ | **0.26 x** | $8.06 \cdot 10^{-3}$ | $3.13 \cdot 10^{-2}$ | **0.26 x** |
| layer4.0.conv3 | $8.23 \cdot 10^{-3}$ | $1.06 \cdot 10^{-2}$ | **0.78 x** | $6.75 \cdot 10^{-3}$ | $1.06 \cdot 10^{-2}$ | **0.63 x** |
| layer4.0.downsample.0 | $4.96 \cdot 10^{-3}$ | $5.94 \cdot 10^{-3}$ | **0.84 x** | $3.59 \cdot 10^{-3}$ | $5.99 \cdot 10^{-3}$ | **0.60 x** |
| layer4.1.conv2 | $6.63 \cdot 10^{-3}$ | $1.40 \cdot 10^{-3}$ | 4.72 x | $6.62 \cdot 10^{-3}$ | $1.45 \cdot 10^{-3}$ | 4.55 x |

#### (g) ConvNeXt-base, input shape (32, 3, 256, 256)

| Name | TN [s] | PT [s] | Factor | TN + opt [s] | PT [s] | Factor |
|---|---|---|---|---|---|---|
| features.0.0 | $5.93 \cdot 10^{-3}$ | $1.99 \cdot 10^{-3}$ | 2.98 x | $1.87 \cdot 10^{-3}$ | $1.97 \cdot 10^{-3}$ | **0.95 x** |
| features.1.0.block.0 | $2.53 \cdot 10^{-2}$ | $1.09 \cdot 10^{-2}$ | 2.33 x | $2.53 \cdot 10^{-2}$ | $1.09 \cdot 10^{-2}$ | 2.33 x |
| features.2.1 | $8.29 \cdot 10^{-3}$ | $4.53 \cdot 10^{-3}$ | 1.83 x | $4.32 \cdot 10^{-3}$ | $4.52 \cdot 10^{-3}$ | **0.96 x** |
| features.3.0.block.0 | $1.23 \cdot 10^{-2}$ | $5.85 \cdot 10^{-3}$ | 2.10 x | $1.22 \cdot 10^{-2}$ | $5.82 \cdot 10^{-3}$ | 2.10 x |
| features.4.1 | $5.74 \cdot 10^{-3}$ | $5.30 \cdot 10^{-3}$ | 1.08 x | $3.74 \cdot 10^{-3}$ | $5.29 \cdot 10^{-3}$ | **0.71 x** |
| features.5.0.block.0 | $6.05 \cdot 10^{-3}$ | $3.63 \cdot 10^{-3}$ | 1.66 x | $6.03 \cdot 10^{-3}$ | $3.64 \cdot 10^{-3}$ | 1.66 x |
| features.6.1 | $4.74 \cdot 10^{-3}$ | $5.28 \cdot 10^{-3}$ | **0.90 x** | $3.53 \cdot 10^{-3}$ | $5.17 \cdot 10^{-3}$ | **0.68 x** |
| features.7.0.block.0 | $9.08 \cdot 10^{-4}$ | $3.13 \cdot 10^{-3}$ | **0.29 x** | $8.87 \cdot 10^{-4}$ | $3.13 \cdot 10^{-3}$ | **0.28 x** |

(h) InceptionV3, input shape (32, 3, 299, 299)

| Name | TN [s] | PT [s] | Factor | TN + opt [s] | PT [s] | Factor |
|---|---|---|---|---|---|---|
| Conv2d_1a_3x3.conv | $1.07 \cdot 10^{-2}$ | $1.70 \cdot 10^{-3}$ | 6.31 x | $1.07 \cdot 10^{-2}$ | $1.70 \cdot 10^{-3}$ | 6.29 x |
| Conv2d_2a_3x3.conv | $6.00 \cdot 10^{-2}$ | $1.16 \cdot 10^{-2}$ | 5.18 x | $3.11 \cdot 10^{-2}$ | $1.16 \cdot 10^{-2}$ | 2.68 x |
| Conv2d_2b_3x3.conv | $6.10 \cdot 10^{-2}$ | $1.34 \cdot 10^{-2}$ | 4.55 x | $4.27 \cdot 10^{-2}$ | $1.53 \cdot 10^{-2}$ | 2.78 x |
| Conv2d_3b_1x1.conv | $5.48 \cdot 10^{-3}$ | $1.82 \cdot 10^{-3}$ | 3.01 x | $2.26 \cdot 10^{-3}$ | $2.12 \cdot 10^{-3}$ | 1.07 x |
| Conv2d_4a_3x3.conv | $5.28 \cdot 10^{-2}$ | $1.29 \cdot 10^{-2}$ | 4.08 x | $3.39 \cdot 10^{-2}$ | $1.29 \cdot 10^{-2}$ | 2.62 x |
| Mixed_5b.branch1x1.conv | $5.14 \cdot 10^{-3}$ | $1.16 \cdot 10^{-3}$ | 4.43 x | $1.41 \cdot 10^{-3}$ | $1.48 \cdot 10^{-3}$ | **0.95 x** |
| Mixed_5b.branch5x5_1.conv | $4.92 \cdot 10^{-3}$ | $1.46 \cdot 10^{-3}$ | 3.37 x | $1.39 \cdot 10^{-3}$ | $1.47 \cdot 10^{-3}$ | **0.95 x** |
| Mixed_5b.branch5x5_2.conv | $9.28 \cdot 10^{-3}$ | $1.28 \cdot 10^{-3}$ | 7.23 x | $4.83 \cdot 10^{-3}$ | $1.28 \cdot 10^{-3}$ | 3.77 x |
| Mixed_5b.branch3x3dbl_2.conv | $7.78 \cdot 10^{-3}$ | $1.75 \cdot 10^{-3}$ | 4.45 x | $4.22 \cdot 10^{-3}$ | $1.75 \cdot 10^{-3}$ | 2.41 x |
| Mixed_5b.branch3x3dbl_3.conv | $1.05 \cdot 10^{-2}$ | $1.86 \cdot 10^{-3}$ | 5.63 x | $6.00 \cdot 10^{-3}$ | $1.87 \cdot 10^{-3}$ | 3.21 x |
| Mixed_5b.branch_pool.conv | $4.52 \cdot 10^{-3}$ | $9.10 \cdot 10^{-4}$ | 4.97 x | $1.16 \cdot 10^{-3}$ | $8.96 \cdot 10^{-4}$ | 1.30 x |
| Mixed_5c.branch1x1.conv | $6.55 \cdot 10^{-3}$ | $2.00 \cdot 10^{-3}$ | 3.27 x | $1.67 \cdot 10^{-3}$ | $1.93 \cdot 10^{-3}$ | **0.86 x** |
| Mixed_5c.branch5x5_1.conv | $6.33 \cdot 10^{-3}$ | $1.93 \cdot 10^{-3}$ | 3.28 x | $1.64 \cdot 10^{-3}$ | $1.86 \cdot 10^{-3}$ | **0.88 x** |
| Mixed_5d.branch1x1.conv | $7.46 \cdot 10^{-3}$ | $2.34 \cdot 10^{-3}$ | 3.19 x | $2.03 \cdot 10^{-3}$ | $2.31 \cdot 10^{-3}$ | **0.88 x** |
| Mixed_5d.branch5x5_1.conv | $7.24 \cdot 10^{-3}$ | $2.16 \cdot 10^{-3}$ | 3.36 x | $2.00 \cdot 10^{-3}$ | $2.15 \cdot 10^{-3}$ | **0.93 x** |
| Mixed_6a.branch3x3 | $1.10 \cdot 10^{-2}$ | $8.34 \cdot 10^{-3}$ | 1.32 x | $1.09 \cdot 10^{-2}$ | $8.34 \cdot 10^{-3}$ | 1.31 x |
| Mixed_6a.branch3x3dbl_3.conv | $2.46 \cdot 10^{-3}$ | $1.12 \cdot 10^{-3}$ | 2.20 x | $2.42 \cdot 10^{-3}$ | $1.12 \cdot 10^{-3}$ | 2.17 x |
| Mixed_6b.branch1x1.conv | $5.17 \cdot 10^{-3}$ | $2.05 \cdot 10^{-3}$ | 2.52 x | $1.86 \cdot 10^{-3}$ | $2.07 \cdot 10^{-3}$ | **0.90 x** |
| Mixed_6b.branch7x7_1.conv | $4.63 \cdot 10^{-3}$ | $1.56 \cdot 10^{-3}$ | 2.96 x | $1.46 \cdot 10^{-3}$ | $1.61 \cdot 10^{-3}$ | **0.91 x** |
| Mixed_6b.branch7x7_2.conv | $3.09 \cdot 10^{-3}$ | $1.59 \cdot 10^{-3}$ | 1.94 x | $2.01 \cdot 10^{-3}$ | $1.64 \cdot 10^{-3}$ | 1.22 x |
| Mixed_6b.branch7x7_3.conv | $3.44 \cdot 10^{-3}$ | $2.29 \cdot 10^{-3}$ | 1.50 x | $2.37 \cdot 10^{-3}$ | $2.33 \cdot 10^{-3}$ | 1.01 x |
| Mixed_6b.branch7x7dbl_2.conv | $4.06 \cdot 10^{-3}$ | $1.64 \cdot 10^{-3}$ | 2.48 x | $1.81 \cdot 10^{-3}$ | $1.68 \cdot 10^{-3}$ | 1.08 x |
| Mixed_6b.branch7x7dbl_5.conv | $2.51 \cdot 10^{-3}$ | $2.24 \cdot 10^{-3}$ | 1.12 x | $2.85 \cdot 10^{-3}$ | $2.29 \cdot 10^{-3}$ | 1.24 x |
| Mixed_6c.branch7x7_1.conv | $4.99 \cdot 10^{-3}$ | $1.98 \cdot 10^{-3}$ | 2.53 x | $1.77 \cdot 10^{-3}$ | $2.03 \cdot 10^{-3}$ | **0.87 x** |
| Mixed_6c.branch7x7_2.conv | $4.87 \cdot 10^{-3}$ | $2.71 \cdot 10^{-3}$ | 1.80 x | $3.30 \cdot 10^{-3}$ | $2.75 \cdot 10^{-3}$ | 1.20 x |
| Mixed_6c.branch7x7_3.conv | $4.85 \cdot 10^{-3}$ | $2.84 \cdot 10^{-3}$ | 1.71 x | $2.99 \cdot 10^{-3}$ | $2.87 \cdot 10^{-3}$ | 1.04 x |
| Mixed_6c.branch7x7dbl_2.conv | $5.43 \cdot 10^{-3}$ | $2.80 \cdot 10^{-3}$ | 1.94 x | $2.95 \cdot 10^{-3}$ | $2.80 \cdot 10^{-3}$ | 1.05 x |
| Mixed_6c.branch7x7dbl_5.conv | $3.20 \cdot 10^{-3}$ | $2.78 \cdot 10^{-3}$ | 1.15 x | $3.41 \cdot 10^{-3}$ | $2.82 \cdot 10^{-3}$ | 1.21 x |
| Mixed_6e.branch7x7_2.conv | $5.96 \cdot 10^{-3}$ | $3.19 \cdot 10^{-3}$ | 1.87 x | $3.83 \cdot 10^{-3}$ | $3.24 \cdot 10^{-3}$ | 1.18 x |
| Mixed_6e.branch7x7_3.conv | $6.50 \cdot 10^{-3}$ | $3.26 \cdot 10^{-3}$ | 1.99 x | $3.40 \cdot 10^{-3}$ | $3.30 \cdot 10^{-3}$ | 1.03 x |
| AuxLogits.conv0.conv | $6.48 \cdot 10^{-4}$ | $3.45 \cdot 10^{-4}$ | 1.87 x | $3.45 \cdot 10^{-4}$ | $3.88 \cdot 10^{-4}$ | **0.89 x** |
| AuxLogits.conv1.conv | $5.34 \cdot 10^{-4}$ | $2.09 \cdot 10^{-4}$ | 2.56 x | $4.59 \cdot 10^{-4}$ | $2.76 \cdot 10^{-4}$ | 1.66 x |
| Mixed_7a.branch3x3_2.conv | $1.80 \cdot 10^{-3}$ | $5.61 \cdot 10^{-4}$ | 3.22 x | $1.78 \cdot 10^{-3}$ | $6.16 \cdot 10^{-4}$ | 2.90 x |
| Mixed_7a.branch7x7x3_4.conv | $1.55 \cdot 10^{-3}$ | $8.46 \cdot 10^{-4}$ | 1.83 x | $1.52 \cdot 10^{-3}$ | $8.50 \cdot 10^{-4}$ | 1.79 x |
| Mixed_7b.branch1x1.conv | $2.08 \cdot 10^{-3}$ | $1.62 \cdot 10^{-3}$ | 1.28 x | $1.07 \cdot 10^{-3}$ | $1.63 \cdot 10^{-3}$ | **0.66 x** |
| Mixed_7b.branch3x3_1.conv | $2.19 \cdot 10^{-3}$ | $1.65 \cdot 10^{-3}$ | 1.33 x | $1.17 \cdot 10^{-3}$ | $1.65 \cdot 10^{-3}$ | **0.71 x** |
| Mixed_7b.branch3x3_2a.conv | $1.56 \cdot 10^{-3}$ | $1.47 \cdot 10^{-3}$ | 1.06 x | $1.20 \cdot 10^{-3}$ | $1.47 \cdot 10^{-3}$ | **0.82 x** |
| Mixed_7b.branch3x3_2b.conv | $1.66 \cdot 10^{-3}$ | $1.50 \cdot 10^{-3}$ | 1.11 x | $1.22 \cdot 10^{-3}$ | $1.50 \cdot 10^{-3}$ | **0.82 x** |
| Mixed_7b.branch3x3dbl_1.conv | $2.34 \cdot 10^{-3}$ | $1.65 \cdot 10^{-3}$ | 1.42 x | $1.33 \cdot 10^{-3}$ | $1.66 \cdot 10^{-3}$ | **0.80 x** |
| Mixed_7b.branch3x3dbl_2.conv | $3.55 \cdot 10^{-3}$ | $1.23 \cdot 10^{-3}$ | 2.90 x | $3.10 \cdot 10^{-3}$ | $1.26 \cdot 10^{-3}$ | 2.45 x |
| Mixed_7b.branch_pool.conv | $1.84 \cdot 10^{-3}$ | $1.46 \cdot 10^{-3}$ | 1.26 x | $8.67 \cdot 10^{-4}$ | $1.46 \cdot 10^{-3}$ | **0.59 x** |
| Mixed_7c.branch1x1.conv | $3.07 \cdot 10^{-3}$ | $3.08 \cdot 10^{-3}$ | **1.00 x** | $1.55 \cdot 10^{-3}$ | $3.12 \cdot 10^{-3}$ | **0.50 x** |
| Mixed_7c.branch3x3_1.conv | $3.30 \cdot 10^{-3}$ | $3.11 \cdot 10^{-3}$ | 1.06 x | $1.79 \cdot 10^{-3}$ | $3.11 \cdot 10^{-3}$ | **0.58 x** |
| Mixed_7c.branch3x3dbl_1.conv | $3.56 \cdot 10^{-3}$ | $3.11 \cdot 10^{-3}$ | 1.15 x | $2.03 \cdot 10^{-3}$ | $3.10 \cdot 10^{-3}$ | **0.65 x** |
| Mixed_7c.branch_pool.conv | $2.70 \cdot 10^{-3}$ | $1.61 \cdot 10^{-3}$ | 1.68 x | $1.22 \cdot 10^{-3}$ | $1.61 \cdot 10^{-3}$ | **0.76 x** |

(i) MobileNetV2, input shape (32, 3, 256, 256)

| Name | TN [s] | PT [s] | Factor | TN + opt [s] | PT [s] | Factor |
|---|---|---|---|---|---|---|
| features.0.0 | $7.70 \cdot 10^{-3}$ | $1.45 \cdot 10^{-3}$ | 5.30 x | $7.67 \cdot 10^{-3}$ | $1.46 \cdot 10^{-3}$ | 5.26 x |
| features.1.conv.0.0 | $1.59 \cdot 10^{-2}$ | $2.46 \cdot 10^{-3}$ | 6.48 x | $1.59 \cdot 10^{-2}$ | $2.47 \cdot 10^{-3}$ | 6.46 x |
| features.1.conv.1 | $1.47 \cdot 10^{-2}$ | $2.40 \cdot 10^{-3}$ | 6.11 x | $2.64 \cdot 10^{-3}$ | $2.39 \cdot 10^{-3}$ | 1.10 x |
| features.2.conv.0.0 | $8.97 \cdot 10^{-3}$ | $4.95 \cdot 10^{-3}$ | 1.81 x | $4.22 \cdot 10^{-3}$ | $4.95 \cdot 10^{-3}$ | **0.85 x** |
| features.2.conv.1.0 | $2.14 \cdot 10^{-2}$ | $2.40 \cdot 10^{-3}$ | 8.90 x | $2.14 \cdot 10^{-2}$ | $2.41 \cdot 10^{-3}$ | 8.89 x |
| features.2.conv.2 | $7.38 \cdot 10^{-3}$ | $1.96 \cdot 10^{-3}$ | 3.76 x | $1.91 \cdot 10^{-3}$ | $1.89 \cdot 10^{-3}$ | 1.01 x |
| features.3.conv.0.0 | $3.55 \cdot 10^{-3}$ | $2.24 \cdot 10^{-3}$ | 1.58 x | $1.83 \cdot 10^{-3}$ | $2.19 \cdot 10^{-3}$ | **0.84 x** |
| features.3.conv.1.0 | $1.34 \cdot 10^{-2}$ | $2.49 \cdot 10^{-3}$ | 5.38 x | $1.34 \cdot 10^{-2}$ | $2.46 \cdot 10^{-3}$ | 5.43 x |
| features.3.conv.2 | $1.04 \cdot 10^{-2}$ | $2.89 \cdot 10^{-3}$ | 3.61 x | $2.89 \cdot 10^{-3}$ | $2.85 \cdot 10^{-3}$ | 1.01 x |
| features.4.conv.1.0 | $7.46 \cdot 10^{-3}$ | $1.01 \cdot 10^{-3}$ | 7.41 x | $7.44 \cdot 10^{-3}$ | $9.84 \cdot 10^{-4}$ | 7.56 x |
| features.4.conv.2 | $3.00 \cdot 10^{-3}$ | $9.40 \cdot 10^{-4}$ | 3.19 x | $9.20 \cdot 10^{-4}$ | $8.94 \cdot 10^{-4}$ | 1.03 x |
| features.5.conv.0.0 | $1.49 \cdot 10^{-3}$ | $7.46 \cdot 10^{-4}$ | 2.00 x | $7.48 \cdot 10^{-4}$ | $7.46 \cdot 10^{-4}$ | 1.00 x |
| features.5.conv.1.0 | $4.77 \cdot 10^{-3}$ | $9.10 \cdot 10^{-4}$ | 5.24 x | $4.75 \cdot 10^{-3}$ | $8.88 \cdot 10^{-4}$ | 5.34 x |
| features.5.conv.2 | $3.62 \cdot 10^{-3}$ | $1.06 \cdot 10^{-3}$ | 3.41 x | $1.04 \cdot 10^{-3}$ | $1.01 \cdot 10^{-3}$ | 1.03 x |
| features.7.conv.1.0 | $2.61 \cdot 10^{-3}$ | $3.66 \cdot 10^{-4}$ | 7.13 x | $2.60 \cdot 10^{-3}$ | $3.65 \cdot 10^{-4}$ | 7.13 x |
| features.7.conv.2 | $1.43 \cdot 10^{-3}$ | $5.55 \cdot 10^{-4}$ | 2.58 x | $4.64 \cdot 10^{-4}$ | $5.55 \cdot 10^{-4}$ | **0.84 x** |
| features.8.conv.0.0 | $1.14 \cdot 10^{-3}$ | $5.36 \cdot 10^{-4}$ | 2.12 x | $5.22 \cdot 10^{-4}$ | $5.34 \cdot 10^{-4}$ | **0.98 x** |
| features.8.conv.1.0 | $2.44 \cdot 10^{-3}$ | $5.67 \cdot 10^{-4}$ | 4.31 x | $2.43 \cdot 10^{-3}$ | $5.68 \cdot 10^{-4}$ | 4.28 x |
| features.8.conv.2 | $2.23 \cdot 10^{-3}$ | $8.32 \cdot 10^{-4}$ | 2.68 x | $6.80 \cdot 10^{-4}$ | $8.82 \cdot 10^{-4}$ | **0.77 x** |
| features.11.conv.2 | $2.36 \cdot 10^{-3}$ | $8.68 \cdot 10^{-4}$ | 2.72 x | $7.89 \cdot 10^{-4}$ | $8.69 \cdot 10^{-4}$ | **0.91 x** |
| features.12.conv.0.0 | $1.55 \cdot 10^{-3}$ | $1.08 \cdot 10^{-3}$ | 1.44 x | $9.00 \cdot 10^{-4}$ | $1.03 \cdot 10^{-3}$ | **0.88 x** |
| features.12.conv.1.0 | $3.52 \cdot 10^{-3}$ | $8.20 \cdot 10^{-4}$ | 4.29 x | $3.50 \cdot 10^{-3}$ | $8.19 \cdot 10^{-4}$ | 4.27 x |
| features.12.conv.2 | $3.27 \cdot 10^{-3}$ | $1.26 \cdot 10^{-3}$ | 2.59 x | $1.10 \cdot 10^{-3}$ | $1.26 \cdot 10^{-3}$ | **0.87 x** |
| features.14.conv.1.0 | $2.07 \cdot 10^{-3}$ | $3.90 \cdot 10^{-4}$ | 5.31 x | $2.05 \cdot 10^{-3}$ | $3.90 \cdot 10^{-4}$ | 5.26 x |
| features.14.conv.2 | $1.06 \cdot 10^{-3}$ | $1.39 \cdot 10^{-3}$ | **0.76 x** | $5.10 \cdot 10^{-4}$ | $1.40 \cdot 10^{-3}$ | **0.36 x** |
| features.15.conv.0.0 | $1.12 \cdot 10^{-3}$ | $7.19 \cdot 10^{-4}$ | 1.56 x | $6.21 \cdot 10^{-4}$ | $7.10 \cdot 10^{-4}$ | **0.87 x** |
| features.15.conv.1.0 | $2.31 \cdot 10^{-3}$ | $5.96 \cdot 10^{-4}$ | 3.87 x | $2.28 \cdot 10^{-3}$ | $5.96 \cdot 10^{-4}$ | 3.83 x |
| features.15.conv.2 | $1.34 \cdot 10^{-3}$ | $1.41 \cdot 10^{-3}$ | **0.95 x** | $6.53 \cdot 10^{-4}$ | $1.40 \cdot 10^{-3}$ | **0.47 x** |
| features.17.conv.2 | $1.59 \cdot 10^{-3}$ | $1.67 \cdot 10^{-3}$ | **0.95 x** | $8.70 \cdot 10^{-4}$ | $1.62 \cdot 10^{-3}$ | **0.54 x** |
| features.18.0 | $1.53 \cdot 10^{-3}$ | $1.68 \cdot 10^{-3}$ | **0.91 x** | $1.04 \cdot 10^{-3}$ | $1.63 \cdot 10^{-3}$ | **0.64 x** |

### F.5 KFC Factor (KFAC-expand)

We compare TN and TN+opt with a PyTorch implementation of the input-based KFC factor based on `torch.nn.functional.unfold`. Figure F22 visualizes the performance ratios for different convolution categories. Table F8 contains the detailed run times and performance factors.

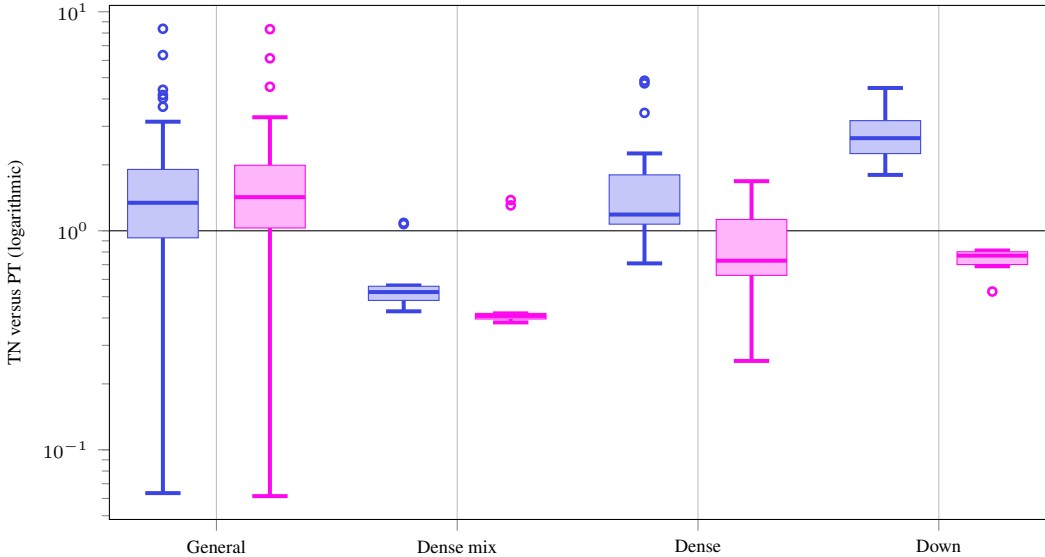

Figure F22: KFC/KFAC-expand factor performance ratios of TN versus PT and TN+opt versus PT for different convolution types on GPU.

Table F8: KFC (KFAC-expand) factor performance comparison on GPU.

(a) 3c3d, CIFAR-10, input shape (128, 3, 32, 32)

| Name | TN [s] | PT [s] | Factor | TN + opt [s] | PT [s] | Factor |
|---|---|---|---|---|---|---|
| conv1.0 | $1.03 \cdot 10^{-3}$ | $2.42 \cdot 10^{-3}$ | **0.43 x** | $1.03 \cdot 10^{-3}$ | $2.52 \cdot 10^{-3}$ | **0.41 x** |
| conv2.0 | $6.69 \cdot 10^{-3}$ | $3.83 \cdot 10^{-3}$ | 1.75 x | $6.97 \cdot 10^{-3}$ | $4.52 \cdot 10^{-3}$ | 1.54 x |
| conv3.1 | $3.27 \cdot 10^{-3}$ | $2.38 \cdot 10^{-3}$ | 1.37 x | $3.53 \cdot 10^{-3}$ | $2.54 \cdot 10^{-3}$ | 1.39 x |

(b) F-MNIST 2c2d, input shape (128, 1, 28, 28)

| Name | TN [s] | PT [s] | Factor | TN + opt [s] | PT [s] | Factor |
|---|---|---|---|---|---|---|
| conv1.1 | $1.22 \cdot 10^{-3}$ | $2.01 \cdot 10^{-3}$ | **0.61 x** | $9.30 \cdot 10^{-4}$ | $1.72 \cdot 10^{-3}$ | **0.54 x** |
| conv2.1 | $1.03 \cdot 10^{-2}$ | $9.54 \cdot 10^{-3}$ | 1.08 x | $1.02 \cdot 10^{-2}$ | $9.47 \cdot 10^{-3}$ | 1.08 x |

(c) CIFAR-100 All-CNN-C, input shape (128, 3, 32, 32)

| Name | TN [s] | PT [s] | Factor | TN + opt [s] | PT [s] | Factor |
|---|---|---|---|---|---|---|
| conv1.1 | $1.37 \cdot 10^{-3}$ | $1.48 \cdot 10^{-3}$ | **0.93 x** | $2.72 \cdot 10^{-3}$ | $2.11 \cdot 10^{-3}$ | 1.29 x |
| conv2.1 | $1.48 \cdot 10^{-1}$ | $6.95 \cdot 10^{-2}$ | 2.13 x | $1.53 \cdot 10^{-1}$ | $7.14 \cdot 10^{-2}$ | 2.15 x |
| conv3.1 | $4.77 \cdot 10^{-2}$ | $1.15 \cdot 10^{-2}$ | 4.17 x | $4.56 \cdot 10^{-2}$ | $1.38 \cdot 10^{-2}$ | 3.30 x |
| conv4.1 | $2.32 \cdot 10^{-2}$ | $1.14 \cdot 10^{-2}$ | 2.03 x | $2.25 \cdot 10^{-2}$ | $1.14 \cdot 10^{-2}$ | 1.98 x |
| conv5.1 | $7.03 \cdot 10^{-2}$ | $5.82 \cdot 10^{-2}$ | 1.21 x | $1.01 \cdot 10^{-1}$ | $6.19 \cdot 10^{-2}$ | 1.63 x |
| conv6.1 | $2.84 \cdot 10^{-2}$ | $1.33 \cdot 10^{-2}$ | 2.14 x | $2.83 \cdot 10^{-2}$ | $9.77 \cdot 10^{-3}$ | 2.90 x |
| conv7.0 | $8.68 \cdot 10^{-3}$ | $5.95 \cdot 10^{-3}$ | 1.46 x | $9.30 \cdot 10^{-3}$ | $6.01 \cdot 10^{-3}$ | 1.55 x |
| conv8.1 | $1.03 \cdot 10^{-3}$ | $9.97 \cdot 10^{-4}$ | 1.03 x | $3.67 \cdot 10^{-4}$ | $1.44 \cdot 10^{-3}$ | **0.25 x** |
| conv9.1 | $1.06 \cdot 10^{-3}$ | $1.49 \cdot 10^{-3}$ | **0.71 x** | $4.61 \cdot 10^{-4}$ | $1.56 \cdot 10^{-3}$ | **0.30 x** |

(d) Alexnet, input shape (32, 3, 256, 256)

| Name | TN [s] | PT [s] | Factor | TN + opt [s] | PT [s] | Factor |
|---|---|---|---|---|---|---|
| features.0 | $5.45 \cdot 10^{-2}$ | $1.35 \cdot 10^{-2}$ | 4.03 x | $6.09 \cdot 10^{-2}$ | $1.34 \cdot 10^{-2}$ | 4.55 x |
| features.3 | $4.57 \cdot 10^{-2}$ | $4.14 \cdot 10^{-2}$ | 1.10 x | $5.31 \cdot 10^{-2}$ | $3.73 \cdot 10^{-2}$ | 1.42 x |
| features.6 | $8.63 \cdot 10^{-3}$ | $7.86 \cdot 10^{-3}$ | 1.10 x | $1.12 \cdot 10^{-2}$ | $8.74 \cdot 10^{-3}$ | 1.28 x |
| features.8 | $3.76 \cdot 10^{-2}$ | $4.10 \cdot 10^{-2}$ | **0.92 x** | $4.57 \cdot 10^{-2}$ | $4.33 \cdot 10^{-2}$ | 1.06 x |
| features.10 | $1.52 \cdot 10^{-2}$ | $1.38 \cdot 10^{-2}$ | 1.10 x | $1.91 \cdot 10^{-2}$ | $1.47 \cdot 10^{-2}$ | 1.30 x |

(e) ResNet18, input shape (32, 3, 256, 256)

| Name | TN [s] | PT [s] | Factor | TN + opt [s] | PT [s] | Factor |
|---|---|---|---|---|---|---|
| conv1 | $5.25 \cdot 10^{-2}$ | $2.07 \cdot 10^{-2}$ | 2.54 x | $5.22 \cdot 10^{-2}$ | $2.06 \cdot 10^{-2}$ | 2.54 x |
| layer1.0.conv1 | $4.36 \cdot 10^{-2}$ | $2.98 \cdot 10^{-2}$ | 1.46 x | $5.57 \cdot 10^{-2}$ | $3.02 \cdot 10^{-2}$ | 1.84 x |
| layer2.0.conv1 | $2.81 \cdot 10^{-2}$ | $6.38 \cdot 10^{-3}$ | 4.41 x | $2.78 \cdot 10^{-2}$ | $1.23 \cdot 10^{-2}$ | 2.25 x |
| layer2.0.conv2 | $2.56 \cdot 10^{-2}$ | $1.90 \cdot 10^{-2}$ | 1.34 x | $3.09 \cdot 10^{-2}$ | $2.01 \cdot 10^{-2}$ | 1.53 x |
| layer2.0.downsample.0 | $3.66 \cdot 10^{-3}$ | $8.14 \cdot 10^{-4}$ | 4.49 x | $6.45 \cdot 10^{-4}$ | $7.94 \cdot 10^{-4}$ | **0.81 x** |
| layer3.0.conv1 | $1.34 \cdot 10^{-2}$ | $9.19 \cdot 10^{-3}$ | 1.46 x | $1.40 \cdot 10^{-2}$ | $9.17 \cdot 10^{-3}$ | 1.53 x |
| layer3.0.conv2 | $1.90 \cdot 10^{-2}$ | $1.84 \cdot 10^{-2}$ | 1.03 x | $2.25 \cdot 10^{-2}$ | $1.95 \cdot 10^{-2}$ | 1.16 x |
| layer3.0.downsample.0 | $1.98 \cdot 10^{-3}$ | $7.00 \cdot 10^{-4}$ | 2.83 x | $4.59 \cdot 10^{-4}$ | $5.72 \cdot 10^{-4}$ | **0.80 x** |
| layer4.0.conv1 | $8.65 \cdot 10^{-3}$ | $4.79 \cdot 10^{-3}$ | 1.81 x | $9.12 \cdot 10^{-3}$ | $4.60 \cdot 10^{-3}$ | 1.98 x |
| layer4.0.conv2 | $2.48 \cdot 10^{-2}$ | $1.63 \cdot 10^{-2}$ | 1.52 x | $2.49 \cdot 10^{-2}$ | $1.88 \cdot 10^{-2}$ | 1.32 x |
| layer4.0.downsample.0 | $1.19 \cdot 10^{-3}$ | $5.45 \cdot 10^{-4}$ | 2.18 x | $2.88 \cdot 10^{-4}$ | $5.45 \cdot 10^{-4}$ | **0.53 x** |

(f) ResNext101, input shape (32, 3, 256, 256)

| Name | TN [s] | PT [s] | Factor | TN + opt [s] | PT [s] | Factor |
|---|---|---|---|---|---|---|
| conv1 | $5.13 \cdot 10^{-2}$ | $2.05 \cdot 10^{-2}$ | 2.50 x | $5.06 \cdot 10^{-2}$ | $2.05 \cdot 10^{-2}$ | 2.46 x |
| layer1.0.conv1 | $3.33 \cdot 10^{-3}$ | $1.85 \cdot 10^{-3}$ | 1.80 x | $1.70 \cdot 10^{-3}$ | $2.08 \cdot 10^{-3}$ | **0.82 x** |
| layer1.0.conv2 | $1.09 \cdot 10^{-1}$ | $6.60 \cdot 10^{-2}$ | 1.66 x | $1.11 \cdot 10^{-1}$ | $8.12 \cdot 10^{-2}$ | 1.37 x |
| layer1.0.conv3 | $1.60 \cdot 10^{-2}$ | $7.49 \cdot 10^{-3}$ | 2.14 x | $1.04 \cdot 10^{-2}$ | $7.52 \cdot 10^{-3}$ | 1.39 x |
| layer2.0.conv1 | $1.60 \cdot 10^{-2}$ | $1.53 \cdot 10^{-2}$ | 1.05 x | $1.04 \cdot 10^{-2}$ | $1.53 \cdot 10^{-2}$ | **0.68 x** |
| layer2.0.conv2 | $1.40 \cdot 10^{-1}$ | $4.44 \cdot 10^{-2}$ | 3.15 x | $1.44 \cdot 10^{-1}$ | $4.54 \cdot 10^{-2}$ | 3.18 x |
| layer2.0.conv3 | $1.14 \cdot 10^{-2}$ | $5.19 \cdot 10^{-3}$ | 2.20 x | $8.41 \cdot 10^{-3}$ | $5.20 \cdot 10^{-3}$ | 1.62 x |
| layer2.0.downsample.0 | $1.40 \cdot 10^{-2}$ | $4.22 \cdot 10^{-3}$ | 3.30 x | $2.92 \cdot 10^{-3}$ | $4.24 \cdot 10^{-3}$ | **0.69 x** |
| layer2.1.conv2 | $5.07 \cdot 10^{-2}$ | $4.23 \cdot 10^{-2}$ | 1.20 x | $5.07 \cdot 10^{-2}$ | $4.24 \cdot 10^{-2}$ | 1.19 x |
| layer3.0.conv1 | $1.14 \cdot 10^{-2}$ | $5.21 \cdot 10^{-3}$ | 2.19 x | $8.42 \cdot 10^{-3}$ | $5.27 \cdot 10^{-3}$ | 1.60 x |
| layer3.0.conv2 | $6.11 \cdot 10^{-2}$ | $3.21 \cdot 10^{-2}$ | 1.90 x | $6.23 \cdot 10^{-2}$ | $2.92 \cdot 10^{-2}$ | 2.14 x |
| layer3.0.conv3 | $8.77 \cdot 10^{-3}$ | $4.30 \cdot 10^{-3}$ | 2.04 x | $7.17 \cdot 10^{-3}$ | $4.33 \cdot 10^{-3}$ | 1.66 x |
| layer3.0.downsample.0 | $7.59 \cdot 10^{-3}$ | $3.08 \cdot 10^{-3}$ | 2.47 x | $2.28 \cdot 10^{-3}$ | $3.08 \cdot 10^{-3}$ | **0.74 x** |
| layer3.1.conv2 | $2.12 \cdot 10^{-2}$ | $1.95 \cdot 10^{-2}$ | 1.09 x | $2.05 \cdot 10^{-2}$ | $1.99 \cdot 10^{-2}$ | 1.03 x |
| layer4.0.conv1 | $8.75 \cdot 10^{-3}$ | $4.15 \cdot 10^{-3}$ | 2.11 x | $7.18 \cdot 10^{-3}$ | $4.26 \cdot 10^{-3}$ | 1.68 x |
| layer4.0.conv2 | $4.70 \cdot 10^{-2}$ | $2.47 \cdot 10^{-2}$ | 1.91 x | $4.71 \cdot 10^{-2}$ | $2.47 \cdot 10^{-2}$ | 1.91 x |
| layer4.0.conv3 | $7.88 \cdot 10^{-3}$ | $7.66 \cdot 10^{-3}$ | 1.03 x | $6.74 \cdot 10^{-3}$ | $7.67 \cdot 10^{-3}$ | **0.88 x** |
| layer4.0.downsample.0 | $4.54 \cdot 10^{-3}$ | $2.52 \cdot 10^{-3}$ | 1.80 x | $2.03 \cdot 10^{-3}$ | $2.54 \cdot 10^{-3}$ | **0.80 x** |
| layer4.1.conv2 | $1.36 \cdot 10^{-2}$ | $1.16 \cdot 10^{-2}$ | 1.16 x | $1.36 \cdot 10^{-2}$ | $1.17 \cdot 10^{-2}$ | 1.16 x |

(g) ConvNeXt-base, input shape (32, 3, 256, 256)

| Name | TN [s] | PT [s] | Factor | TN + opt [s] | PT [s] | Factor |
|---|---|---|---|---|---|---|
| features.0.0 | $9.94 \cdot 10^{-3}$ | $2.11 \cdot 10^{-3}$ | 4.71 x | $1.18 \cdot 10^{-3}$ | $2.11 \cdot 10^{-3}$ | **0.56 x** |
| features.1.0.block.0 | $4.09 \cdot 10^{-2}$ | $1.37 \cdot 10^{-1}$ | **0.30 x** | $5.25 \cdot 10^{-2}$ | $1.42 \cdot 10^{-1}$ | **0.37 x** |
| features.2.1 | $2.37 \cdot 10^{-2}$ | $4.90 \cdot 10^{-3}$ | 4.85 x | $7.81 \cdot 10^{-3}$ | $4.93 \cdot 10^{-3}$ | 1.59 x |
| features.3.0.block.0 | $1.61 \cdot 10^{-2}$ | $6.99 \cdot 10^{-2}$ | **0.23 x** | $1.57 \cdot 10^{-2}$ | $7.12 \cdot 10^{-2}$ | **0.22 x** |
| features.4.1 | $1.41 \cdot 10^{-2}$ | $4.08 \cdot 10^{-3}$ | 3.45 x | $6.88 \cdot 10^{-3}$ | $4.15 \cdot 10^{-3}$ | 1.66 x |
| features.5.0.block.0 | $3.98 \cdot 10^{-3}$ | $3.35 \cdot 10^{-2}$ | **0.12 x** | $3.96 \cdot 10^{-3}$ | $3.43 \cdot 10^{-2}$ | **0.12 x** |
| features.6.1 | $6.82 \cdot 10^{-3}$ | $3.30 \cdot 10^{-3}$ | 2.06 x | $4.77 \cdot 10^{-3}$ | $3.31 \cdot 10^{-3}$ | 1.44 x |
| features.7.0.block.0 | $1.02 \cdot 10^{-3}$ | $1.61 \cdot 10^{-2}$ | **0.06 x** | $1.00 \cdot 10^{-3}$ | $1.63 \cdot 10^{-2}$ | **0.06 x** |

(h) InceptionV3, input shape (32, 3, 299, 299)

| Name | TN [s] | PT [s] | Factor | TN + opt [s] | PT [s] | Factor |
|---|---|---|---|---|---|---|
| Conv2d_1a_3x3.conv | $3.42 \cdot 10^{-2}$ | $4.09 \cdot 10^{-3}$ | 8.36 x | $3.43 \cdot 10^{-2}$ | $4.11 \cdot 10^{-3}$ | 8.33 x |
| Conv2d_2a_3x3.conv | $1.58 \cdot 10^{-1}$ | $9.29 \cdot 10^{-2}$ | 1.70 x | $1.91 \cdot 10^{-1}$ | $9.18 \cdot 10^{-2}$ | 2.08 x |
| Conv2d_2b_3x3.conv | $1.56 \cdot 10^{-1}$ | $9.66 \cdot 10^{-2}$ | 1.61 x | $1.88 \cdot 10^{-1}$ | $9.44 \cdot 10^{-2}$ | 1.99 x |
| Conv2d_3b_1x1.conv | $5.26 \cdot 10^{-3}$ | $2.33 \cdot 10^{-3}$ | 2.26 x | $1.74 \cdot 10^{-3}$ | $2.41 \cdot 10^{-3}$ | **0.72 x** |
| Conv2d_4a_3x3.conv | $1.01 \cdot 10^{-1}$ | $6.34 \cdot 10^{-2}$ | 1.58 x | $1.08 \cdot 10^{-1}$ | $6.16 \cdot 10^{-2}$ | 1.76 x |
| Mixed_5b.branch1x1.conv | $4.13 \cdot 10^{-3}$ | $2.02 \cdot 10^{-3}$ | 2.05 x | $2.40 \cdot 10^{-3}$ | $2.13 \cdot 10^{-3}$ | 1.13 x |
| Mixed_5b.branch5x5_1.conv | $4.12 \cdot 10^{-3}$ | $3.66 \cdot 10^{-3}$ | 1.13 x | $2.41 \cdot 10^{-3}$ | $3.65 \cdot 10^{-3}$ | **0.66 x** |
| Mixed_5b.branch5x5_2.conv | $4.29 \cdot 10^{-2}$ | $3.27 \cdot 10^{-2}$ | 1.31 x | $4.39 \cdot 10^{-2}$ | $3.29 \cdot 10^{-2}$ | 1.33 x |
| Mixed_5b.branch3x3dbl_2.conv | $8.57 \cdot 10^{-3}$ | $7.27 \cdot 10^{-3}$ | 1.18 x | $1.38 \cdot 10^{-2}$ | $7.31 \cdot 10^{-3}$ | 1.89 x |
| Mixed_5b.branch3x3dbl_3.conv | $1.72 \cdot 10^{-2}$ | $1.42 \cdot 10^{-2}$ | 1.21 x | $2.46 \cdot 10^{-2}$ | $1.42 \cdot 10^{-2}$ | 1.73 x |
| Mixed_5b.branch_pool.conv | $4.12 \cdot 10^{-3}$ | $2.05 \cdot 10^{-3}$ | 2.01 x | $2.40 \cdot 10^{-3}$ | $2.02 \cdot 10^{-3}$ | 1.19 x |
| Mixed_5c.branch1x1.conv | $5.43 \cdot 10^{-3}$ | $4.89 \cdot 10^{-3}$ | 1.11 x | $3.27 \cdot 10^{-3}$ | $4.89 \cdot 10^{-3}$ | **0.67 x** |
| Mixed_5c.branch5x5_1.conv | $5.40 \cdot 10^{-3}$ | $4.87 \cdot 10^{-3}$ | 1.11 x | $3.27 \cdot 10^{-3}$ | $4.88 \cdot 10^{-3}$ | **0.67 x** |
| Mixed_5d.branch1x1.conv | $7.24 \cdot 10^{-3}$ | $6.66 \cdot 10^{-3}$ | 1.09 x | $4.88 \cdot 10^{-3}$ | $6.68 \cdot 10^{-3}$ | **0.73 x** |
| Mixed_5d.branch5x5_1.conv | $7.25 \cdot 10^{-3}$ | $6.67 \cdot 10^{-3}$ | 1.09 x | $4.88 \cdot 10^{-3}$ | $6.69 \cdot 10^{-3}$ | **0.73 x** |
| Mixed_6a.branch3x3.conv | $7.76 \cdot 10^{-2}$ | $3.28 \cdot 10^{-2}$ | 2.37 x | $7.78 \cdot 10^{-2}$ | $3.23 \cdot 10^{-2}$ | 2.41 x |
| Mixed_6a.branch3x3dbl_3.conv | $1.29 \cdot 10^{-2}$ | $3.50 \cdot 10^{-3}$ | 3.69 x | $1.41 \cdot 10^{-2}$ | $7.15 \cdot 10^{-3}$ | 1.97 x |
| Mixed_6b.branch1x1.conv | $6.56 \cdot 10^{-3}$ | $5.66 \cdot 10^{-3}$ | 1.16 x | $4.80 \cdot 10^{-3}$ | $4.22 \cdot 10^{-3}$ | 1.14 x |
| Mixed_6b.branch7x7_1.conv | $6.55 \cdot 10^{-3}$ | $6.02 \cdot 10^{-3}$ | 1.09 x | $4.80 \cdot 10^{-3}$ | $6.03 \cdot 10^{-3}$ | **0.80 x** |
| Mixed_6b.branch7x7_2.conv | $2.01 \cdot 10^{-3}$ | $3.60 \cdot 10^{-3}$ | **0.56 x** | $1.50 \cdot 10^{-3}$ | $3.58 \cdot 10^{-3}$ | **0.42 x** |
| Mixed_6b.branch7x7_3.conv | $1.92 \cdot 10^{-3}$ | $3.50 \cdot 10^{-3}$ | **0.55 x** | $1.46 \cdot 10^{-3}$ | $3.58 \cdot 10^{-3}$ | **0.41 x** |
| Mixed_6b.branch7x7dbl_2.conv | $1.94 \cdot 10^{-3}$ | $3.54 \cdot 10^{-3}$ | **0.55 x** | $1.45 \cdot 10^{-3}$ | $3.56 \cdot 10^{-3}$ | **0.41 x** |
| Mixed_6b.branch7x7dbl_5.conv | $1.97 \cdot 10^{-3}$ | $3.49 \cdot 10^{-3}$ | **0.56 x** | $1.46 \cdot 10^{-3}$ | $3.49 \cdot 10^{-3}$ | **0.42 x** |
| Mixed_6c.branch7x7_1.conv | $6.59 \cdot 10^{-3}$ | $4.60 \cdot 10^{-3}$ | 1.43 x | $4.80 \cdot 10^{-3}$ | $4.90 \cdot 10^{-3}$ | **0.98 x** |
| Mixed_6c.branch7x7_2.conv | $2.59 \cdot 10^{-3}$ | $5.14 \cdot 10^{-3}$ | **0.50 x** | $2.08 \cdot 10^{-3}$ | $5.08 \cdot 10^{-3}$ | **0.41 x** |
| Mixed_6c.branch7x7_3.conv | $2.58 \cdot 10^{-3}$ | $5.32 \cdot 10^{-3}$ | **0.48 x** | $2.04 \cdot 10^{-3}$ | $5.23 \cdot 10^{-3}$ | **0.39 x** |
| Mixed_6c.branch7x7dbl_2.conv | $2.51 \cdot 10^{-3}$ | $5.32 \cdot 10^{-3}$ | **0.47 x** | $2.05 \cdot 10^{-3}$ | $5.25 \cdot 10^{-3}$ | **0.39 x** |
| Mixed_6c.branch7x7dbl_5.conv | $2.53 \cdot 10^{-3}$ | $5.21 \cdot 10^{-3}$ | **0.49 x** | $2.04 \cdot 10^{-3}$ | $5.12 \cdot 10^{-3}$ | **0.40 x** |
| Mixed_6e.branch7x7_2.conv | $3.35 \cdot 10^{-3}$ | $7.81 \cdot 10^{-3}$ | **0.43 x** | $2.90 \cdot 10^{-3}$ | $7.61 \cdot 10^{-3}$ | **0.38 x** |
| Mixed_6e.branch7x7_3.conv | $3.35 \cdot 10^{-3}$ | $7.52 \cdot 10^{-3}$ | **0.45 x** | $2.91 \cdot 10^{-3}$ | $7.31 \cdot 10^{-3}$ | **0.40 x** |
| AuxLogits.conv0.conv | $1.09 \cdot 10^{-3}$ | $6.14 \cdot 10^{-4}$ | 1.77 x | $3.82 \cdot 10^{-4}$ | $6.10 \cdot 10^{-4}$ | **0.63 x** |
| AuxLogits.conv1.conv | $8.95 \cdot 10^{-4}$ | $1.07 \cdot 10^{-3}$ | **0.84 x** | $8.52 \cdot 10^{-4}$ | $1.09 \cdot 10^{-3}$ | **0.78 x** |
| Mixed_7a.branch3x3_2.conv | $6.56 \cdot 10^{-3}$ | $2.67 \cdot 10^{-3}$ | 2.45 x | $6.98 \cdot 10^{-3}$ | $2.68 \cdot 10^{-3}$ | 2.60 x |
| Mixed_7a.branch7x7x3_4.conv | $6.93 \cdot 10^{-3}$ | $2.93 \cdot 10^{-3}$ | 2.36 x | $7.04 \cdot 10^{-3}$ | $2.94 \cdot 10^{-3}$ | 2.39 x |
| Mixed_7b.branch1x1.conv | $3.27 \cdot 10^{-3}$ | $1.82 \cdot 10^{-3}$ | 1.80 x | $2.39 \cdot 10^{-3}$ | $1.76 \cdot 10^{-3}$ | 1.36 x |
| Mixed_7b.branch3x3_1.conv | $3.66 \cdot 10^{-3}$ | $3.34 \cdot 10^{-3}$ | 1.10 x | $2.83 \cdot 10^{-3}$ | $3.36 \cdot 10^{-3}$ | **0.84 x** |
| Mixed_7b.branch3x3_2a.conv | $2.51 \cdot 10^{-3}$ | $2.34 \cdot 10^{-3}$ | 1.07 x | $3.03 \cdot 10^{-3}$ | $2.32 \cdot 10^{-3}$ | 1.31 x |
| Mixed_7b.branch3x3_2b.conv | $2.43 \cdot 10^{-3}$ | $2.24 \cdot 10^{-3}$ | 1.09 x | $2.98 \cdot 10^{-3}$ | $2.16 \cdot 10^{-3}$ | 1.38 x |
| Mixed_7b.branch3x3dbl_1.conv | $3.70 \cdot 10^{-3}$ | $2.57 \cdot 10^{-3}$ | 1.44 x | $2.83 \cdot 10^{-3}$ | $2.43 \cdot 10^{-3}$ | 1.17 x |
| Mixed_7b.branch3x3dbl_2.conv | $2.03 \cdot 10^{-2}$ | $1.45 \cdot 10^{-2}$ | 1.40 x | $1.94 \cdot 10^{-2}$ | $1.40 \cdot 10^{-2}$ | 1.39 x |
| Mixed_7b.branch_pool.conv | $2.89 \cdot 10^{-3}$ | $1.57 \cdot 10^{-3}$ | 1.84 x | $2.26 \cdot 10^{-3}$ | $1.57 \cdot 10^{-3}$ | 1.44 x |
| Mixed_7c.branch1x1.conv | $7.88 \cdot 10^{-3}$ | $7.66 \cdot 10^{-3}$ | 1.03 x | $6.73 \cdot 10^{-3}$ | $7.66 \cdot 10^{-3}$ | **0.88 x** |
| Mixed_7c.branch3x3_1.conv | $7.88 \cdot 10^{-3}$ | $7.66 \cdot 10^{-3}$ | 1.03 x | $6.73 \cdot 10^{-3}$ | $7.66 \cdot 10^{-3}$ | **0.88 x** |
| Mixed_7c.branch3x3dbl_1.conv | $7.92 \cdot 10^{-3}$ | $7.67 \cdot 10^{-3}$ | 1.03 x | $6.74 \cdot 10^{-3}$ | $7.67 \cdot 10^{-3}$ | **0.88 x** |
| Mixed_7c.branch_pool.conv | $7.92 \cdot 10^{-3}$ | $7.67 \cdot 10^{-3}$ | 1.03 x | $6.74 \cdot 10^{-3}$ | $7.67 \cdot 10^{-3}$ | **0.88 x** |

(i) MobileNetV2, input shape (32, 3, 256, 256)

| Name | TN [s] | PT [s] | Factor | TN + opt [s] | PT [s] | Factor |
|---|---|---|---|---|---|---|
| features.0.0 | $2.30 \cdot 10^{-2}$ | $3.63 \cdot 10^{-3}$ | 6.34 x | $2.24 \cdot 10^{-2}$ | $3.65 \cdot 10^{-3}$ | 6.13 x |
| features.1.conv.0.0 | $2.84 \cdot 10^{-2}$ | $3.71 \cdot 10^{-2}$ | **0.76 x** | $2.84 \cdot 10^{-2}$ | $3.72 \cdot 10^{-2}$ | **0.76 x** |
| features.1.conv.1 | $7.35 \cdot 10^{-3}$ | $5.48 \cdot 10^{-3}$ | 1.34 x | $2.95 \cdot 10^{-3}$ | $5.50 \cdot 10^{-3}$ | **0.54 x** |
| features.2.conv.0.0 | $4.28 \cdot 10^{-3}$ | $2.92 \cdot 10^{-3}$ | 1.47 x | $1.50 \cdot 10^{-3}$ | $2.91 \cdot 10^{-3}$ | **0.51 x** |
| features.2.conv.1.0 | $3.98 \cdot 10^{-2}$ | $2.51 \cdot 10^{-2}$ | 1.59 x | $3.98 \cdot 10^{-2}$ | $2.51 \cdot 10^{-2}$ | 1.59 x |
| features.2.conv.2 | $5.33 \cdot 10^{-3}$ | $4.67 \cdot 10^{-3}$ | 1.14 x | $3.06 \cdot 10^{-3}$ | $5.04 \cdot 10^{-3}$ | **0.61 x** |
| features.3.conv.0.0 | $1.63 \cdot 10^{-3}$ | $1.37 \cdot 10^{-3}$ | 1.19 x | $7.02 \cdot 10^{-4}$ | $1.39 \cdot 10^{-3}$ | **0.50 x** |
| features.3.conv.1.0 | $2.07 \cdot 10^{-2}$ | $3.67 \cdot 10^{-2}$ | **0.56 x** | $2.06 \cdot 10^{-2}$ | $3.68 \cdot 10^{-2}$ | **0.56 x** |
| features.3.conv.2 | $9.72 \cdot 10^{-3}$ | $9.38 \cdot 10^{-3}$ | 1.04 x | $6.47 \cdot 10^{-3}$ | $9.36 \cdot 10^{-3}$ | **0.69 x** |
| features.4.conv.1.0 | $1.15 \cdot 10^{-2}$ | $1.02 \cdot 10^{-2}$ | 1.13 x | $1.15 \cdot 10^{-2}$ | $1.02 \cdot 10^{-2}$ | 1.13 x |
| features.4.conv.2 | $2.82 \cdot 10^{-3}$ | $2.65 \cdot 10^{-3}$ | 1.06 x | $1.77 \cdot 10^{-3}$ | $2.64 \cdot 10^{-3}$ | **0.67 x** |
| features.5.conv.0.0 | $1.05 \cdot 10^{-3}$ | $7.05 \cdot 10^{-4}$ | 1.49 x | $3.84 \cdot 10^{-4}$ | $7.08 \cdot 10^{-4}$ | **0.54 x** |
| features.5.conv.1.0 | $6.38 \cdot 10^{-3}$ | $1.19 \cdot 10^{-2}$ | **0.54 x** | $6.36 \cdot 10^{-3}$ | $1.19 \cdot 10^{-2}$ | **0.53 x** |
| features.5.conv.2 | $3.39 \cdot 10^{-3}$ | $3.16 \cdot 10^{-3}$ | 1.07 x | $2.10 \cdot 10^{-3}$ | $3.18 \cdot 10^{-3}$ | **0.66 x** |
| features.7.conv.1.0 | $3.66 \cdot 10^{-3}$ | $3.66 \cdot 10^{-3}$ | 1.00 x | $3.69 \cdot 10^{-3}$ | $3.67 \cdot 10^{-3}$ | 1.01 x |
| features.7.conv.2 | $1.41 \cdot 10^{-3}$ | $1.28 \cdot 10^{-3}$ | 1.10 x | $7.93 \cdot 10^{-4}$ | $1.28 \cdot 10^{-3}$ | **0.62 x** |
| features.8.conv.0.0 | $9.96 \cdot 10^{-4}$ | $6.18 \cdot 10^{-4}$ | 1.61 x | $3.37 \cdot 10^{-4}$ | $6.26 \cdot 10^{-4}$ | **0.54 x** |
| features.8.conv.1.0 | $2.88 \cdot 10^{-3}$ | $6.25 \cdot 10^{-3}$ | **0.46 x** | $2.87 \cdot 10^{-3}$ | $6.26 \cdot 10^{-3}$ | **0.46 x** |
| features.8.conv.2 | $2.36 \cdot 10^{-3}$ | $2.24 \cdot 10^{-3}$ | 1.06 x | $1.55 \cdot 10^{-3}$ | $2.24 \cdot 10^{-3}$ | **0.69 x** |
| features.11.conv.2 | $2.33 \cdot 10^{-3}$ | $2.22 \cdot 10^{-3}$ | 1.05 x | $1.55 \cdot 10^{-3}$ | $2.24 \cdot 10^{-3}$ | **0.69 x** |
| features.12.conv.0.0 | $9.43 \cdot 10^{-4}$ | $7.06 \cdot 10^{-4}$ | 1.34 x | $3.87 \cdot 10^{-4}$ | $7.07 \cdot 10^{-4}$ | **0.55 x** |
| features.12.conv.1.0 | $4.07 \cdot 10^{-3}$ | $8.89 \cdot 10^{-3}$ | **0.46 x** | $4.04 \cdot 10^{-3}$ | $8.90 \cdot 10^{-3}$ | **0.45 x** |
| features.12.conv.2 | $3.97 \cdot 10^{-3}$ | $3.84 \cdot 10^{-3}$ | 1.03 x | $2.97 \cdot 10^{-3}$ | $3.85 \cdot 10^{-3}$ | **0.77 x** |
| features.14.conv.1.0 | $2.41 \cdot 10^{-3}$ | $2.66 \cdot 10^{-3}$ | **0.91 x** | $2.39 \cdot 10^{-3}$ | $2.66 \cdot 10^{-3}$ | **0.90 x** |
| features.14.conv.2 | $1.50 \cdot 10^{-3}$ | $1.23 \cdot 10^{-3}$ | 1.22 x | $9.00 \cdot 10^{-4}$ | $1.23 \cdot 10^{-3}$ | **0.73 x** |
| features.15.conv.0.0 | $9.14 \cdot 10^{-4}$ | $6.34 \cdot 10^{-4}$ | 1.44 x | $3.38 \cdot 10^{-4}$ | $6.23 \cdot 10^{-4}$ | **0.54 x** |
| features.15.conv.1.0 | $9.60 \cdot 10^{-4}$ | $4.01 \cdot 10^{-3}$ | **0.24 x** | $9.83 \cdot 10^{-4}$ | $4.03 \cdot 10^{-3}$ | **0.24 x** |
| features.15.conv.2 | $2.57 \cdot 10^{-3}$ | $2.35 \cdot 10^{-3}$ | 1.10 x | $1.85 \cdot 10^{-3}$ | $2.35 \cdot 10^{-3}$ | **0.79 x** |
| features.17.conv.2 | $2.57 \cdot 10^{-3}$ | $2.35 \cdot 10^{-3}$ | 1.10 x | $1.85 \cdot 10^{-3}$ | $2.35 \cdot 10^{-3}$ | **0.79 x** |
| features.18.0 | $1.15 \cdot 10^{-3}$ | $7.91 \cdot 10^{-4}$ | 1.46 x | $4.79 \cdot 10^{-4}$ | $7.91 \cdot 10^{-4}$ | **0.61 x** |

### F.6 KFAC-reduce Factor

We compare TN and TN+opt with a PyTorch implementation of the input-based KFAC-reduce factor based on `torch.nn.functional.unfold`. Figure F23 visualizes the performance ratios for different convolution categories. Table F9 contains the detailed run times and performance factors.

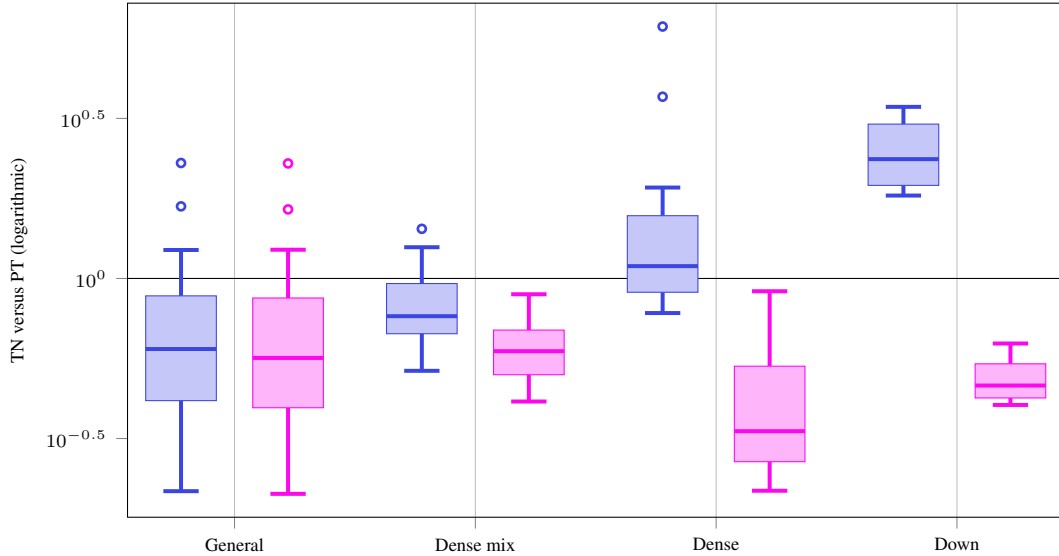

Figure F23: KFAC-reduce factor performance ratios of TN versus PT and TN+opt versus PT for different convolution types on GPU.

**Table F9: KFAC-reduce factor performance comparison on GPU.**

**(a) 3c3d, CIFAR-10, input shape (128, 3, 32, 32)**

| Name | TN [s] | PT [s] | Factor | TN + opt [s] | PT [s] | Factor |
|---|---|---|---|---|---|---|
| conv1.0 | $8.88 \cdot 10^{-4}$ | $2.26 \cdot 10^{-3}$ | **0.39 x** | $8.59 \cdot 10^{-4}$ | $2.41 \cdot 10^{-3}$ | **0.36 x** |
| conv2.0 | $1.41 \cdot 10^{-3}$ | $1.79 \cdot 10^{-3}$ | **0.79 x** | $1.29 \cdot 10^{-3}$ | $1.75 \cdot 10^{-3}$ | **0.74 x** |
| conv3.1 | $1.33 \cdot 10^{-3}$ | $2.31 \cdot 10^{-3}$ | **0.57 x** | $1.46 \cdot 10^{-3}$ | $2.37 \cdot 10^{-3}$ | **0.61 x** |

**(b) F-MNIST 2c2d, input shape (128, 1, 28, 28)**

| Name | TN [s] | PT [s] | Factor | TN + opt [s] | PT [s] | Factor |
|---|---|---|---|---|---|---|
| conv1.1 | $1.10 \cdot 10^{-3}$ | $1.67 \cdot 10^{-3}$ | **0.66 x** | $1.01 \cdot 10^{-3}$ | $1.83 \cdot 10^{-3}$ | **0.55 x** |
| conv2.1 | $1.58 \cdot 10^{-3}$ | $2.57 \cdot 10^{-3}$ | **0.62 x** | $1.54 \cdot 10^{-3}$ | $2.76 \cdot 10^{-3}$ | **0.56 x** |

**(c) CIFAR-100 All-CNN-C, input shape (128, 3, 32, 32)**

| Name | TN [s] | PT [s] | Factor | TN + opt [s] | PT [s] | Factor |
|---|---|---|---|---|---|---|
| conv1.1 | $1.11 \cdot 10^{-3}$ | $1.84 \cdot 10^{-3}$ | **0.60 x** | $1.07 \cdot 10^{-3}$ | $1.89 \cdot 10^{-3}$ | **0.56 x** |
| conv2.1 | $3.26 \cdot 10^{-3}$ | $8.32 \cdot 10^{-3}$ | **0.39 x** | $3.19 \cdot 10^{-3}$ | $8.24 \cdot 10^{-3}$ | **0.39 x** |
| conv3.1 | $3.09 \cdot 10^{-3}$ | $3.56 \cdot 10^{-3}$ | **0.87 x** | $3.08 \cdot 10^{-3}$ | $3.56 \cdot 10^{-3}$ | **0.87 x** |
| conv4.1 | $1.45 \cdot 10^{-3}$ | $3.09 \cdot 10^{-3}$ | **0.47 x** | $1.44 \cdot 10^{-3}$ | $3.09 \cdot 10^{-3}$ | **0.47 x** |
| conv5.1 | $2.59 \cdot 10^{-3}$ | $5.63 \cdot 10^{-3}$ | **0.46 x** | $2.55 \cdot 10^{-3}$ | $5.62 \cdot 10^{-3}$ | **0.45 x** |
| conv6.1 | $2.46 \cdot 10^{-3}$ | $3.49 \cdot 10^{-3}$ | **0.71 x** | $2.43 \cdot 10^{-3}$ | $3.48 \cdot 10^{-3}$ | **0.70 x** |
| conv7.0 | $1.55 \cdot 10^{-3}$ | $3.03 \cdot 10^{-3}$ | **0.51 x** | $1.53 \cdot 10^{-3}$ | $3.02 \cdot 10^{-3}$ | **0.51 x** |
| conv8.1 | $1.14 \cdot 10^{-3}$ | $1.46 \cdot 10^{-3}$ | **0.78 x** | $3.59 \cdot 10^{-4}$ | $1.36 \cdot 10^{-3}$ | **0.26 x** |
| conv9.1 | $1.14 \cdot 10^{-3}$ | $1.46 \cdot 10^{-3}$ | **0.79 x** | $3.59 \cdot 10^{-4}$ | $1.36 \cdot 10^{-3}$ | **0.26 x** |

**(d) Alexnet, input shape (32, 3, 256, 256)**

| Name | TN [s] | PT [s] | Factor | TN + opt [s] | PT [s] | Factor |
|---|---|---|---|---|---|---|
| features.0 | $1.86 \cdot 10^{-3}$ | $4.22 \cdot 10^{-3}$ | **0.44 x** | $1.84 \cdot 10^{-3}$ | $4.22 \cdot 10^{-3}$ | **0.44 x** |
| features.3 | $1.60 \cdot 10^{-3}$ | $3.88 \cdot 10^{-3}$ | **0.41 x** | $1.53 \cdot 10^{-3}$ | $3.89 \cdot 10^{-3}$ | **0.39 x** |
| features.6 | $1.51 \cdot 10^{-3}$ | $1.64 \cdot 10^{-3}$ | **0.92 x** | $1.43 \cdot 10^{-3}$ | $1.63 \cdot 10^{-3}$ | **0.88 x** |
| features.8 | $1.77 \cdot 10^{-3}$ | $3.02 \cdot 10^{-3}$ | **0.59 x** | $1.73 \cdot 10^{-3}$ | $3.02 \cdot 10^{-3}$ | **0.57 x** |
| features.10 | $1.56 \cdot 10^{-3}$ | $1.96 \cdot 10^{-3}$ | **0.79 x** | $1.51 \cdot 10^{-3}$ | $1.96 \cdot 10^{-3}$ | **0.77 x** |

**(e) ResNet18, input shape (32, 3, 256, 256)**

| Name | TN [s] | PT [s] | Factor | TN + opt [s] | PT [s] | Factor |
|---|---|---|---|---|---|---|
| conv1 | $1.79 \cdot 10^{-3}$ | $5.41 \cdot 10^{-3}$ | **0.33 x** | $1.78 \cdot 10^{-3}$ | $5.40 \cdot 10^{-3}$ | **0.33 x** |
| layer1.0.conv1 | $2.24 \cdot 10^{-3}$ | $5.35 \cdot 10^{-3}$ | **0.42 x** | $2.20 \cdot 10^{-3}$ | $5.32 \cdot 10^{-3}$ | **0.41 x** |
| layer2.0.conv1 | $2.23 \cdot 10^{-3}$ | $1.99 \cdot 10^{-3}$ | 1.12 x | $2.16 \cdot 10^{-3}$ | $1.96 \cdot 10^{-3}$ | 1.10 x |
| layer2.0.conv2 | $1.47 \cdot 10^{-3}$ | $3.02 \cdot 10^{-3}$ | **0.49 x** | $1.47 \cdot 10^{-3}$ | $3.04 \cdot 10^{-3}$ | **0.49 x** |
| layer2.0.downsample.0 | $1.85 \cdot 10^{-3}$ | $7.70 \cdot 10^{-4}$ | 2.40 x | $3.24 \cdot 10^{-4}$ | $7.49 \cdot 10^{-4}$ | **0.43 x** |
| layer3.0.conv1 | $1.46 \cdot 10^{-3}$ | $1.21 \cdot 10^{-3}$ | 1.21 x | $1.45 \cdot 10^{-3}$ | $1.21 \cdot 10^{-3}$ | 1.20 x |
| layer3.0.conv2 | $1.49 \cdot 10^{-3}$ | $1.96 \cdot 10^{-3}$ | **0.76 x** | $1.36 \cdot 10^{-3}$ | $1.95 \cdot 10^{-3}$ | **0.70 x** |
| layer3.0.downsample.0 | $1.26 \cdot 10^{-3}$ | $5.44 \cdot 10^{-4}$ | 2.31 x | $2.68 \cdot 10^{-4}$ | $5.44 \cdot 10^{-4}$ | **0.49 x** |
| layer4.0.conv1 | $1.49 \cdot 10^{-3}$ | $1.33 \cdot 10^{-3}$ | 1.12 x | $1.44 \cdot 10^{-3}$ | $1.33 \cdot 10^{-3}$ | 1.08 x |
| layer4.0.conv2 | $1.60 \cdot 10^{-3}$ | $1.86 \cdot 10^{-3}$ | **0.86 x** | $1.62 \cdot 10^{-3}$ | $1.86 \cdot 10^{-3}$ | **0.87 x** |
| layer4.0.downsample.0 | $9.63 \cdot 10^{-4}$ | $5.25 \cdot 10^{-4}$ | 1.83 x | $2.57 \cdot 10^{-4}$ | $4.11 \cdot 10^{-4}$ | **0.63 x** |

**(f) ResNext101, input shape (32, 3, 256, 256)**

| Name | TN [s] | PT [s] | Factor | TN + opt [s] | PT [s] | Factor |
|---|---|---|---|---|---|---|
| conv1 | $1.78 \cdot 10^{-3}$ | $5.38 \cdot 10^{-3}$ | **0.33 x** | $1.77 \cdot 10^{-3}$ | $5.39 \cdot 10^{-3}$ | **0.33 x** |
| layer1.0.conv1 | $1.87 \cdot 10^{-3}$ | $1.73 \cdot 10^{-3}$ | 1.08 x | $4.47 \cdot 10^{-4}$ | $1.73 \cdot 10^{-3}$ | **0.26 x** |
| layer1.0.conv2 | $4.44 \cdot 10^{-2}$ | $1.94 \cdot 10^{-2}$ | 2.29 x | $4.43 \cdot 10^{-2}$ | $1.94 \cdot 10^{-2}$ | 2.29 x |
| layer1.0.conv3 | $6.09 \cdot 10^{-3}$ | $5.57 \cdot 10^{-3}$ | 1.09 x | $1.21 \cdot 10^{-3}$ | $5.57 \cdot 10^{-3}$ | **0.22 x** |
| layer2.0.conv1 | $6.09 \cdot 10^{-3}$ | $5.57 \cdot 10^{-3}$ | 1.09 x | $1.21 \cdot 10^{-3}$ | $5.58 \cdot 10^{-3}$ | **0.22 x** |
| layer2.0.conv2 | $1.37 \cdot 10^{-2}$ | $1.18 \cdot 10^{-2}$ | 1.16 x | $1.37 \cdot 10^{-2}$ | $1.18 \cdot 10^{-2}$ | 1.16 x |
| layer2.0.conv3 | $3.81 \cdot 10^{-3}$ | $3.02 \cdot 10^{-3}$ | 1.26 x | $7.44 \cdot 10^{-4}$ | $3.02 \cdot 10^{-3}$ | **0.25 x** |
| layer2.0.downsample.0 | $6.08 \cdot 10^{-3}$ | $1.77 \cdot 10^{-3}$ | 3.44 x | $7.12 \cdot 10^{-4}$ | $1.77 \cdot 10^{-3}$ | **0.40 x** |
| layer2.1.conv2 | $4.16 \cdot 10^{-3}$ | $9.91 \cdot 10^{-3}$ | **0.42 x** | $4.16 \cdot 10^{-3}$ | $9.90 \cdot 10^{-3}$ | **0.42 x** |
| layer3.0.conv1 | $3.81 \cdot 10^{-3}$ | $3.02 \cdot 10^{-3}$ | 1.26 x | $7.32 \cdot 10^{-4}$ | $3.02 \cdot 10^{-3}$ | **0.24 x** |
| layer3.0.conv2 | $7.88 \cdot 10^{-3}$ | $6.42 \cdot 10^{-3}$ | 1.23 x | $7.90 \cdot 10^{-3}$ | $6.43 \cdot 10^{-3}$ | 1.23 x |
| layer3.0.conv3 | $1.61 \cdot 10^{-3}$ | $1.78 \cdot 10^{-3}$ | **0.91 x** | $5.42 \cdot 10^{-4}$ | $1.80 \cdot 10^{-3}$ | **0.30 x** |
| layer3.0.downsample.0 | $3.80 \cdot 10^{-3}$ | $1.17 \cdot 10^{-3}$ | 3.24 x | $5.10 \cdot 10^{-4}$ | $1.21 \cdot 10^{-3}$ | **0.42 x** |
| layer3.1.conv2 | $2.25 \cdot 10^{-3}$ | $5.41 \cdot 10^{-3}$ | **0.42 x** | $2.26 \cdot 10^{-3}$ | $5.41 \cdot 10^{-3}$ | **0.42 x** |
| layer4.0.conv1 | $1.61 \cdot 10^{-3}$ | $1.77 \cdot 10^{-3}$ | **0.91 x** | $5.44 \cdot 10^{-4}$ | $1.80 \cdot 10^{-3}$ | **0.30 x** |
| layer4.0.conv2 | $4.21 \cdot 10^{-3}$ | $5.29 \cdot 10^{-3}$ | **0.80 x** | $4.21 \cdot 10^{-3}$ | $5.29 \cdot 10^{-3}$ | **0.80 x** |
| layer4.0.conv3 | $1.23 \cdot 10^{-3}$ | $1.45 \cdot 10^{-3}$ | **0.85 x** | $7.68 \cdot 10^{-4}$ | $1.44 \cdot 10^{-3}$ | **0.53 x** |
| layer4.0.downsample.0 | $1.62 \cdot 10^{-3}$ | $8.91 \cdot 10^{-4}$ | 1.82 x | $4.97 \cdot 10^{-4}$ | $8.93 \cdot 10^{-4}$ | **0.56 x** |
| layer4.1.conv2 | $2.18 \cdot 10^{-3}$ | $4.73 \cdot 10^{-3}$ | **0.46 x** | $2.16 \cdot 10^{-3}$ | $4.72 \cdot 10^{-3}$ | **0.46 x** |

**(g) ConvNeXt-base, input shape (32, 3, 256, 256)**

| Name | TN [s] | PT [s] | Factor | TN + opt [s] | PT [s] | Factor |
|---|---|---|---|---|---|---|
| features.0.0 | $1.72 \cdot 10^{-3}$ | $1.02 \cdot 10^{-3}$ | 1.69 x | $7.52 \cdot 10^{-4}$ | $1.01 \cdot 10^{-3}$ | **0.74 x** |
| features.1.0.block.0 | $1.53 \cdot 10^{-2}$ | $4.41 \cdot 10^{-2}$ | **0.35 x** | $1.53 \cdot 10^{-2}$ | $4.40 \cdot 10^{-2}$ | **0.35 x** |
| features.2.1 | $3.80 \cdot 10^{-3}$ | $1.99 \cdot 10^{-3}$ | 1.91 x | $8.44 \cdot 10^{-4}$ | $1.99 \cdot 10^{-3}$ | **0.43 x** |
| features.3.0.block.0 | $8.21 \cdot 10^{-3}$ | $2.22 \cdot 10^{-2}$ | **0.37 x** | $8.19 \cdot 10^{-3}$ | $2.22 \cdot 10^{-2}$ | **0.37 x** |
| features.4.1 | $2.32 \cdot 10^{-3}$ | $1.21 \cdot 10^{-3}$ | 1.92 x | $6.85 \cdot 10^{-4}$ | $1.18 \cdot 10^{-3}$ | **0.58 x** |
| features.5.0.block.0 | $4.62 \cdot 10^{-3}$ | $1.18 \cdot 10^{-2}$ | **0.39 x** | $4.57 \cdot 10^{-3}$ | $1.16 \cdot 10^{-2}$ | **0.40 x** |
| features.6.1 | $1.40 \cdot 10^{-3}$ | $1.10 \cdot 10^{-3}$ | 1.27 x | $9.28 \cdot 10^{-4}$ | $1.02 \cdot 10^{-3}$ | **0.91 x** |
| features.7.0.block.0 | $1.38 \cdot 10^{-3}$ | $6.35 \cdot 10^{-3}$ | **0.22 x** | $1.35 \cdot 10^{-3}$ | $6.34 \cdot 10^{-3}$ | **0.21 x** |

(h) InceptionV3, input shape (32, 3, 299, 299)

| Name | TN [s] | PT [s] | Factor | TN + opt [s] | PT [s] | Factor |
|---|---|---|---|---|---|---|
| Conv2d_1a_3x3.conv | $2.39 \cdot 10^{-3}$ | $2.03 \cdot 10^{-3}$ | 1.18 x | $2.36 \cdot 10^{-3}$ | $2.00 \cdot 10^{-3}$ | 1.18 x |
| Conv2d_2a_3x3.conv | $4.42 \cdot 10^{-3}$ | $1.30 \cdot 10^{-2}$ | **0.34 x** | $4.38 \cdot 10^{-3}$ | $1.30 \cdot 10^{-2}$ | **0.34 x** |
| Conv2d_2b_3x3.conv | $4.33 \cdot 10^{-3}$ | $1.30 \cdot 10^{-2}$ | **0.33 x** | $4.32 \cdot 10^{-3}$ | $1.30 \cdot 10^{-2}$ | **0.33 x** |
| Conv2d_3b_1x1.conv | $1.32 \cdot 10^{-2}$ | $2.16 \cdot 10^{-3}$ | 6.12 x | $5.53 \cdot 10^{-4}$ | $2.16 \cdot 10^{-3}$ | **0.26 x** |
| Conv2d_4a_3x3.conv | $2.72 \cdot 10^{-3}$ | $8.00 \cdot 10^{-3}$ | **0.34 x** | $2.74 \cdot 10^{-3}$ | $8.02 \cdot 10^{-3}$ | **0.34 x** |
| Mixed_5b.branch1x1.conv | $1.43 \cdot 10^{-3}$ | $1.57 \cdot 10^{-3}$ | **0.91 x** | $4.52 \cdot 10^{-4}$ | $1.57 \cdot 10^{-3}$ | **0.29 x** |
| Mixed_5b.branch5x5_1.conv | $1.43 \cdot 10^{-3}$ | $1.57 \cdot 10^{-3}$ | **0.91 x** | $4.53 \cdot 10^{-4}$ | $1.57 \cdot 10^{-3}$ | **0.29 x** |
| Mixed_5b.branch5x5_2.conv | $1.56 \cdot 10^{-3}$ | $3.73 \cdot 10^{-3}$ | **0.42 x** | $1.35 \cdot 10^{-3}$ | $3.72 \cdot 10^{-3}$ | **0.36 x** |
| Mixed_5b.branch3x3dbl_2.conv | $1.57 \cdot 10^{-3}$ | $1.91 \cdot 10^{-3}$ | **0.82 x** | $1.44 \cdot 10^{-3}$ | $1.91 \cdot 10^{-3}$ | **0.76 x** |
| Mixed_5b.branch3x3dbl_3.conv | $1.48 \cdot 10^{-3}$ | $2.66 \cdot 10^{-3}$ | **0.56 x** | $1.41 \cdot 10^{-3}$ | $2.66 \cdot 10^{-3}$ | **0.53 x** |
| Mixed_5b.branch_pool.conv | $1.45 \cdot 10^{-3}$ | $1.59 \cdot 10^{-3}$ | **0.91 x** | $4.62 \cdot 10^{-4}$ | $1.59 \cdot 10^{-3}$ | **0.29 x** |
| Mixed_5c.branch1x1.conv | $1.79 \cdot 10^{-3}$ | $2.00 \cdot 10^{-3}$ | **0.90 x** | $5.48 \cdot 10^{-4}$ | $2.00 \cdot 10^{-3}$ | **0.27 x** |
| Mixed_5c.branch5x5_1.conv | $1.79 \cdot 10^{-3}$ | $2.00 \cdot 10^{-3}$ | **0.90 x** | $5.46 \cdot 10^{-4}$ | $2.00 \cdot 10^{-3}$ | **0.27 x** |
| Mixed_5d.branch1x1.conv | $1.99 \cdot 10^{-3}$ | $2.22 \cdot 10^{-3}$ | **0.90 x** | $5.90 \cdot 10^{-4}$ | $2.21 \cdot 10^{-3}$ | **0.27 x** |
| Mixed_5d.branch5x5_1.conv | $1.97 \cdot 10^{-3}$ | $2.20 \cdot 10^{-3}$ | **0.89 x** | $5.83 \cdot 10^{-4}$ | $2.18 \cdot 10^{-3}$ | **0.27 x** |
| Mixed_6a.branch3x3.conv | $2.81 \cdot 10^{-3}$ | $3.18 \cdot 10^{-3}$ | **0.88 x** | $2.76 \cdot 10^{-3}$ | $3.18 \cdot 10^{-3}$ | **0.87 x** |
| Mixed_6a.branch3x3dbl_3.conv | $1.30 \cdot 10^{-3}$ | $1.22 \cdot 10^{-3}$ | 1.07 x | $1.44 \cdot 10^{-3}$ | $1.29 \cdot 10^{-3}$ | 1.11 x |
| Mixed_6b.branch1x1.conv | $1.49 \cdot 10^{-3}$ | $1.64 \cdot 10^{-3}$ | **0.91 x** | $5.47 \cdot 10^{-4}$ | $1.62 \cdot 10^{-3}$ | **0.34 x** |
| Mixed_6b.branch7x7_1.conv | $1.46 \cdot 10^{-3}$ | $1.64 \cdot 10^{-3}$ | **0.89 x** | $5.60 \cdot 10^{-4}$ | $1.64 \cdot 10^{-3}$ | **0.34 x** |
| Mixed_6b.branch7x7_2.conv | $1.01 \cdot 10^{-3}$ | $1.07 \cdot 10^{-3}$ | **0.94 x** | $7.56 \cdot 10^{-4}$ | $1.10 \cdot 10^{-3}$ | **0.69 x** |
| Mixed_6b.branch7x7_3.conv | $9.45 \cdot 10^{-4}$ | $1.23 \cdot 10^{-3}$ | **0.77 x** | $7.61 \cdot 10^{-4}$ | $1.23 \cdot 10^{-3}$ | **0.62 x** |
| Mixed_6b.branch7x7dbl_2.conv | $1.06 \cdot 10^{-3}$ | $1.23 \cdot 10^{-3}$ | **0.86 x** | $7.62 \cdot 10^{-4}$ | $1.23 \cdot 10^{-3}$ | **0.62 x** |
| Mixed_6b.branch7x7dbl_5.conv | $1.14 \cdot 10^{-3}$ | $1.10 \cdot 10^{-3}$ | 1.04 x | $7.56 \cdot 10^{-4}$ | $1.10 \cdot 10^{-3}$ | **0.69 x** |
| Mixed_6c.branch7x7_1.conv | $1.43 \cdot 10^{-3}$ | $1.62 \cdot 10^{-3}$ | **0.88 x** | $5.47 \cdot 10^{-4}$ | $1.64 \cdot 10^{-3}$ | **0.33 x** |
| Mixed_6c.branch7x7_2.conv | $1.01 \cdot 10^{-3}$ | $1.34 \cdot 10^{-3}$ | **0.75 x** | $7.64 \cdot 10^{-4}$ | $1.35 \cdot 10^{-3}$ | **0.56 x** |
| Mixed_6c.branch7x7_3.conv | $1.07 \cdot 10^{-3}$ | $1.69 \cdot 10^{-3}$ | **0.63 x** | $7.69 \cdot 10^{-4}$ | $1.68 \cdot 10^{-3}$ | **0.46 x** |
| Mixed_6c.branch7x7dbl_2.conv | $8.59 \cdot 10^{-4}$ | $1.67 \cdot 10^{-3}$ | **0.51 x** | $7.67 \cdot 10^{-4}$ | $1.69 \cdot 10^{-3}$ | **0.45 x** |
| Mixed_6c.branch7x7dbl_5.conv | $1.01 \cdot 10^{-3}$ | $1.33 \cdot 10^{-3}$ | **0.76 x** | $7.64 \cdot 10^{-4}$ | $1.35 \cdot 10^{-3}$ | **0.57 x** |
| Mixed_6e.branch7x7_2.conv | $1.01 \cdot 10^{-3}$ | $1.48 \cdot 10^{-3}$ | **0.69 x** | $7.68 \cdot 10^{-4}$ | $1.49 \cdot 10^{-3}$ | **0.51 x** |
| Mixed_6e.branch7x7_3.conv | $9.53 \cdot 10^{-4}$ | $1.77 \cdot 10^{-3}$ | **0.54 x** | $7.38 \cdot 10^{-4}$ | $1.79 \cdot 10^{-3}$ | **0.41 x** |
| AuxLogits.conv0.conv | $9.97 \cdot 10^{-4}$ | $6.04 \cdot 10^{-4}$ | 1.65 x | $3.66 \cdot 10^{-4}$ | $6.58 \cdot 10^{-4}$ | **0.56 x** |
| AuxLogits.conv1.conv | $1.05 \cdot 10^{-3}$ | $1.09 \cdot 10^{-3}$ | **0.97 x** | $9.31 \cdot 10^{-4}$ | $1.09 \cdot 10^{-3}$ | **0.85 x** |
| Mixed_7a.branch3x3_2.conv | $1.30 \cdot 10^{-3}$ | $7.75 \cdot 10^{-4}$ | 1.68 x | $1.26 \cdot 10^{-3}$ | $7.67 \cdot 10^{-4}$ | 1.64 x |
| Mixed_7a.branch7x7x3_4.conv | $1.34 \cdot 10^{-3}$ | $1.14 \cdot 10^{-3}$ | 1.18 x | $1.36 \cdot 10^{-3}$ | $1.14 \cdot 10^{-3}$ | 1.19 x |
| Mixed_7b.branch1x1.conv | $1.03 \cdot 10^{-3}$ | $9.07 \cdot 10^{-4}$ | 1.13 x | $5.08 \cdot 10^{-4}$ | $9.07 \cdot 10^{-4}$ | **0.56 x** |
| Mixed_7b.branch3x3_1.conv | $1.16 \cdot 10^{-3}$ | $9.30 \cdot 10^{-4}$ | 1.25 x | $5.20 \cdot 10^{-4}$ | $9.10 \cdot 10^{-4}$ | **0.57 x** |
| Mixed_7b.branch3x3_2a.conv | $1.13 \cdot 10^{-3}$ | $7.94 \cdot 10^{-4}$ | 1.43 x | $6.89 \cdot 10^{-4}$ | $7.93 \cdot 10^{-4}$ | **0.87 x** |
| Mixed_7b.branch3x3_2b.conv | $1.07 \cdot 10^{-3}$ | $8.53 \cdot 10^{-4}$ | 1.25 x | $7.60 \cdot 10^{-4}$ | $8.51 \cdot 10^{-4}$ | **0.89 x** |
| Mixed_7b.branch3x3dbl_1.conv | $1.16 \cdot 10^{-3}$ | $9.32 \cdot 10^{-4}$ | 1.25 x | $5.38 \cdot 10^{-4}$ | $9.32 \cdot 10^{-4}$ | **0.58 x** |
| Mixed_7b.branch3x3dbl_2.conv | $1.67 \cdot 10^{-3}$ | $1.55 \cdot 10^{-3}$ | 1.08 x | $1.59 \cdot 10^{-3}$ | $1.55 \cdot 10^{-3}$ | 1.02 x |
| Mixed_7b.branch_pool.conv | $1.16 \cdot 10^{-3}$ | $6.79 \cdot 10^{-4}$ | 1.71 x | $5.19 \cdot 10^{-4}$ | $6.85 \cdot 10^{-4}$ | **0.76 x** |
| Mixed_7c.branch1x1.conv | $1.23 \cdot 10^{-3}$ | $1.45 \cdot 10^{-3}$ | **0.85 x** | $7.69 \cdot 10^{-4}$ | $1.44 \cdot 10^{-3}$ | **0.53 x** |
| Mixed_7c.branch3x3_1.conv | $1.21 \cdot 10^{-3}$ | $1.44 \cdot 10^{-3}$ | **0.84 x** | $7.80 \cdot 10^{-4}$ | $1.45 \cdot 10^{-3}$ | **0.54 x** |
| Mixed_7c.branch3x3dbl_1.conv | $1.21 \cdot 10^{-3}$ | $1.43 \cdot 10^{-3}$ | **0.84 x** | $7.66 \cdot 10^{-4}$ | $1.44 \cdot 10^{-3}$ | **0.53 x** |
| Mixed_7c.branch_pool.conv | $1.21 \cdot 10^{-3}$ | $1.44 \cdot 10^{-3}$ | **0.84 x** | $7.80 \cdot 10^{-4}$ | $1.45 \cdot 10^{-3}$ | **0.54 x** |

(i) MobileNetV2, input shape (32, 3, 256, 256)

| Name | TN [s] | PT [s] | Factor | TN + opt [s] | PT [s] | Factor |
|---|---|---|---|---|---|---|
| features.0.0 | $1.90 \cdot 10^{-3}$ | $1.66 \cdot 10^{-3}$ | 1.15 x | $1.91 \cdot 10^{-3}$ | $1.68 \cdot 10^{-3}$ | 1.14 x |
| features.1.conv.0.0 | $2.69 \cdot 10^{-3}$ | $9.87 \cdot 10^{-3}$ | **0.27 x** | $2.70 \cdot 10^{-3}$ | $9.89 \cdot 10^{-3}$ | **0.27 x** |
| features.1.conv.1 | $1.12 \cdot 10^{-2}$ | $3.03 \cdot 10^{-3}$ | 3.70 x | $7.12 \cdot 10^{-4}$ | $3.00 \cdot 10^{-3}$ | **0.24 x** |
| features.2.conv.0.0 | $1.80 \cdot 10^{-3}$ | $1.77 \cdot 10^{-3}$ | 1.02 x | $4.64 \cdot 10^{-4}$ | $1.76 \cdot 10^{-3}$ | **0.26 x** |
| features.2.conv.1.0 | $7.01 \cdot 10^{-3}$ | $9.06 \cdot 10^{-3}$ | **0.77 x** | $6.99 \cdot 10^{-3}$ | $9.06 \cdot 10^{-3}$ | **0.77 x** |
| features.2.conv.2 | $2.59 \cdot 10^{-3}$ | $2.38 \cdot 10^{-3}$ | 1.09 x | $6.08 \cdot 10^{-4}$ | $2.40 \cdot 10^{-3}$ | **0.25 x** |
| features.3.conv.0.0 | $1.44 \cdot 10^{-3}$ | $9.19 \cdot 10^{-4}$ | 1.57 x | $2.96 \cdot 10^{-4}$ | $9.40 \cdot 10^{-4}$ | **0.31 x** |
| features.3.conv.1.0 | $2.99 \cdot 10^{-3}$ | $1.12 \cdot 10^{-2}$ | **0.27 x** | $2.99 \cdot 10^{-3}$ | $1.12 \cdot 10^{-2}$ | **0.27 x** |
| features.3.conv.2 | $3.65 \cdot 10^{-3}$ | $3.38 \cdot 10^{-3}$ | 1.08 x | $7.92 \cdot 10^{-4}$ | $3.40 \cdot 10^{-3}$ | **0.23 x** |
| features.4.conv.1.0 | $3.01 \cdot 10^{-3}$ | $3.76 \cdot 10^{-3}$ | **0.80 x** | $2.99 \cdot 10^{-3}$ | $3.77 \cdot 10^{-3}$ | **0.79 x** |
| features.4.conv.2 | $1.38 \cdot 10^{-3}$ | $1.16 \cdot 10^{-3}$ | 1.19 x | $3.53 \cdot 10^{-4}$ | $1.16 \cdot 10^{-3}$ | **0.30 x** |
| features.5.conv.0.0 | $8.51 \cdot 10^{-4}$ | $5.17 \cdot 10^{-4}$ | 1.65 x | $2.77 \cdot 10^{-4}$ | $5.34 \cdot 10^{-4}$ | **0.52 x** |
| features.5.conv.1.0 | $1.38 \cdot 10^{-3}$ | $3.99 \cdot 10^{-3}$ | **0.34 x** | $1.36 \cdot 10^{-3}$ | $3.99 \cdot 10^{-3}$ | **0.34 x** |
| features.5.conv.2 | $1.68 \cdot 10^{-3}$ | $1.36 \cdot 10^{-3}$ | 1.24 x | $3.94 \cdot 10^{-4}$ | $1.35 \cdot 10^{-3}$ | **0.29 x** |
| features.7.conv.1.0 | $1.37 \cdot 10^{-3}$ | $1.69 \cdot 10^{-3}$ | **0.81 x** | $1.35 \cdot 10^{-3}$ | $1.69 \cdot 10^{-3}$ | **0.80 x** |
| features.7.conv.2 | $8.59 \cdot 10^{-4}$ | $7.05 \cdot 10^{-4}$ | 1.22 x | $2.52 \cdot 10^{-4}$ | $7.00 \cdot 10^{-4}$ | **0.36 x** |
| features.8.conv.0.0 | $8.45 \cdot 10^{-4}$ | $4.92 \cdot 10^{-4}$ | 1.72 x | $2.49 \cdot 10^{-4}$ | $4.93 \cdot 10^{-4}$ | **0.51 x** |
| features.8.conv.1.0 | $1.16 \cdot 10^{-3}$ | $2.36 \cdot 10^{-3}$ | **0.49 x** | $1.12 \cdot 10^{-3}$ | $2.35 \cdot 10^{-3}$ | **0.47 x** |
| features.8.conv.2 | $8.73 \cdot 10^{-4}$ | $9.30 \cdot 10^{-4}$ | **0.94 x** | $3.06 \cdot 10^{-4}$ | $9.29 \cdot 10^{-4}$ | **0.33 x** |
| features.11.conv.2 | $9.89 \cdot 10^{-4}$ | $9.49 \cdot 10^{-4}$ | 1.04 x | $3.06 \cdot 10^{-4}$ | $9.25 \cdot 10^{-4}$ | **0.33 x** |
| features.12.conv.0.0 | $9.55 \cdot 10^{-4}$ | $5.32 \cdot 10^{-4}$ | 1.80 x | $2.50 \cdot 10^{-4}$ | $5.14 \cdot 10^{-4}$ | **0.49 x** |
| features.12.conv.1.0 | $1.51 \cdot 10^{-3}$ | $3.23 \cdot 10^{-3}$ | **0.47 x** | $1.27 \cdot 10^{-3}$ | $3.22 \cdot 10^{-3}$ | **0.39 x** |
| features.12.conv.2 | $1.14 \cdot 10^{-3}$ | $1.24 \cdot 10^{-3}$ | **0.92 x** | $3.94 \cdot 10^{-4}$ | $1.17 \cdot 10^{-3}$ | **0.34 x** |
| features.14.conv.1.0 | $1.51 \cdot 10^{-3}$ | $1.61 \cdot 10^{-3}$ | **0.94 x** | $1.45 \cdot 10^{-3}$ | $1.61 \cdot 10^{-3}$ | **0.90 x** |
| features.14.conv.2 | $1.14 \cdot 10^{-3}$ | $6.83 \cdot 10^{-4}$ | 1.67 x | $3.67 \cdot 10^{-4}$ | $6.80 \cdot 10^{-4}$ | **0.54 x** |
| features.15.conv.0.0 | $9.53 \cdot 10^{-4}$ | $5.23 \cdot 10^{-4}$ | 1.82 x | $2.74 \cdot 10^{-4}$ | $5.23 \cdot 10^{-4}$ | **0.52 x** |
| features.15.conv.1.0 | $1.41 \cdot 10^{-3}$ | $2.25 \cdot 10^{-3}$ | **0.63 x** | $1.37 \cdot 10^{-3}$ | $2.25 \cdot 10^{-3}$ | **0.61 x** |
| features.15.conv.2 | $1.15 \cdot 10^{-3}$ | $8.81 \cdot 10^{-4}$ | 1.31 x | $4.46 \cdot 10^{-4}$ | $8.83 \cdot 10^{-4}$ | **0.51 x** |
| features.17.conv.2 | $1.16 \cdot 10^{-3}$ | $8.80 \cdot 10^{-4}$ | 1.31 x | $4.36 \cdot 10^{-4}$ | $8.58 \cdot 10^{-4}$ | **0.51 x** |
| features.18.0 | $9.51 \cdot 10^{-4}$ | $5.40 \cdot 10^{-4}$ | 1.76 x | $2.50 \cdot 10^{-4}$ | $5.22 \cdot 10^{-4}$ | **0.48 x** |

# G   Memory Evaluation Details (CPU)

Here, we investigate the peak memory consumption of our proposed TN implementations.

## G.1   Theoretical & Empirical Analysis for KFAC-reduce Factor

We assume a two-dimensional convolution with input $\mathbf{X}$ of shape $(C_{\text{in}}, I_1, I_2)$, output of shape $(C_{\text{out}}, O_1, O_2)$ and kernel of shape $(C_{\text{out}}, C_{\text{in}}, K_1, K_2)$. The analysis with a batch dimension is analogous; hence we suppress it here to de-clutter the notation.

The main difference between the default and our proposed TN implementation of $\hat{\mathbf{\Omega}}$ from §3.3 lies in the computation of the averaged unfolded input $[\![\mathbf{X}]\!]^{(\text{avg})} := 1/(O_1 O_2) \mathbf{1}_{O_1 O_2}^{\top} [\![\mathbf{X}]\!]$ which consists of $C_{\text{in}} K_1 K_2$ numbers. In the following, we will look at the extra memory on top of storing the input $\mathbf{X}$, the averaged unfolded input $[\![\mathbf{X}]\!]^{(\text{avg})}$, and the result $\hat{\mathbf{\Omega}}$.

**Default implementation:**   The standard implementation computes $[\![\mathbf{X}]\!]^{(\text{avg})}$ via the unfolded input $[\![X]\!]$ and thus requires extra storage of $C_{\text{in}} K_1 K_2 O_1 O_2$ numbers.

**TN implementation (general case):**   The TN implementation requires storing the averaged index patterns $\mathbf{\Pi}^{(i,\text{avg})} := 1/O_i \sum_{o=1}^{O_i} [\mathbf{\Pi}^{(i)}]_{:,o,:}$ for $i = 1, 2$. These are directly computed via a slight modification of Algorithm D1 and require storing $I_1 K_1 + I_2 K_2$ numbers. In contrast to the default implementation, spatial dimensions are de-coupled and there is no dependency on $C_{\text{in}}$.

**TN implementation (structured case):**   For structured convolutions (Figure 5) we can describe the action of the index pattern tensor through reshape and narrowing operations. ML libraries usually perform these without allocating additional memory. Hence, our symbolic simplifications completely eliminate the allocation of temporary intermediates to compute $[\![\mathbf{X}]\!]^{(\text{avg})}$.

**Empirical results:**   To demonstrate the memory reduction inside the computation of $\hat{\mathbf{\Omega}}$ we measure its peak memory with the `memory-profiler` library and subtract the memory required to store $\mathbf{X}$ and $\hat{\mathbf{\Omega}}$. This approximates the extra internal memory requirement of an implementation. With the setup of §F we report the minimum additional memory over 50 independent runs in Table G10. We consistently observe that the TN implementation has lower peak memory, which is further reduced by our symbolic simplifications (see for example the effect on ResNext101's dense and down-sampling convolutions in Table G10f).

Our theoretical analysis from above suggests that the peak memory difference becomes most visible for many channels with large kernel and output sizes. One example are ConxNeXt-base's `features.1.0.block.0` convolutions with $K_1 = K_2 = 7$, $O_1 = O_2 = 64$, and $C_{\text{in}} = 128$ (Table E4g). For those convolutions, we observe that the default implementation requires an additional $3,140\,\text{MiB}$ ($\approx 3\,\text{GiB!}$) of memory, whereas the TN implementation has zero extra memory demand (Table G10g). This is consistent with our theoretical analysis in that the overhead is storing the unfolded input, which has $(N = 32) \cdot (C_{\text{in}} = 128) \cdot (O_1 = 64) \cdot (O_2 = 64) \cdot (K_1 = 7) \cdot (K_2 = 7) = 822,083,584$ `float32` entries, corresponding to $3,136\,\text{MiB}$.

Table G10: Additional internally required memory to compute the KFAC-reduce factor (measured on CPU). The value 0 indicates that an implementation's peak memory matches the memory consumption of its input $\mathbf{X}$ and result $\hat{\Omega}$.

(a) 3c3d, CIFAR-10, input shape (128, 3, 32, 32)

| Name | TN [MiB] | TN + opt [MiB] | PT [MiB] | Type |
|---|---|---|---|---|
| conv1.0 | 0.0 | 0.0 | 0.0 | General |
| conv2.0 | 0.0 | 0.0 | 0.0 | General |
| conv3.1 | 0.0 | 0.0 | 0.0 | General |

(b) F-MNIST 2c2d, input shape (128, 1, 28, 28)

| Name | TN [MiB] | TN + opt [MiB] | PT [MiB] | Type |
|---|---|---|---|---|
| conv1.1 | 0.0 | 0.0 | 0.0 | General |
| conv2.1 | 0.0 | 0.0 | 0.0 | General |

(c) CIFAR-100 All-CNN-C, input shape (128, 3, 32, 32)

| Name | TN [MiB] | TN + opt [MiB] | PT [MiB] | Type |
|---|---|---|---|---|
| conv1.1 | 0.0 | 0.0 | 0.0 | General |
| conv2.1 | 0.0 | 0.0 | 431 | General |
| conv3.1 | 0.0 | 0.0 | 0.0 | General |
| conv4.1 | 0.0 | 0.0 | 0.0 | General |
| conv5.1 | 0.0 | 0.0 | 215 | General |
| conv6.1 | 0.0 | 0.0 | 0.0 | General |
| conv7.0 | 0.0 | 0.0 | 0.0 | General |
| conv8.1 | 0.0 | 0.0 | 0.0 | Dense |
| conv9.1 | 0.0 | 0.0 | 0.0 | Dense |

(d) Alexnet, input shape (32, 3, 256, 256)

| Name | TN [MiB] | TN + opt [MiB] | PT [MiB] | Type |
|---|---|---|---|---|
| features.0 | 0.0 | 0.0156 | 175 | General |
| features.3 | 0.0 | 0.0 | 186 | General |
| features.6 | 0.0 | 0.0156 | 0.0 | General |
| features.8 | 0.0 | 0.0156 | 93.8 | General |
| features.10 | 0.0 | 0.0195 | 0.0 | General |

(e) ResNet18, input shape (32, 3, 256, 256)

| Name | TN [MiB] | TN + opt [MiB] | PT [MiB] | Type |
|---|---|---|---|---|
| conv1 | 0.0 | 0.0 | 293 | General |
| layer1.0.conv1 | 0.0 | 0.0 | 287 | General |
| layer2.0.conv1 | 31.7 | 0.0 | 71.1 | General |
| layer2.0.conv2 | 0.0 | 0.0 | 143 | General |
| layer2.0.downsample.0 | 0.0 | 0.0 | 0.0 | Down |
| layer3.0.conv1 | 0.0 | 0.0 | 0.0 | General |
| layer3.0.conv2 | 0.0 | 0.0 | 70.8 | General |
| layer3.0.downsample.0 | 0.0 | 0.0 | 0.0 | Down |
| layer4.0.conv1 | 0.0 | 0.0 | 0.0 | General |
| layer4.0.conv2 | 0.0 | 80.3 | 0.0 | General |
| layer4.0.downsample.0 | 0.0 | 0.0 | 0.0 | Down |

(f) ResNext101, input shape (32, 3, 256, 256)

| Name | TN [MiB] | TN + opt [MiB] | PT [MiB] | Type |
|---|---|---|---|---|
| conv1 | 0.0 | 0.0 | 293 | General |
| layer1.0.conv1 | 0.0 | 0.0 | 0.0 | Dense |
| layer1.0.conv2 | 576 | 576 | 1150 | General |
| layer1.0.conv3 | 128 | 0.0 | 127 | Dense |
| layer2.0.conv1 | 128 | 0.0 | 127 | Dense |
| layer2.0.conv2 | 256 | 256 | 575 | General |
| layer2.0.conv3 | 0.0 | 0.0 | 0.0 | Dense |
| layer2.0.downsample.0 | 128 | 0.0 | 19.3 | Down |
| layer2.1.conv2 | 0.0 | 0.0 | 575 | General |
| layer3.0.conv1 | 0.0 | 0.0 | 0.0 | Dense |
| layer3.0.conv2 | 128 | 128 | 288 | General |
| layer3.0.conv3 | 0.0 | 0.0 | 0.0 | Dense |
| layer3.0.downsample.0 | 0.0 | 0.0 | 0.0 | Down |
| layer3.1.conv2 | 0.0 | 0.0 | 288 | General |
| layer4.0.conv1 | 0.0 | 0.0 | 0.0 | Dense |
| layer4.0.conv2 | 0.0 | 0.0 | 144 | General |
| layer4.0.conv3 | 0.0 | 0.0 | 0.0 | Dense |
| layer4.0.downsample.0 | 0.0 | 0.0 | 0.0 | Down |
| layer4.1.conv2 | 0.0 | 0.0 | 144 | General |

(g) ConvNeXt-base, input shape (32, 3, 256, 256)

| Name | TN [MiB] | TN + opt [MiB] | PT [MiB] | Type |
|---|---|---|---|---|
| features.0.0 | 0.0 | 0.0 | 0.0 | Dense |
| features.1.0.block.0 | 0.0 | 0.0 | 3140 | General |
| features.2.1 | 0.0 | 0.0 | 0.0 | Dense |
| features.3.0.block.0 | 0.0 | 0.0 | 1570 | General |
| features.4.1 | 0.0 | 0.0 | 0.0 | Dense |
| features.5.0.block.0 | 0.0 | 0.0 | 784 | General |
| features.6.1 | 0.0 | 0.0 | 0.0 | Dense |
| features.7.0.block.0 | 0.0 | 0.0 | 392 | General |

(h) InceptionV3, input shape (32, 3, 299, 299)

| Name | TN [MiB] | TN + opt [MiB] | PT [MiB] | Type |
|---|---|---|---|---|
| Conv2d_1a_3x3.conv | 54.6 | 0.0 | 73.0 | General |
| Conv2d_2a_3x3.conv | 86.7 | 86.7 | 759 | General |
| Conv2d_2b_3x3.conv | 84.4 | 84.4 | 758 | General |
| Conv2d_3b_1x1.conv | 166 | 0.0 | 0.0 | Dense |
| Conv2d_4a_3x3.conv | 52.0 | 0.0 | 442 | General |
| Mixed_5b.branch1x1.conv | 0.0 | 0.0 | 0.0 | Dense |
| Mixed_5b.branch5x5_1.conv | 0.0 | 0.0 | 0.0 | Dense |
| Mixed_5b.branch5x5_2.conv | 0.0 | 0.0 | 178 | General |
| Mixed_5b.branch3x3dbl_2.conv | 0.0 | 0.0 | 84.8 | General |
| Mixed_5b.branch3x3dbl_3.conv | 0.0 | 0.0 | 128 | General |
| Mixed_5b.branch_pool.conv | 0.0 | 0.0 | 0.0 | Dense |
| Mixed_5c.branch1x1.conv | 0.0 | 0.0 | 0.0 | Dense |
| Mixed_5c.branch5x5_1.conv | 0.0 | 0.0 | 0.0 | Dense |
| Mixed_5d.branch1x1.conv | 42.7 | 0.0 | 0.0 | Dense |
| Mixed_5d.branch5x5_1.conv | 42.8 | 0.0 | 0.0 | Dense |
| Mixed_6a.branch3x3.conv | 0.0 | 0.0 | 0.0 | General |
| Mixed_6a.branch3x3dbl_3.conv | 0.0 | 0.0 | 0.0 | General |
| Mixed_6b.branch1x1.conv | 0.0 | 0.0 | 0.0 | Dense |
| Mixed_6b.branch7x7_1.conv | 0.0 | 0.0 | 0.0 | Dense |
| Mixed_6b.branch7x7_2.conv | 0.0 | 0.0 | 0.0 | Dense mix |
| Mixed_6b.branch7x7_3.conv | 0.0 | 0.0 | 0.0 | Dense mix |
| Mixed_6b.branch7x7dbl_2.conv | 0.0 | 0.0 | 0.0 | Dense mix |
| Mixed_6b.branch7x7dbl_5.conv | 0.0 | 0.0 | 0.0 | Dense mix |
| Mixed_6c.branch7x7_1.conv | 0.0195 | 0.0 | 0.0 | Dense |
| Mixed_6c.branch7x7_2.conv | 0.0156 | 0.0 | 0.0 | Dense mix |
| Mixed_6c.branch7x7_3.conv | 0.0 | 0.0 | 0.0 | Dense mix |
| Mixed_6c.branch7x7dbl_2.conv | 0.0 | 0.0 | 0.0 | Dense mix |
| Mixed_6c.branch7x7dbl_5.conv | 0.0 | 0.0 | 0.0 | Dense mix |
| Mixed_6e.branch7x7_2.conv | 0.0 | 0.0 | 0.0 | Dense mix |
| Mixed_6e.branch7x7_3.conv | 0.0 | 0.0 | 0.0 | Dense mix |
| AuxLogits.conv0.conv | 0.0 | 0.0 | 0.0 | Dense |
| AuxLogits.conv1.conv | 0.0 | 0.0 | 0.0 | General |
| Mixed_7a.branch3x3_2.conv | 0.0 | 0.0 | 0.0 | General |
| Mixed_7a.branch7x7x3_4.conv | 0.0 | 0.0 | 0.0 | General |
| Mixed_7b.branch1x1.conv | 0.0 | 0.0 | 0.0 | Dense |
| Mixed_7b.branch3x3_1.conv | 0.0 | 0.0 | 0.0 | Dense |
| Mixed_7b.branch3x3_2a.conv | 0.0 | 0.0 | 0.0 | Dense mix |
| Mixed_7b.branch3x3_2b.conv | 0.0 | 0.0 | 0.0 | Dense mix |
| Mixed_7b.branch3x3dbl_1.conv | 0.0 | 0.0 | 0.0 | Dense |
| Mixed_7b.branch3x3dbl_2.conv | 0.0 | 0.0 | 0.0 | General |
| Mixed_7b.branch_pool.conv | 0.0 | 0.0 | 0.0 | Dense |
| Mixed_7c.branch1x1.conv | 0.0 | 0.0 | 0.0 | Dense |
| Mixed_7c.branch3x3_1.conv | 0.0 | 0.0 | 0.0 | Dense |
| Mixed_7c.branch3x3dbl_1.conv | 0.0 | 0.0 | 0.0 | Dense |
| Mixed_7c.branch_pool.conv | 0.0 | 0.0 | 0.0 | Dense |

(i) MobileNetV2, input shape (32, 3, 256, 256)

| Name | TN [MiB] | TN + opt [MiB] | PT [MiB] | Type |
|---|---|---|---|---|
| features.0.0 | 0.0 | 0.0 | 53.8 | General |
| features.1.conv.0.0 | 26.1 | 26.1 | 576 | General |
| features.1.conv.1 | 128 | 0.0 | 63.8 | Dense |
| features.2.conv.0.0 | 0.0 | 0.0 | 0.0 | Dense |
| features.2.conv.1.0 | 192 | 192 | 432 | General |
| features.2.conv.2 | 0.0 | 0.0 | 0.0 | Dense |
| features.3.conv.0.0 | 0.0 | 0.0 | 0.0 | Dense |
| features.3.conv.1.0 | 34.1 | 70.4 | 648 | General |
| features.3.conv.2 | 71.7 | 0.0 | 71.4 | Dense |
| features.4.conv.1.0 | 59.5 | 55.7 | 162 | General |
| features.4.conv.2 | 0.0 | 0.0 | 0.0 | Dense |
| features.5.conv.0.0 | 0.0 | 0.0 | 0.0 | Dense |
| features.5.conv.1.0 | 0.0 | 0.0 | 215 | General |
| features.5.conv.2 | 0.0 | 0.0 | 0.0 | Dense |
| features.7.conv.1.0 | 0.0 | 0.0 | 53.3 | General |
| features.7.conv.2 | 0.0 | 0.0 | 0.0 | Dense |
| features.8.conv.0.0 | 0.0 | 0.0 | 0.0 | Dense |
| features.8.conv.1.0 | 0.0 | 0.0 | 107 | General |
| features.8.conv.2 | 0.0 | 0.0 | 0.0 | Dense |
| features.11.conv.2 | 0.0 | 0.0 | 0.0 | Dense |
| features.12.conv.0.0 | 0.0 | 0.0 | 0.0 | Dense |
| features.12.conv.1.0 | 0.0 | 0.0 | 161 | General |
| features.12.conv.2 | 0.0 | 0.0 | 0.0 | Dense |
| features.14.conv.1.0 | 0.0 | 0.0 | 39.7 | General |
| features.14.conv.2 | 0.0 | 0.0 | 0.0 | Dense |
| features.15.conv.0.0 | 0.0 | 0.0 | 0.0 | Dense |
| features.15.conv.1.0 | 0.0 | 0.0 | 63.8 | General |
| features.15.conv.2 | 0.0 | 0.0 | 0.0 | Dense |
| features.17.conv.2 | 0.0 | 0.0 | 0.0 | Dense |
| features.18.0 | 0.0 | 0.0 | 0.0 | Dense |

# H   Miscellaneous

## H.1   Example: Associativity of Tensor Multiplication

Here, we demonstrate associativity of tensor multiplication through an example. The technical challenge is that an index can only be summed once there are no remaining tensors sharing it. Therefore, we must carry indices that are summed in later multiplications in the intermediate results, which requires some set arithmetic on the index sets.

Let $S_1, S_2, S_3$ be index tuples of the input tensors $\mathbf{A}, \mathbf{B}, \mathbf{C}$, and $S_4 \subseteq (S_1 \cup S_2 \cup S_3)$ a valid output index tuple of their tensor multiplication $\mathbf{D} = *_{(S_1, S_2, S_3, S_4)}(\mathbf{A}, \mathbf{B}, \mathbf{C})$. We can either first multiply $\mathbf{A}$ with $\mathbf{B}$ to obtain an intermediate tensor of index structure $S_{1,2}$, or $\mathbf{B}$ with $\mathbf{C}$ to obtain an intermediate tensor of index structure $S_{2,3}$, before carrying out the remaining multiplications. To construct the intermediate index structures, we divide the indices $\tilde{S} = (S_1 \cup S_2 \cup S_3) \setminus S_4$ that are summed over into those only shared between $\mathbf{A}, \mathbf{B}$ given by $\tilde{S}_{1,2} = (S_1 \cup S_2) \setminus (S_4 \cup S_3)$, and those only shared among $\mathbf{B}, \mathbf{C}$ given by $\tilde{S}_{2,3} = (S_2 \cup S_3) \setminus (S_4 \cup S_1)$. This yields the intermediate indices $S_{1,2} = (S_1 \cup S_2) \setminus \tilde{S}_{1,2}$ and $S_{2,3} = (S_2 \cup S_3) \setminus \tilde{S}_{2,3}$, and the parenthesizations

$$[\mathbf{D}]_{S_4} = \left( \sum_{\tilde{S} \setminus \tilde{S}_{1,2}} \left( \sum_{\tilde{S}_{1,2}} [\mathbf{A}]_{S_1} [\mathbf{B}]_{S_2} \right) [\mathbf{C}]_{S_3} \right) = \left( \sum_{\tilde{S} \setminus \tilde{S}_{2,3}} [\mathbf{A}]_{S_1} \left( \sum_{\tilde{S}_{2,3}} [\mathbf{B}]_{S_2} [\mathbf{C}]_{S_3} \right) \right) \quad \text{(H20)}$$

$$\Leftrightarrow \mathbf{D} = *_{(S_{1,2}, S_3, S_4)} \left( *_{(S_2, S_3, S_{2,3})}(\mathbf{A}, \mathbf{B}), \mathbf{C} \right) = *_{(S_1, S_{2,3}, S_4)} \left( \mathbf{A}, *_{(S_1, S_2, S_{1,2})}(\mathbf{B}, \mathbf{C}) \right) .$$

This generalizes to $n$-ary multiplication, allowing to break it down into smaller multiplications. However, the index notation and set arithmetic from Equation (H20) quickly becomes impractical.

## H.2   Example: Matrix-matrix Multiplication as Tensor Multiplication

Here we provide a small self-contained example that demonstrates Equation (3) for matrix-matrix multiplication.

Consider two matrices $\mathbf{A}, \mathbf{B}$ which are compatible for multiplication and let $\mathbf{C} = \mathbf{AB}$. In index notation, we have

$$[\mathbf{C}]_{i,k} = \sum_j [\mathbf{A}]_{i,j} [\mathbf{B}]_{j,k} .$$

The index tuples are $S_{\mathbf{A}} = (i, j)$, $S_{\mathbf{B}} = (j, k)$, and $S_{\mathbf{C}} = (i, k)$. Next, we evaluate which indices are summed over. Since the order of those indices does not matter, we can interpret the tuples as sets and use set arithmetic:

$$(S_{\mathbf{A}} \cup S_{\mathbf{B}}) \setminus S_{\mathbf{C}} = ((i, j) \cup (j, k)) \setminus (i, k) = (j) \setminus (i, k) = (j) .$$

Now we see that matrix-matrix multiplication is a case of tensor multiplication (Equation (3)),

$$[\mathbf{C}]_{S_{\mathbf{C}}} = \sum_{(S_{\mathbf{A}} \cup S_{\mathbf{B}}) \setminus S_{\mathbf{C}}} [\mathbf{A}]_{S_{\mathbf{A}}} [\mathbf{B}]_{S_{\mathbf{B}}} = *_{(S_{\mathbf{A}}, S_{\mathbf{B}}, S_{\mathbf{C}})}(\mathbf{A}, \mathbf{B}) .$$

