# OpenReview forum: "Convolutions and More as Einsum: A Tensor Network Perspective with Advances for Second-Order Methods"
_NeurIPS.cc/2024/Conference — NeurIPS 2024 poster_

### Official Review · Reviewer_vbTD · 2024-07-10

**Soundness:** 3
**Presentation:** 3
**Contribution:** 2
**Rating:** 7
**Confidence:** 3

**Summary:**

This paper provides a tensor network (TN) view of studying convolutional networks, including forward passes, first-order and second order derivatives. Based on these notations, the authors show how to efficiently implement some algorithms such as KFAC and structured dropout. Experimental results show that these implementations have faster computation while being memory efficient.

**Strengths:**

1. The paper is well written with nice tensor network graphs. While many papers in the tensor decomposition field suffer from notorious indexes, the notation in this paper is consistent and clear.

2. Although the TN graphs for matrix multiplications and derivatives themselves are not new, this paper seems to be the first to provide a comprehensive study of these representations and computations in CNNs. Based on these representations, the authors show how to efficiently implement the KFAC and structured dropout algorithm, which is important for optimization.

3. Experiments show the computational and memory efficiency of the proposed TN implementations.

**Weaknesses:**

The authors claimed some potential impacts of the proposed representation, including theoretical and practical benefits. However, they only present applications on the KFAC and structured dropout. Moreover, the derivations seem to require expertise in TNs. Therefore, I feel the potential usefulness of the proposed representation is still unclear.

**Questions:**

The authors show faster speed of KFAC computation. Does this implementation bring better convergence, training speed or results in practice? Is there any result for the trained network?

---

> ### Author Rebuttal · Authors · 2024-08-05
>
> Dear Reviewer vbTD,
>
> thanks for your support! We appreciate the time and effort you put into reviewing our work.
>
> > The authors show faster speed of KFAC computation. Does this implementation bring better convergence, training speed or results in practice?
>
> You have a point that we did not evaluate the impact of our achieved speed-up on a real second-order method. We are happy to inform you that we will add such an experiment to the main text (please see our [global rebuttal](https://openreview.net/forum?id=cDS8WxnMVP&noteId=dKv8oyjgLS) for a detailed description). In this experiment, we were able to halve the computational overhead of a KFAC-based method compared to SGD, and dramatically reduce its peak memory (sometimes by a factor of 2) to a value very close to that of SGD.
>
> A direct consequence of these findings is that they increase training speed, or allow using hardware with less memory.
>
> One could also experiment with more frequent pre-conditioner updates or larger batch sizes to obtain convergence in fewer steps. We believe these are interesting directions to explore in the future that so far have been disregarded due to the high computational extra cost. To produce statistically significant statements and recommendations though, we believe it will take a larger ablation that is beyond the scope of this work.
>
> Thanks again, and let us know if you have further questions.

---

> > ### Comment · Reviewer_vbTD · 2024-08-13
> >
> > Thanks for the authors' response. I will maintain the score.

---

### Official Review · Reviewer_cYbp · 2024-07-12

**Soundness:** 3
**Presentation:** 4
**Contribution:** 4
**Rating:** 7
**Confidence:** 3

**Summary:**

The paper presents a novel approach to simplifying and optimising TN operations for CNNs. The authors introduce an abstraction for tensor networks to efficiently handle convolutions and leverage TN simplifications to improve computational performance. The paper details the theoretical foundations, implementation strategies and results. The results suggest improvements in run-time and memory efficiency over standard methods.

**Strengths:**

The key strengths are:
- The paper includes rigorous mathematical formulations and proofs to support the proposed methods.
- Provides thorough details on implementation, aiding reproducibility and practical adoption.
- Applicable to a wide range of convolutional operations, including those with various hyper-parameters, batching, and channel groups. The generality enhances utility and flexibility
- Good use of visualisations and diagrams to clarify and explain the networks
- Interest to the research community with strong potential for future extensions

**Weaknesses:**

The key weakness is that the performance demonstrated is not very convincing in many cases. This is understandable as the authors mention, standard routines have been aggressively optimised whereas the TN counterparts have been explored significantly less. However this by itself is not a reason to reject given the potential further research that can be built on this work and the interest it

Minor things:
- Line 57 does not appear to have the reference to 2.2 in the right place
- Two pages are spent on the preliminaries. I see the value of having this to set the context for the later sections but I wonder, if some of those were moved to the appendix, what new content could be added to the main method and results?
- The NeurIPS format states "Place one line space before the figure caption and one line space after the figure." Some of the Figure captions are placed next to the figure in a two-column setup which I am not sure if satisfies the style rules
- Standard errors, rather than standard deviation, might be more informative in Figure 8

**Questions:**

Questions for authors:
- Why are the (non simplified) TN implementations slower in general? Do the standard implementations have further efficiency optimisations?
- Are there any practical ways to estimate a priori whether a selected TN has high contraction order variability? Note: to know the approximate variability range, not necessarily the exact values of each order
- Does figure 6 include variability wrt contraction order?
- How many of the operations support GPU/hardware acceleration? Would there be performance gains if they were implemented?

**Limitations:**

There is a section discussing limitations, but the authors could mention more practical factors. Such as: scalability issues and how the methods perform with larger datasets (should be straightforward with synthetic data), potential biases introduced and their impact on different types of data and potential real-world scenarios where the methods might fail or underperform.

---

> ### Author Rebuttal · Authors · 2024-08-05
>
> Dear Reviewer cYbp,
>
> thanks for your strong support and detailed review! We will apply your suggested improvements to the text.
>
> We would like to inform you that we conducted a new experiment with a real second-order method based on KFAC-reduce (see our [global rebuttal](https://openreview.net/forum?id=cDS8WxnMVP&noteId=dKv8oyjgLS)). Our TN implementation was able to halve the second-order method's run time overhead compared to SGD, and dramatically reduce its peak memory (sometimes by a factor of 2). We hope this provides further evidence that our approach is indeed useful to advance second-order methods and close their computational gap.
>
> ## Questions
>
> - > Why are the (non simplified) TN implementations slower in general? Do the standard implementations have further efficiency optimisations?
>
>   This observation is correct. Non-simplified TNs perform contractions using a dense version of $\Pi$, which is sparse. This causes wasteful multiplies by zero. Our simplifications reformulate the contraction with $\Pi$ into `reshape`s or `slice`s, which remove these zero multiplications from the TN (see e.g. the bottom right panel of Fig. 1) and thereby speed up the computation.
>
>   Whenever simplification is not possible, the main difficulty for using the index pattern's sparsity is that PyTorch's `einsum` only accepts dense tensors. We experimented with alternatives that support mixed-format tensors, and have some ideas how to make progress on this frontier in future work. If you are interested in knowing more, feel free to have a look at our [response to Reviewer iQrS](https://openreview.net/forum?id=cDS8WxnMVP&noteId=u5s1zM6z7X).
>
> - > Does figure 6 include variability wrt contraction order?
>
>   This depends on the heuristic used to find a contraction path. Generally speaking, the heuristic depends on the number of tensors in the TN. The tensor networks in Fig. 6 contain at most 6 tensors. After looking into `opt_einsum`'s internals, we believe that the TNs with at most 4 tensors do not contain variability w.r.t. contraction path schedule, while the others might:
>
>   In all our experiments, we used `opt_einsum`'s `contract_path` method with the default `'auto'` strategy to find contraction schedules. `opt_einsum` supports many strategies and according to the [documentation](https://optimized-einsum.readthedocs.io/en/stable/path_finding.html#introduction), using `'auto'` "will select the best of these it can while aiming to keep path finding times below around 1ms". The exact behavior depends on the number of tensors to be contracted and is described [here](https://optimized-einsum.readthedocs.io/en/stable/path_finding.html#performance-comparison): For TNs with up to 4 tensors, the optimal schedule is searched. For TNs with 5-6 tensors, a restricted search is carried out. We are unsure whether this restricted search is deterministic or stochastic, so it may be that this approach causes some extent of variability in contraction order.
>
> - > Are there any practical ways to estimate a priori whether a selected TN has high contraction order variability? Note: to know the approximate variability range, not necessarily the exact values of each order
>
>   This is indeed an interesting question. We did not look into this and decided to rely completely on `opt_einsum`, which uses a sophisticated heuristic based on [performance comparisons](https://optimized-einsum.readthedocs.io/en/stable/path_finding.html#performance-comparison) to strike a good balance between contraction path search time and contraction time. We believe this is a good representation of what a practitioner would do.
>
> - > How many of the operations support GPU/hardware acceleration? Would there be performance gains if they were implemented?
>
>   All operations we propose are purely PyTorch and therefore support GPU execution out of the box.
>
> Thanks again, and let us know if you have further questions.

---

### Official Review · Reviewer_iQrS · 2024-07-17

**Soundness:** 3
**Presentation:** 4
**Contribution:** 3
**Rating:** 5
**Confidence:** 4

**Summary:**

### Summary of "Convolutions and More as Einsum: A Tensor Network Perspective with Advances for Second-Order Methods"

The paper "Convolutions and More as Einsum: A Tensor Network Perspective with Advances for Second-Order Methods" presents a compelling and innovative recast of convolution operations into tensor networks, offering significant benefits for both theoretical understanding and practical applications in machine learning. Here’s a motivated summary with the key points:

1. **Recast of Convolution Operation into Tensor Networks**:
   - The authors provide a nice review and recast of convolutions by viewing them as tensor networks (TNs). This perspective simplifies the analysis of convolutions, allowing for intuitive reasoning about underlying tensor multiplications through easily interpretable diagrams. This approach makes it possible to perform function transformations like differentiation more effectively and efficiently.

2. **Tensor Networks for Machine Learning**:
   - This paper is very good material for studying tensor networks within the context of machine learning. By representing convolutions as TNs, the authors bridge the gap between complex convolutional operations and the more straightforward tensor manipulations, making advanced theoretical concepts accessible to practitioners and researchers in machine learning.

3. **Implementation with Einsum and Opt_Einsum**:
   - The authors implement convolutions using the einsum and opt_einsum functions in PyTorch/Python, providing a detailed comparison of the efficiency of these implementations. The tensor network approach not only accelerates the computation but also reduces memory overhead significantly. The paper demonstrates that tensor network implementations can achieve substantial performance improvements, particularly for less standard routines like Kronecker-factored Approximate Curvature (KFAC).

4. **Insights into Second-Order Optimization**:
   - The implementation of KFAC benefits significantly from the tensor network view of convolutions. The TN perspective provides new insights into second-order optimization methods, allowing for more efficient computation of curvature approximations. The paper shows that the tensor network approach can lead to more frequent pre-conditioner updates, larger batch sizes without memory issues, and extends KFAC to transpose convolutions, demonstrating its broad applicability to second-order optimization in neural networks.

In summary, this paper provides a well-rounded and innovative approach to understanding and optimizing convolutions through tensor networks. It not only offers theoretical insights but also demonstrates practical performance benefits, making it a valuable resource for researchers and practitioners in machine learning.

**Strengths:**

### Strengths of "Convolutions and More as Einsum: A Tensor Network Perspective with Advances for Second-Order Methods"

1. **Innovative Perspective**:
   - The paper introduces an innovative perspective by recasting convolution operations into tensor networks (TNs). This novel approach simplifies the analysis and implementation of convolutions, making complex operations more understandable and accessible.

2. **Advances in Second-Order Optimization**:
   - The paper shows that the tensor network view of convolutions can greatly benefit second-order optimization methods, particularly Kronecker-factored Approximate Curvature (KFAC). This provides new insights and practical improvements in the computation of curvature approximations.

3. **Comprehensive Diagrams and Transformations**:
   - The inclusion of detailed diagrams and specific transformations based on the connectivity pattern of convolutions enhances the clarity and utility of the proposed methods. These visual aids make it easier to understand and implement the tensor network approach.

4. **Flexibility and Modifiability**:
   - The tensor network framework allows for easy modifications and supports various convolutional configurations, including different hyper-parameters, batching, channel groups, and convolution dimensions. This flexibility makes the approach broadly applicable.

5. **Experimental Validation**:
   - The paper provides robust experimental validation, demonstrating the practical benefits of the tensor network approach in various convolutional operations and configurations. The significant speed-ups and memory savings observed in the experiments highlight the effectiveness of the proposed methods.

**Weaknesses:**

### Weaknesses of "Convolutions and More as Einsum: A Tensor Network Perspective with Advances for Second-Order Methods"

While the paper "Convolutions and More as Einsum: A Tensor Network Perspective with Advances for Second-Order Methods" offers numerous strengths, it also has some notable weaknesses:

1. **Suboptimal Use of Einsum for Convolutions**:
   - Using einsum for convolution operations is generally not an optimal approach. The paper itself acknowledges that an optimal method may not exist, as indicated by the mixed results in Tables F4-7, where the performance comparison between the default PyTorch (PT) implementation and the tensor network (TN) approach shows that TN is not consistently faster. This suggests that while TNs offer theoretical benefits, their practical efficiency can vary significantly.

2. **Performance Trade-offs for Specific Architectures**:
   - The einsum-based approach favors specific architectures but makes some of the frequently used models slower. This raises questions about the general applicability of the proposed method. There is a lack of intuition or explanation provided for why certain architectures benefit from the TN approach while others do not. Understanding these trade-offs is crucial for evaluating the practical utility of the method across different use cases.

3. **Limited Comparison with Optimized Convolution Algorithms**:
   - The comparison between TN and PyTorch does not specify which convolutional algorithm is used in PyTorch's implementation. Furthermore, the paper does not compare the TN approach with other highly optimized versions of convolution, such as Winograd or Fourier transform-based methods. These optimized algorithms are known to significantly improve convolution performance, and a comparison with them would provide a more comprehensive assessment of the TN approach's efficiency.

In summary, while the paper presents a novel and theoretically sound method for implementing convolutions using tensor networks, its practical applicability is limited by the suboptimal performance of einsum for convolutions, unexplained performance trade-offs across different architectures, and a lack of comparison with other optimized convolution algorithms. Addressing these weaknesses could enhance the robustness and utility of the proposed approach.

**Questions:**

Please see weakness

**Limitations:**

Please see weakness

---

> ### Author Rebuttal · Authors · 2024-08-05
>
> Dear Reviewer iQrS,
>
> thanks a lot for your thorough review; we appreciate the work you put into it and are glad you find the paper innovative and support our idea of making the complex yet powerful tensor network toolbox accessible to the ML community.
>
> We would like to inform you that we conducted a new experiment with a real second-order method based on KFAC-reduce (see our [global rebuttal](https://openreview.net/forum?id=cDS8WxnMVP&noteId=dKv8oyjgLS)). Our TN implementation was able to halve the second-order method's run time overhead compared to SGD, and dramatically reduce its peak memory (sometimes by a factor of 2). We hope this provides further evidence that our approach is indeed useful to advance second-order methods and close their computational gap.
>
> ## **Weaknesses 1 & 2**
>   > Using einsum for convolution operations is generally not an optimal approach
>
>   > The einsum-based approach favors specific architectures
>
>   You are right that there are additional degrees of freedom to further accelerate our TN implementation. To give some intuition, we would like to point out that one main problem is that we cannot always leverage the index pattern's sparsity: For many convolutions, we can exploit the sparsity thanks to our proposed symbolic simplifications. However, whenever they cannot be applied, we must consider $\Pi$ as dense because PyTorch's `einsum` does not allow contracting tensors with mixed formats. This means we must sometimes carry out multiplies by zero, which is wasteful.
>
> We attempted to address the scenario where our simplifications cannot be applied and have ideas how to progress on this frontier. We will describe them in an updated version of the paper as follows:
>
>   - We experimented with TACO [2], a tensor algebra compiler that can optimize contractions of mixed-format tensors. Unfortunately, TACO's Python frontend cannot yet generate GPU code, and on CPU we ran into a memory leak in the Python frontend.
>
>   - It would be interesting to benchmark our approach on specialized hardware, e.g. [cerebras' wafer-scale engine](https://www.cerebras.net/product-chip/) processor, which [supports unstructured sparsity](https://www.cerebras.net/blog/sparsity-made-easy-introducing-the-cerebras-pytorch-sparsity-library). This would allow to remove many multiplications by zero. While we do not have access to such a chip, we are excited our approach could benefit from such hardware in the future.
>
>
> - To the best of our knowledge, we believe that contraction path optimization of mixed-format tensor networks has not been studied sufficiently. This may be due to the absence of hardware that can leverage this information to speed up contraction. We hypothesize further performance improvements might be possible by taking the operand sparsity into account when optimizing the contraction path.
>
> Please let us know if you think there is an additional approach we could try to boost the performance of our framework.
>
> ## **Weakness 3: Limited comparison with optimized convolution algorithms**
>
> > The comparison between TN and PyTorch does not specify which convolutional algorithm is used in PyTorch's implementation.
>
> You are right that we do not mention which convolution implementation is used in the comparison. We use PyTorch's [`torch.nn.functional.conv2d`](https://pytorch.org/docs/stable/generated/torch.nn.functional.conv2d.html), which automatically selects a 'good' implementation according to heuristics. We think this is a good baseline implementation as most engineers and researchers rely on it in practise.
>
> > Furthermore, the paper does not compare the TN approach with other highly optimized versions of convolution
>
> Consequently you do have a point that we did not look into comparing our approach with different convolution implementations. We found that PyTorch indeed has [different convolution algorithms](https://github.com/pytorch/pytorch/blob/1848cad10802db9fa0aa066d9de195958120d863/aten/src/ATen/native/cudnn/Conv.cpp#L486-L494) implemented. However, it seems like one [cannot specify manually](https://discuss.pytorch.org/t/manually-set-cudnn-convolution-algorithm/101596/2) which algorithm should be used.
>
> We will try to look into this and identify other convolution implementations in PyTorch in order to provide a more fine-grained comparison. If you have any pointers, please let us know.
>
> ## References
>
> [2] Kjolstad, F., Chou, S., Lugato, D., Kamil, S., & Amarasinghe, S. (2017). TACO: A tool to generate tensor algebra kernels. IEEE/ACM International Conference on Automated Software Engineering (ASE).

---

### Author Rebuttal · Authors · 2024-08-05

# Evaluation on a real second-order method

Dear Reviewers,

We are glad to inform you that we successfully applied our work to a real second-order method to complement the speed-ups of fundamental operations shown in the manuscript. Specifically, we took the KFAC-based SINGD optimizer [1] and benchmarked the impact of our TN implementation on memory and run time in comparison to SGD. We will add the experiment to the main text (details below).

**TL;DR: Our TN implementation can reduce SINGD's run time overhead by 50\% and almost completely eliminate the memory overhead. Sometimes, it even reduces the optimizer's memory footprint by a factor of 2. This enables using larger batches or more frequently updating the pre-conditioner.**

We hope that this provides further evidence for the utility of our approach and demonstrates its capability to significantly reduce the computational gap between approximate second-order methods like SINGD and first-order methods like SGD.

Please let us know if you have any follow-up questions. We would be  happy to discuss.

---

**Details:** We applied SINGD with KFAC-reduce and diagonal pre-conditioners to ResNet18 and VGG19 on ImageNet using a batch size of 128. We measured per-iteration time and peak memory on an NVIDIA A40 with 48 GiB of RAM. For SINGD, we compare computing the Kronecker factors with the standard approach via input unfolding versus our TN implementation.

- **ResNet18 with inputs of shape `(128, 3, 256, 256)`:**

  | Optimizer | Per iteration [s] | Peak memory [GiB] |
  |-----------|-------------------|------------------------|
  | SGD | 0.12 (1.0x) | 3.6 (1.0x) |
  | SINGD | 0.19 (1.7x) | 4.5 (1.3x) |
  | **SINGD+TN (ours)** | 0.16 (1.3x) | 3.6 (1.0x) |

  The TN implementation  reduces SINGD's run time overhead compared to SGD by 50\%, and reduces its peak memory to that of SGD.

- **VGG19 with inputs of shape `(128, 3, 256, 256)`:**

  | Optimizer | Per iteration [s] | Peak memory [GiB] |
  |-----------|-------------------|------------------------|
  | SGD | 0.69 (1.0x) | 14 (1.0x) |
  | SINGD | 1.0 (1.5x) | 32 (2.3x) |
  | **SINGD+TN (ours)** | 0.80 (1.2x) | 16 (1.1x) |

  Again, our TN implementation halves SINGD's run time overhead compared to SGD. On this network, it also dramatically lowers the memory overhead, cutting down the peak memory by a factor of 2 from 32 GiB to 16 GiB.

## References

[1] Lin, W., Dangel, F., Eschenhagen, R., Neklyudov, K., Kristiadi, A., Turner, R. E., & Makhzani, A.. Structured inverse-free natural gradient descent: Memory-efficient & numerically-stable KFAC. ICML 2024.

---

### Author Response · Authors · 2024-08-12

We would once again like to thank all reviewers for their time and effort and their constructive feedback which helped us further improve the manuscript.

If there are any remaining questions, please let us know before the end of the discussion phase.

---

### Decision · Program_Chairs · 2024-09-25

**Decision:**

Accept (poster)

**Comment:**

The reviewers all found the paper well-written, well-presented, and the proposed framing of convolutions as tensor networks compelling. Despite some concerns about practical performance, they believe the paper will be of interest to the community. The authors addressed the questions in their rebuttal.